# RANDOM-PROJECTION ENSEMBLE DIMENSION REDUCTION

**Wenxing Zhou, Timothy I. Cannings**
School of Mathematics and Maxwell Institute for Mathematical Sciences
University of Edinburgh
W.Zhou-21@sms.ed.ac.uk, timothy.cannings@ed.ac.uk

## ABSTRACT

We introduce a new, flexible, and theoretically justified framework for dimension reduction in high-dimensional regression, based on an ensemble of random projections. Specifically, we consider disjoint groups of independent random projections, retain the best projection in each group according to the empirical regression performance on the projected covariates, and then aggregate the selected projections via singular value decomposition. The singular values quantify the relative importance of corresponding projection directions and guide the dimension selection process. We investigate various aspects of our framework, including the choice of projection distribution and the number of projections used. Our theoretical results show that the expected estimation error decreases as the number of groups of projections increases. Finally, we demonstrate that our proposal consistently matches or outperforms state-of-the-art methods through extensive numerical studies on simulated and real data.

## 1 INTRODUCTION

In regression, we seek to understand the relationship between a $p$-dimensional vector of predictors $X \in \mathbb{R}^p$ and a response $Y \in \mathbb{R}$ based on $n$ observations. In many modern datasets, $p$ is large and may even exceed the sample size $n$. In these high-dimensional problems, many classical methods suffer from the *curse of dimensionality* and can become intractable. For instance, ordinary least squares is not applicable when $p > n$ because the sample covariance matrix is singular. Moreover, nonparametric methods, which are often based on smoothness assumptions, hinge on having sufficient data around the points of interest for prediction. As a result, the sample size $n$ required for such methods scales exponentially with $p$ (e.g., Wainwright, 2019, Chapter 1.2).

One general strategy to address the curse of dimensionality is to seek a low-dimensional representation of the predictors that preserves all the information in $X$ about $Y$. In particular, *sufficient dimension reduction* (SDR) (Cook, 2007) aims to find a function $f : \mathbb{R}^p \to \mathbb{R}^d$, for some $d < p$, such that $X$ and $Y$ are conditionally independent given $f(X)$. A closely related problem focuses on the so-called *central mean subspace* (Cook & Li, 2002), where the conditional mean of $Y$ given $X$ can be written in terms of $f(X)$. In practice, $f$ is often restricted to be linear, which yields more interpretable representations.

### OUR CONTRIBUTIONS

1. We propose *random projection ensemble dimension reduction (RPEDR)* (Algorithm 1), a novel framework that is distinct from existing SDR methods. This is supplemented by two further algorithms: Algorithm 2 selects an appropriate projection dimension and Algorithm 3 provides a further refinement in some cases (see the diagram in Figure 2).

2. We systematically study the key tuning choices in our framework (e.g., projection distributions, number of random projections) and provide default recommendations.

3. We establish theoretical results for our proposal (Theorem 1 and Theorem 2), and demonstrate its strong performance through extensive simulated and real data experiments.

The main idea in Algorithm 1 is to apply many low-dimensional random projections to the data, fit a base regression model after each projection, and choose *good* projections based on their empirical performance. The selected projections are then aggregated through singular value decomposition (SVD). Figure 1 illustrates our random projection-based method on a toy regression problem. A randomly chosen 2-dimensional projection retains little of the relationship between $X$ and $Y$, whereas a projection carefully selected from a group of $M = 200$ random projections yields more promising results. By repeating this process and aggregating the chosen projections, we are able to retain a large amount of the structure observed after applying the oracle projection.

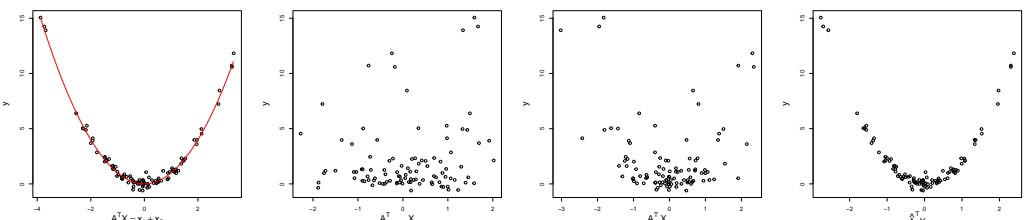

Figure 1: Dimension reduction example ($n = 100$, $p = 20$). True model: $Y = A_0^\top X + 0.3\epsilon$ with $A_0 = (1, 1, 0, \ldots, 0)^\top / \sqrt{2} \in \mathbb{R}^{20}$, $X \sim N_{20}(0, I)$ and $\epsilon \sim N(0, 1)$. Left to right: $Y$ against the oracle projection $x_1 + x_2$ (red line: true regression function); $Y$ against a random projection; $Y$ against the best projection among a group of 200 random projections; $Y$ against the projection estimated by our proposal (with $L = 200$).

Our proposed framework is highly flexible and allows user-specified choices of the random projection distribution and the base regression method, depending on the problem at hand – these choices, along with other practical aspects, are investigated in Section 3 and Appendix A. In particular, Theorem 1 in Section 3.2 shows that the expected estimation error converges at a rate no slower than $L^{-1/2}$, where $L$ is the number of projection groups. This rate is observed in empirical investigations.

One key feature of our method is that the singular values in the SVD step provide a measure of the relative importance of the corresponding singular vectors. These values can therefore be used to determine the projection dimension if needed. In Section 3.3, we propose Algorithm 2 to select the projection dimension by comparing the singular values obtained under our method with those obtained when the group size is one (no selection). We also explore the potential benefits of applying Algorithm 1 a second time on the projected data from the initial application in Section 3.4. This refinement step, formalized in Algorithm 3, can improve performance in certain settings by combining signals and removing irrelevant directions. A more detailed demonstration of Algorithm 3, together with the associated theoretical result (Theorem 2), is provided in Appendix A.5.

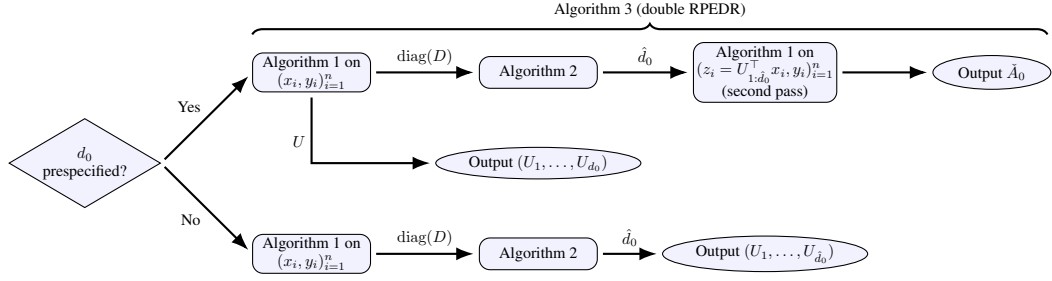

Figure 2: Workflow of Algorithms 1–3.

Section 4 presents the empirical performance of our proposal on simulated data. We compare our method with several other state-of-the-art approaches to SDR. Our proposal is very competitive across a wide range of settings: it achieves the best results in fifteen out of eighteen cases in the simulation study, as well as in six out of ten scenarios in the real-data analysis (see Appendix D), ranking second in all remaining cases. Moreover, in some settings, our method is the only approach that demonstrates non-trivial performance. Notably, in cases where $p > n$, some comparators become intractable, whereas our method remains effective in such regimes.

The remainder of our paper is as follows: an overview of the related work is presented in Section 1.1. In Section 2, we introduce the statistical setting and present our main algorithm. Section 3 focuses on practical considerations and provides guidance on selecting method inputs. The main simulation results are presented in Section 4. We conclude with a discussion in Section 5. The appendices contain additional details on practical considerations (Appendix A), the theoretical proofs (Appendix B), the full results of our simulation study (Appendix C) and real-data experiments (Appendix D).

## 1.1 RELATED WORK

Work on sufficient dimension reduction dates back to the idea of *inverse regression*, which targets the expectation of $X$ given $Y$ rather than regressing $Y$ on $X$. The earliest work is *sliced inverse regression* (SIR) (Li, 1991), which uses 'slices' of $Y$ to estimate $\mathrm{Cov}(X \mid Y)$, the eigenvectors of this estimate are then used as the basis of the dimension reduction subspace. This idea led to a long line of work, including *sliced average variance estimation* (SAVE) (Cook & Weisberg, 1991), *principal Hessian directions* (PHD) (Li, 1992), *sparse SIR* (Li & Nachtsheim, 2006) and *directional regression* (DR) (Li & Wang, 2007). In another line of work, the nonparametric *minimum average variance estimation* (MAVE) (Xia et al., 2002) estimates the dimension reduction subspace by constructing a minimization problem that quantifies how well the model fits the observations. Comprehensive reviews are provided by Ma & Zhu (2013). More recent proposals include *gradient-based kernel dimension reduction* (gKDR) (Fukumizu & Leng, 2014), *dimension reduction and MARS* (drMARS) (Liu et al., 2023), *robust SDR* (Huang et al., 2024a), *sparse kernel SDR* (Liu & Xue, 2024) and *Neural Network based SDR* (Xu & Yu, 2025); see also the survey by Ghojogh et al. (2021).

Random projections offer a natural approach to dimension reduction. The Johnson-Lindenstrauss lemma (Johnson & Lindenstrauss, 1984; Dasgupta & Gupta, 2003) shows that applying a random projection can approximately preserve pairwise distances between observations. However, as illustrated in Figure 1, naively applying a single random projection typically fails. Our proposal is motivated by Cannings & Samworth (2017), who introduced a random-projection ensemble method for classification by aggregating the results of applying a base classifier to the projected data from many carefully selected random projections. Related ideas have been used in sparse principal component analysis (Gataric et al., 2020), sparse sliced inverse regression (Zhang et al., 2025), and semi-supervised learning (Wang et al., 2025). Other works on (unsupervised) dimension reduction using random projection techniques include Bingham & Mannila (2001) and Reeve et al. (2024). Work on other problems includes precision matrix estimation (Marzetta et al., 2011), two-sample testing (Lopes et al., 2011), clustering (Fern & Brodley, 2003; Anderlucci et al., 2022), high-dimensional regression (Thanei et al., 2017; Slawski, 2018; Dobriban & Liu, 2019; Ahfock et al., 2020), and differential privacy (Huang et al., 2024b). See also Xie et al. (2017) and Cannings (2021).

## 2 STATISTICAL SETTING AND METHODOLOGY

Let $P$ denote a distribution on $\mathbb{R}^p \times \mathbb{R}$ and suppose that the covariate-response pair $(X, Y) \sim P$. Let $\eta(x) := \mathbb{E}(Y \mid X = x)$ be the regression function. We seek to find a dimension $d \in \{1, \ldots, p - 1\}$ and a corresponding projection matrix $A \in \mathcal{A}_{p \times d} := \{A \in \mathbb{R}^{p \times d} : A^\top A = I_{d \times d}\}$ for which there exists a function $g : \mathbb{R}^d \to \mathbb{R}$ satisfying

$$\eta(x) = g(A^\top x). \tag{1}$$

Here $A^\top$ maps $X$ to a $d$-dimensional subspace, and the representation $A^\top X$ contains all the information available in $X$ about the conditional mean of $Y$ given $X$. Equation 1 is of course satisfied by taking $d = p$, $A = I_{p \times p}$, and $g = \eta$. The main interest, however, is in finding a solution to equation 1 for a minimal choice of $d$, which we will denote by $d_0$. The corresponding projection matrix and link function will be denoted by $A_0$ and $g_0$, respectively. While $d_0$ is unique, the associated $A_0$ is not. Indeed, if $A_0 \in \mathcal{A}_{p \times d_0}$ and $g_0$ satisfy equation 1, then so do $A_1 = A_0 B$ and $g_1(\cdot) = g_0(B\cdot)$, for any $B \in \mathcal{A}_{d_0 \times d_0}$.

Since $A_0$ is not uniquely identifiable, our focus is on the space spanned by the columns of $A_0$. For a projection matrix $A \in \mathcal{A}_{p \times d}$, we write $\mathcal{S}(A) = \mathrm{span}(A)$ for the $d$-dimensional subspace spanned by the columns of $A$. If $A$ satisfies equation 1, then $\mathcal{S}(A)$ is called a *mean dimension reduction subspace* (Cook & Li, 2002). The space spanned by the columns of the $d_0$-dimensional projection $A_0$, $\mathcal{S}(A_0)$, is called the *central mean subspace (CMS)*.

For the purposes of this paper, we assume the existence and uniqueness of the CMS. This assumption is standard and mild (see, for example, Cook, 1996, Lemma 1), and allows us to focus on our main goal of estimating the space $\mathcal{S}(A_0)$ based on a dataset of $n$ independent and identically distributed pairs $(X_1, Y_1), \ldots, (X_n, Y_n) \sim P$. Our estimate in this problem will be the space spanned by the columns of a projection $\hat{A}_0 \in \mathcal{A}_{p \times \hat{d}_0}$ for some $\hat{d}_0 \in [p]$, where $\hat{d}_0$ and $\hat{A}_0$ are (possibly randomised) functions of the data $\mathcal{D} = \{(X_1, Y_1), \ldots, (X_n, Y_n)\}$. To measure the distance between the spaces $\mathcal{S}(\hat{A}_0)$ and $\mathcal{S}(A_0)$, we will focus on $d_{\mathrm{F}}\big(\mathcal{S}(\hat{A}_0), \mathcal{S}(A_0)\big)$, where

$$d_{\mathrm{F}}^2\big(\mathcal{S}(\hat{A}_0), \mathcal{S}(A_0)\big) := \frac{1}{2}\big\|\hat{A}_0 \hat{A}_0^\top - A_0 A_0^\top\big\|_{\mathrm{F}}^2. \tag{2}$$

The scaling in equation 2 ensures that, if $d_0 = \hat{d}_0$, then $d_{\mathrm{F}}$ is equivalent to the sin-theta distance (Davis & Kahan, 1970; Yu et al., 2015).

## 2.1 RANDOM PROJECTION ENSEMBLE DIMENSION REDUCTION

We now introduce our new procedure for dimension reduction, which is presented in Algorithm 1. We first outline the notation used. Given a dataset of $n_1$ covariate-response pairs $((z_1, y_1), \ldots, (z_{n_1}, y_{n_1})) \in (\mathbb{R}^d \times \mathbb{R})^{n_1}$, a $d$-dimensional data-dependent regression method is a measurable function $\hat{g} : \mathbb{R}^d \times (\mathbb{R}^d \times \mathbb{R})^{n_1} \to \mathbb{R}$. The function $\hat{g}$ uses the data $(z_1, y_1), \ldots, (z_{n_1}, y_{n_1})$ to estimate a regression function $g : \mathbb{R}^d \to \mathbb{R}$. We often drop all but the first argument of $\hat{g}(\cdot; (z_1, y_1), \ldots, (z_{n_1}, y_{n_1}))$ and simply write $\hat{g}(\cdot)$ for the estimated regression function. We write $\mathcal{G}_{d,n_1}$ for the set of all such regression methods. In Algorithm 1, $z_1, \ldots, z_{n_1}$ are different (randomly chosen) projections of a subset of the $p$-dimensional covariate observations $x_1, \ldots, x_n$.

The algorithm also takes as inputs a projection dimension $d \in [p]$; a corresponding projection distribution $Q$ on $\mathcal{Q}_{p \times d} := \{A \in \mathbb{R}^{p \times d} : \mathrm{diag}(A^\top A) = (1, \ldots, 1)^\top\}$; a number of groups of projections $L \in \mathbb{N}$; a group size $M \in \mathbb{N}$; a subsample size $n_1 \in [n-1]$; and a base regression method $\hat{g} \in \mathcal{G}_{d,n_1}$. A full investigation, alongside recommendations on how to choose these inputs in practice is given in Sections 3 and Appendix A.

---

**Algorithm 1:** Random-projection ensemble dimension reduction (RPEDR)

1 **Input**: Data $((x_1, y_1), \ldots, (x_n, y_n)) \in (\mathbb{R}^p \times \mathbb{R})^n$, projection dimension $d \in [p]$, projection distribution $Q$ on $\mathcal{Q}_{p \times d}$, number of groups $L$, group size $M$, sample-split size $n_1 \in [n-1]$, base regression method $\hat{g} \in \mathcal{G}_{d,n_1}$.

2 Let $\mathbf{P}_{1,1}, \ldots, \mathbf{P}_{L,M} \overset{\text{i.i.d.}}{\sim} Q$.

3 **for** $\ell \in [L]$ **do**

4      Let $\mathcal{N}_1$ be a random sample of size $n_1$ chosen without replacement from $\{1, \ldots, n\}$.

5      **for** $m \in [M]$ **do**

6          for $i \in \mathcal{N}_1^c$, let $\hat{h}_{\ell,m}(x_i) = \hat{g}\big(\mathbf{P}_{\ell,m}^\top x_i; (\mathbf{P}_{\ell,m}^\top x_j, y_j)_{j \in \mathcal{N}_1}\big)$.

7          Let $\hat{R}_{\ell,m} = \frac{1}{n-n_1} \sum_{i \in \mathcal{N}_1^c} \big\{y_i - \hat{h}_{\ell,m}(x_i)\big\}^2$.

8      Choose the projection that minimizes the mean-squared error within the group, taking the one with the smallest index in the case of ties: $\mathbf{P}_{\ell,*} := \mathbf{P}_{\ell,m_\ell^*}$, where $m_\ell^* := \mathrm{sargmin}_{m \in [M]} \hat{R}_{\ell,m}$.

9 Set $\hat{\Pi} := \frac{1}{L} \sum_{\ell=1}^{L} \mathbf{P}_{\ell,*} \mathbf{P}_{\ell,*}^\top$.

10 Calculate the singular value decomposition of $\hat{\Pi} = UDU^\top$.

11 **Output**: The matrix $U \in \mathbb{R}^{p \times p}$ and $\mathrm{diag}(D) \in \mathbb{R}^p$.

---

Algorithm 1 starts by generating $L \cdot M$ random projection matrices $\mathbf{P}_{\ell,m} \in \mathbb{R}^{p \times d}$ from the distribution $Q$ and partitioning them into $L$ disjoint groups of size $M$. For each projection $\mathbf{P}_{\ell,m}$, we apply it to all observed covariates, fit the base regressor $\hat{g}$ on a subsample of the projected data, evaluate the mean-squared error on the complement, and retain the best-performing projection in each group. These selected projections $\mathbf{P}_{\ell,*}$ are then aggregated by averaging their outer products to obtain $\hat{\Pi}$. Finally, the algorithm returns the matrix $U$ and the vector $\mathrm{diag}(D)$, provided by the SVD of $\hat{\Pi}$.

The columns of $U$ are the estimated dimension reduction directions, and the singular values in $D$ provide information about the relative importance of each direction. If a target dimension $\hat{d}_0$ is

predetermined, then we set $\hat{A}_0 = (U_1, \ldots, U_{\hat{d}_0})$ as our estimate of $A_0$. Otherwise, the singular values provide guidance – we propose a data-dependent choice of $\hat{d}_0$ in Algorithm 2 in section 3.3.

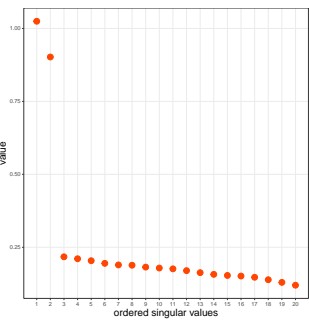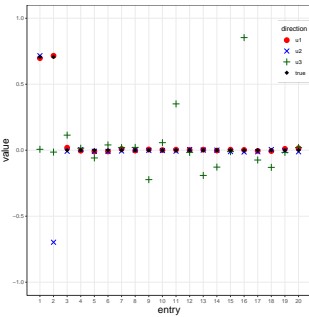

Figure 3: Output of Algorithm 1 applied to the data used in Figure 1. Inputs: $d = 5$, $L = 200$, $M = 200$, $n_1 = 66$, $Q$: entries are rescaled standard Cauchy, $\hat{g}$: quadratic least squares. Left: $\mathrm{diag}(D)$. Right: entries of $U_1$ (red filled circles), $U_2$ (blue $\times$), $U_3$ (green $+$), true dimension reduction direction (black filled diamonds).

To demonstrate how our approach works, we applied Algorithm 1 to the data used in Figure 1. In Figure 3, the first two entries of $D$ are relatively large, suggesting that the singular vectors $U_1$ and $U_2$ capture most of the signal. In this case, the true SDR direction $A_0$ is one-dimensional, and nearly all the signal is found in the first singular vector, with $|U_1^\top A_0| = 0.996$. The second singular vector contains only a small portion of the signal, with $|U_2^\top A_0| = 0.048$. The remaining entries of $D$ are small, suggesting that $U_3, \ldots, U_p$ can be discarded (indeed, $\max_{j=3,\ldots,p} |U_j^\top A_0| < 0.01$).

## 3 PRACTICAL CONSIDERATIONS

### 3.1 RANDOM PROJECTION GENERATING DISTRIBUTION

In this section, we investigate how the choice of distribution for generating random projections affects Algorithm 1. Projections with independent Gaussian entries are widely used in the literature and have desirable rotational invariance properties (see, for example, Vershynin, 2018, Proposition 3.3.2). On the other hand, Cauchy random projections can offer distinct advantages in certain applications - for example, Li et al. (2007) extended the Johnson-Lindenstrauss lemma to the $\ell_1$-norm, and Ramirez et al. (2012) used them for sparse signal reconstruction. As the columns of a projection represent directions in the original covariate space, we generate random projections slightly differently from standard practice. Formally, we construct a $p \times d$ projection matrix $\mathbf{P}$ by drawing $d$ i.i.d. unit-norm columns $P_j \in \mathbb{R}^p$ and concatenating them, i.e., $\mathbf{P} = (P_1, \ldots, P_d)$. We focus on two choices for the distribution of these columns:

*1. Gaussian columns:* We write $P_j \sim Q_{\mathrm{N}}$ if $P_j \overset{d}{=} Z/\|Z\|$, where $Z \sim N_p(0, I)$.

*2. Cauchy columns:* We write $P_j \sim Q_{\mathrm{C}}$ if $P_j \overset{d}{=} W/\|W\|$, where $W = (W_1, \ldots, W_p)$ and $W_1, \ldots, W_p \overset{\text{i.i.d.}}{\sim} \mathrm{Cauchy}(0, 1)$.

We conduct a numerical experiment using three different versions of the following model:

**Model 1**: Let $X \sim N_p(0, I)$ and $Y = 2(A_0^\top X)^2 + \epsilon$, where $\epsilon \sim N(0, \frac{1}{4})$, $A_0^\top = \frac{1}{\sqrt{q}}(\mathbf{1}_q, \mathbf{0}_{p-q})$, for $q \in [p]$. Here $\mathbf{1}_q = (1, \ldots, 1) \in \mathbb{R}^q$ and $\mathbf{0}_{p-q} = (0, \ldots, 0) \in \mathbb{R}^{p-q}$.

We set $p = 20$ and vary $q$ to simulate different sparsity levels: $q = 2$ (Model 1a), $q = 10$ (Model 1b) and $q = 20$ (Model 1c). Three different projection distributions are considered, namely $Q_{\mathrm{N}}^{\otimes d}$ (Gaussian), $Q_{\mathrm{C}}^{\otimes d}$ (Cauchy) and $\frac{1}{2}Q_{\mathrm{N}}^{\otimes d} + \frac{1}{2}Q_{\mathrm{C}}^{\otimes d}$ (mixture, i.e., with probability $\frac{1}{2}$ we draw $\mathbf{P} \sim Q_{\mathrm{N}}^{\otimes d}$, and with probability $\frac{1}{2}$ we draw $\mathbf{P} \sim Q_{\mathrm{C}}^{\otimes d}$). The goal is to examine how the projection-generation choice affects performance across sparsity levels. Signal strength is designed to be the same across settings, and all other inputs to Algorithm 1 are held fixed.

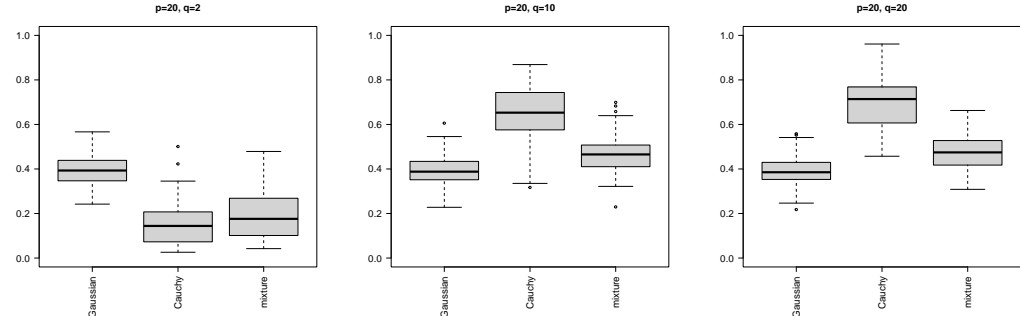

Figure 4: Boxplots of the sin-theta distance (see equation 2) between $U_1$ from the output of Algorithm 1 and the true direction $A_0^\top = \frac{1}{\sqrt{q}}(\mathbf{1}_q, \mathbf{0}_{p-q})$ over 100 repeats of the simulation. We present the results for three different instances of Model 1, namely when $q = 2$ (left), $q = 10$ (middle) and $q = 20$ (right). We compare the results when the projection distribution $Q$ is $Q_N^{\otimes d}$ (Gaussian), $Q_C^{\otimes d}$ (Cauchy), and $\frac{1}{2}Q_N^{\otimes d} + \frac{1}{2}Q_C^{\otimes d}$ (mixture).

Figure 4 shows that, in sparse models, Cauchy projections outperform Gaussian projections; however, the heavy-tailed Cauchy distribution is less suitable in settings without sparsity. Due to the rotational invariance of the problem, Gaussian projections exhibit the same performance regardless of the sparsity level. Using a 50-50 mixture of Gaussian and Cauchy projections yields good performance in both sparse and dense settings. Therefore, when the sparsity level of $A_0$ is unknown, we recommend using this mixture as the default option. This strategy will be employed in all of the numerical experiments presented later in the paper.

## 3.2 CHOICE OF $L$, $M$ AND $d$

Algorithm 1 combines the results of evaluating the empirical performance of the base regression method across a total of $L \cdot M \cdot d$ projection directions. More precisely, we aggregate the results of $L$ ($d$-dimensional) projections, where each projection is selected as the best from a disjoint group of size $M$. In this section, we explore how varying these parameters affects performance. As we will show, the performance typically improves as $L$ increases, in the sense that the sin-theta distance decreases at rate $L^{-1/2}$ (assuming the other quantities fixed); see Theorem 1 and our numerical results in Figure 9. Overall, these results lead us to recommend $L = 200$ as the default. The effects of $M$ and $d$ are slightly less straightforward, and their choices are influenced by the ambient dimension $p$. We recommend $M = 10p$ and $d = \min\{\lceil p^{1/2} \rceil, 10\}$.

We focus here on the choice of $L$; details regarding the choices of $M$ and $d$ are provided in Appendix A.2. Theorem 1 below compares the sin-theta distance between the first $d_0$ columns of $U$ from the output of Algorithm 1 run with $L$ groups of projections, with the corresponding output when considering an infinite-simulation version of the algorithm which takes "$L = \infty$". More precisely, given data $\{(x_1, y_1), \dots, (x_n, y_n)\}$ (here treated as fixed pairs in $\mathbb{R}^p \times \mathbb{R}$), $d \in [p]$, a projection distribution $Q$ on $\mathcal{Q}_{p \times d}$, $L \in \mathbb{N}$, $M \in \mathbb{N}$, $n_1 \in [n-1]$ and $\hat{g} \in \mathcal{G}_{d,n_1}$, let $\hat{A}_0^L := \hat{A}_0 = (U_1, \dots, U_{d_0})$ denote the first $d_0$ columns of the output $U$. We explicitly emphasise the dependence on $L$ and assume the true $d_0$ is known. Further define $\Pi^\infty := \mathbb{E}(\mathbf{P}_{\ell,*}\mathbf{P}_{\ell,*}^\top) \in \mathbb{S}_{p \times p}$ to be the expected value of $\hat{\Pi}$ in line 9 of Algorithm 1; the expectation is taken only over the randomness in the projections and the sample-split in line 4 of the algorithm, but not over the data. Let $\hat{A}_0^\infty := (U_1^\infty, \dots, U_{d_0}^\infty)$, where $U_1^\infty, \dots, U_p^\infty$ are the singular vectors of $\Pi^\infty$.

**Theorem 1.** *We have*

$$\mathbb{E}d_F\big(\mathcal{S}(\hat{A}_0^L), \mathcal{S}(A_0)\big) \le 2d_0^{1/2}\|\Pi^\infty - A_0 A_0^\top\|_{op} + \frac{2(2\pi)^{1/2}d_0^{1/2}dp}{L^{1/2}}.$$

The proof of Theorem 1 is given in Appendix B. The second term in the bound suggests that the error of our algorithm decreases at rate $L^{-1/2}$ as $L$ increases. The first term does not depend on $L$ and can be seen as the infinite-simulation error of our method; it depends on the choices of $M$ and

$d$ (as well as the base regression method and projection distribution) and would be small when, on average, the projections selected are close to $A_0$. See Appendix A.2 for a numerical demonstration.

### 3.3 CHOICE OF $\hat{d}_0$

In some scenarios, the user may have a predetermined target dimension, $\hat{d}_0$, in mind. In such cases, one should simply retain the first $\hat{d}_0$ columns of $U$ from Algorithm 1 and set $\hat{A}_0 = (U_1, \ldots, U_{\hat{d}_0})$. Indeed, the singular vectors of $U$ can be interpreted as the most frequently selected dimensions (in decreasing order of importance) across projection groups in the algorithm. When the desired dimension is unknown, we propose to use the information contained in the singular values $D$ from the output of Algorithm 1. The main idea is to compare these singular values with the component-wise median of those obtained by applying the same procedure to random projections with the same distribution, but without selection within each group (i.e., $M = 1$); see Algorithm 2 below. We use the median rather than the mean, as it provides a more stable summary of the no-selection random projection baseline and is less sensitive to outliers, which can occur under heavy-tailed $Q$.

---

**Algorithm 2:** RPEDR Dimension Estimator

1 **Input**: Projection dimension $d \in [p]$, projection distribution $Q$ on $\mathcal{Q}_{p \times d}$, number of groups $L$, and the vector $\mathrm{diag}(D) = (D_1, \ldots, D_p)$ of singular values from the output of Algorithm 1, as well as a random projection resample size $R \in \mathbb{N}$.

2 Let $\mathbf{P}_{1,1}, \ldots, \mathbf{P}_{L,R} \overset{\text{i.i.d.}}{\sim} Q$.

3 **for** $r \in [R]$ **do**

4      Set $\hat{\Pi}^{(r)} = \frac{1}{L} \sum_{\ell=1}^{L} \mathbf{P}_{\ell,r} \mathbf{P}_{\ell,r}^{\top}$.

5      Calculate the singular value decomposition of $\hat{\Pi}^{(r)} = U^{(r)} D^{(r)} (U^{(r)})^{\top}$ and let $\mathrm{diag}(D^{(r)}) = (D_1^{(r)}, \ldots, D_p^{(r)})$.

6 Let $T_j = \frac{1}{R} \sum_{r=1}^{R} \mathbb{1}_{\{D_j^{(r)} \leq D_j\}}$

7 **Output**: The dimension $\hat{d}_0 := \max\{j \in [p] : T_\ell > 1/2, \text{ for all } \ell \leq j\}$. (Here the maximum over the empty set is taken to be zero, interpreted as evidence that there is no correlation between $X$ and $Y$.)

---

The motivation is that, for a particular direction to be selected, there should be evidence that it carries more signal than would be expected at random. For the leading singular vector $U_1$ (with associated value $D_1$), we require evidence that this direction is more likely to be selected by our algorithm than under random chance. For subsequent components, note that since our projections are rescaled to have trace $d$, we have $\sum_{j=1}^{p} D_j = \mathrm{trace}(\hat{\Pi}) = d$. This induces a natural penalty for selecting more dimensions when earlier singular values in $D$ are relatively large.

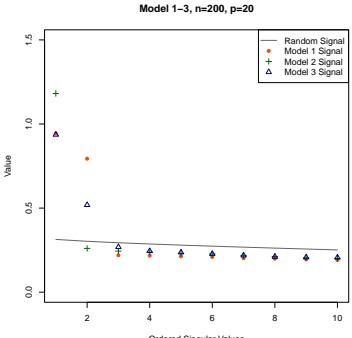
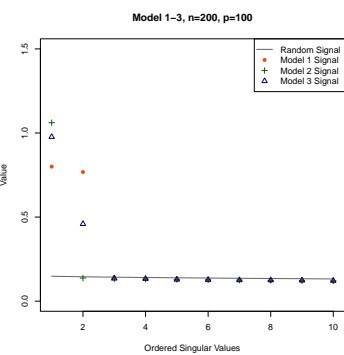

Figure 5: Comparison between the vector $D$ of singular values from the output of Algorithm 1 for Models 1a, 2 and 3, and the median of corresponding $D^{(r)}$ in Algorithm 2 for $p = 20$ (left) and $p = 100$ (right). We present only the first 10 singular values in each case.

Figure 5 illustrates the procedure for selecting $\hat{d}_0$ in Models 1a, 2 and 3. For Model 1a, the output suggests retaining the first two singular vectors. This is partly because, as seen in Figure 3, the single

signal $\frac{1}{\sqrt{2}}(X_1 + X_2)$ is often split into two. The reason is, in part, that even though the direction $\frac{1}{\sqrt{2}}(1, -1, 0, \ldots, 0)^\top$ is orthogonal to the true signal direction, the response $Y$ is not (marginally) independent of $X_1 - X_2$. As a result, our algorithm suggests selecting (approximately) this direction. We demonstrate in Section 3.4 how applying our algorithm twice in such cases can accurately recover only the true one-dimensional signal. For the other two models, there is a clear indication that we should take $\hat{d}_0 = d_0$, i.e., $\hat{d}_0 = 1$ in Model 2 and $\hat{d}_0 = 2$ in Model 3, and in fact, we recover most of the signal: for Model 2, $|U_1^\top e_3| > 0.999$; for Model 3, $\min\{|U_1^\top e_6|, |U_2^\top e_7|\} \geq 0.995$. Further evaluation of $\hat{d}_0$ selection can be found in the full simulation study in Appendix C.

### 3.4 DOUBLE RANDOM PROJECTION ENSEMBLE DIMENSION REDUCTION

In this subsection, we explore how applying our main algorithm a second time may help to further reduce the dimension in cases where the user is not satisfied with the suggested dimension from an initial application of Algorithms 1 and 2. This double application strategy is particularly effective when we have a desired dimension, $\check{d}_0$ say, in mind and the number of covariates contributing to $\mathcal{S}(A_0)$ exceeds $\check{d}_0$. In such situations, a single application of our algorithm may favour the coordinate axes of the relevant covariates, but fail to combine these effectively into the smallest possible number of dimension reduction directions.

One perspective of this approach, is that the first application of our algorithm acts as a screening step to select potentially relevant directions, while the second application then combines these directions to yield a lower-dimensional projection that still retains a large amount of the signal. The full procedure is presented in Algorithm 3, and a detailed demonstration is provided in Appendix A.5, where we also provide an extension of Theorem 1 applicable to this algorithm (see Theorem 2).

---

**Algorithm 3:** Double RPEDR

---

1 **Input**: Data $((x_1, y_1), \ldots, (x_n, y_n)) \in (\mathbb{R}^p \times \mathbb{R})^n$ and desired dimension $\check{d}_0 \in [p]$.
2 Let $U$ and $D$ be the output of Algorithm 1 applied to $(x_1, y_1), \ldots, (x_n, y_n)$ with $L = 200$, $M = 10p$, $d = \min\{\lceil p^{1/2} \rceil, 10\}$, $n_1 = \lceil 2n/3 \rceil$, $Q = \frac{1}{2}Q_N^{\otimes d} + \frac{1}{2}Q_C^{\otimes d}$, and $\hat{g} = \hat{g}_{MARS}$.
3 Let $\hat{d}_0$ be the output of Algorithm 2 applied to $D$, with $L = 200$, $d = \min\{\lceil p^{1/2} \rceil, 10\}$, $Q = \frac{1}{2}Q_N^{\otimes d} + \frac{1}{2}Q_C^{\otimes d}$.
4 **if** $\hat{d}_0 > \check{d}_0$ **then**
5      Set $\hat{A}_0 := (U_1, \ldots, U_{\hat{d}_0})$ and $z_i := \hat{A}_0^\top x_i$ for $i \in [n]$.
6      Let $\check{U}$ and $\check{D}$ be the output of Algorithm 1 applied to $(z_1, y_1), \ldots, (z_n, y_n)$ with $L = 200$, $M = 10\hat{d}_0$, $d = \min\{\lceil \sqrt{\hat{d}_0} \rceil, 10\}$, $n_1 = \lceil 2n/3 \rceil$, $Q = \frac{1}{2}Q_N^{\otimes d} + \frac{1}{2}Q_C^{\otimes d}$, and $\hat{g} = \hat{g}_{MARS}$.
7      Let $\check{A}_0 = \hat{A}_0(\check{U}_1, \ldots, \check{U}_{\check{d}_0}) \in \mathbb{R}^{p \times \check{d}_0}$
8 **else**
9      Let $\check{A}_0 = (U_1, \ldots, U_{\check{d}_0}) \in \mathbb{R}^{p \times \check{d}_0}$
10 **Output**: The projection $\check{A}_0$.

---

**Additional practical considerations** – including the selection of base regressor, the choice of the split ratio $n_1/n$, and computational notes – are presented in Appendix A.

### 4 NUMERICAL SIMULATIONS

In this section, we summarize the main findings from a large simulation study comparing the performance of our proposal with several existing methods for estimating the central mean subspace $\mathcal{S}(A_0)$; the full experimental design and complete results are provided in Appendix C. We evaluate three versions of our proposed method (see also Figure 2). The first variant applies Algorithm 1 on its own, and the second is the double dimension reduction approach presented in Algorithm 3. Both variants are suitable when the true projection dimension $d_0$ is known. The third variant, which combines Algorithm 1 and 2, is designed for situations where $d_0$ is unknown. For comparison, we use the default implementations of several existing methods, including SIR (Li, 1991), pHd (Li, 1992), MAVE (Xia et al., 2002), DR (Li & Wang, 2007), gKDR (Fukumizu & Leng, 2014), and drMARS

(Liu et al., 2023). The models used in our experiments include Model 1 (with $q = 2$, introduced in Section 3.1), as well as eight additional models given in Appendix A.1, A.5, and C.

In the main text we focus on the slightly simpler scenario where the dimension $d_0$ of the central mean subspace $\mathcal{S}(A_0)$ is known. In this case, our first proposal, denoted as **RPE** in Figures 6 and 7, sets $\hat{A}_0 = (U_1, \ldots, U_{d_0})$, where $U = (U_1, \ldots, U_p)$ is the output of Algorithm 1 with the default inputs recommended. In particular, we set $d = \min\{\lceil p^{1/2} \rceil, 10\}$, $L = 200$, $M = 10p$, $n_1 = \lceil 2n/3 \rceil$, $Q = \frac{1}{2}Q_N^{\otimes d} + \frac{1}{2}Q_C^{\otimes d}$, and $\hat{g} = \hat{g}_{\text{MARS}}$. We also present the performance of Algorithm 3 with $\check{d}_0 = d_0$, which is denoted **RPE2** in Figures 6 and 7. For the competing methods, we use the R packages `dr` (Weisberg, 2002) for SIR and pHd, `MAVE` (Hang & Xia, 2021) for MAVE, and the relevant code available via GitHub for DR (`https://github.com/JSongLab/SDR_HC`), gKDR and drMARS (`https://github.com/liuyu-star/drMARS`). In each case, we use the corresponding default recommendations to select appropriate tuning parameters.

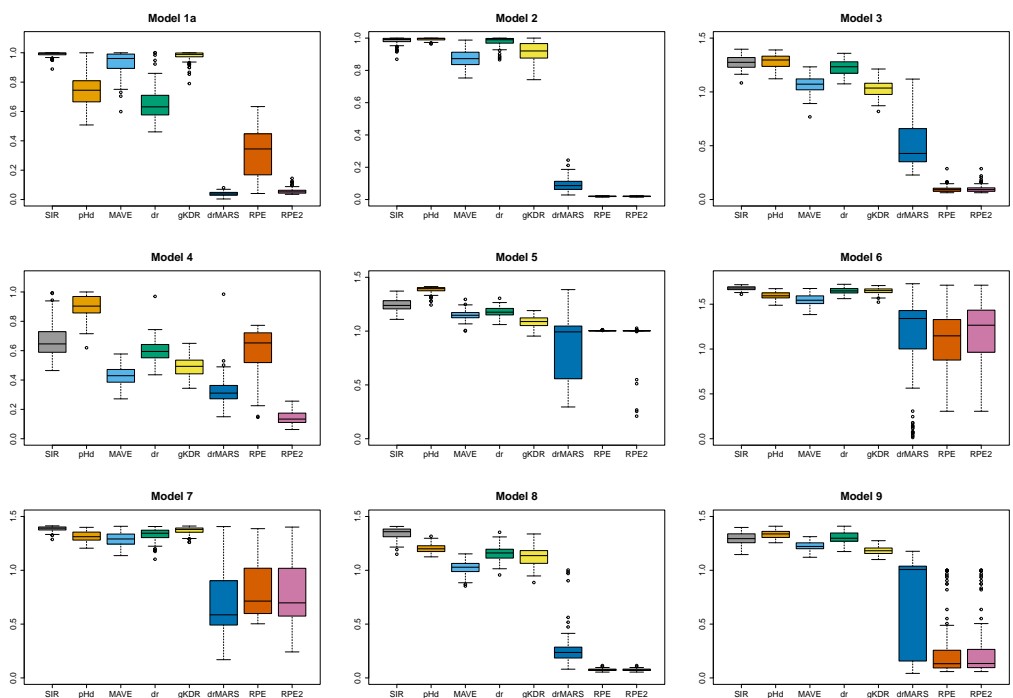

Figure 6: Boxplots of sin-theta distance between the subspaces spanned by the estimated projection $\hat{A}_0$ and true projection $A_0$ for our RPE-based methods and the six competing methods when $d_0$ is known. We present the results for models 1a-9 over 100 repeats of the experiment, with $n = 200$ and $p = 50$.

Figures 6 and 7 present boxplots of the sin-theta distance $d_F(\mathcal{S}(\hat{A}_0), \mathcal{S}(A_0))$ when $n = 200$ and $p \in \{50, 500\}$ setting across 100 replicates for the nine models. We observe that both RPE and RPE2 approaches are highly effective. In Models 1a and 4, RPE2 demonstrates a clear improvement over RPE, as highlighted in Section 3.4. For the other models, a single application of Algorithm 1 performs very well. The drMARS approach is also competitive in Models 1a, 5, 6 and 7. The other competitors suffer prohibitively due to the curse of dimensionality and are typically ineffective in the 50-dimensional examples shown here; when $p = 500$, SIR, pHd and DR are intractable because $p > n$. In the high-dimensional setting, our RPE-based methods show even more pronounced advantages: RPE or RPE2 achieve the best performance across all nine models. There are a few cases where none of the methods achieve a sin-theta distance close to zero: In Models 5 and 7, our algorithm tends to find the space spanned by $e_1$ and $e_2$, but misses the weak signal in the $e_3$ direction; Model 6 is particularly challenging, as all three covariates $X_1$, $X_2$ and $X_3$ are needed to obtain a non-trivial prediction, and the regression problem remains difficult even given the oracle projection. We observe better performance in these models when $p = 20$ and $n = 500$ – see Appendix C.

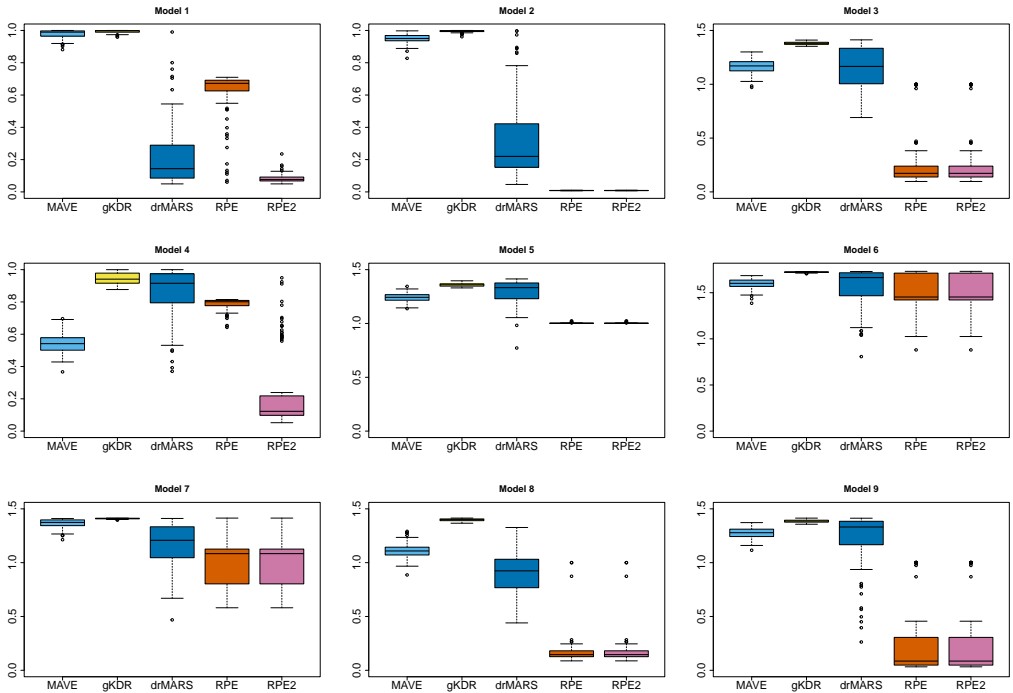

Figure 7: Boxplots of sin-theta distance between the subspaces spanned by the estimated projection $\hat{A}_0$ and true projection $A_0$ for our RPE-based methods and the three competing methods when $d_0$ is known. We present the results for models 1a-9 over 100 repeats of the experiment, with $n = 200$ and $p = 500$.

## 5 DISCUSSION AND EXTENSIONS

We introduced a new general approach to sufficient dimension reduction, based on aggregating an ensemble of carefully chosen projections. The framework is highly flexible, allowing different distributions of random projections and a user-specified choice of base regression method. We have provided default recommendations for these aspects and other tuning parameters based on extensive numerical experiments. While our current theory focuses primarily on the effect of $L$, extending the theory to other tuning parameters and developing more automated defaults are promising directions for future work. Our default implementation prioritises statistical performance and can be slower than classical SDR methods. However, the computation is highly parallel across projection groups and candidates and, as discussed in Appendix C.3, competitive performance is often achievable with modest runtimes using multi-core CPU parallelism. Therefore, an obvious direction for further improving scalability is to exploit GPU acceleration.

There are several ways in which our framework could be extended further. First, a natural assumption in high-dimensional settings is sparsity, specifically the case where the projection $A_0$ is sparse. In this scenario, it may be more effective to consider sparse random projections; perhaps more importantly, employing sparse SVD (see, for example, Yang et al., 2014) to aggregate the chosen projections can encourage a sparse estimate $\hat{A}_0$. Second, in the context of additive index models (see, for example, Friedman & Stuetzle, 1981; Ruan & Yuan, 2010), we may seek to sequentially find the signal directions one by one. For the first direction, apply Algorithm 1 and keep only the leading singular vector. To identify the subsequent directions, one could then apply a modification of our procedure that always includes the signals found so far and considers random projections that are orthogonal to the previously identified signals. Finally, while this paper has focused on dimension reduction—specifically estimating the projection matrix $A_0$—it would be interesting to consider alternative ways of aggregating the results when the primary objective is to predict the response $Y$. In particular, improved predictive performance might be achieved by directly aggregating the predictions from the chosen projections, as opposed to fitting a new regression model after projecting the data using the estimated projection $\hat{A}_0$. We leave these extensions for future investigation.

## REPRODUCIBILITY STATEMENT

Code to implement our algorithms in `R` is submitted in a .zip file as supplementary material, and is also available at `https://github.com/Wenxing99/RPEDR`. The algorithms are fully specified in Algorithm 1 (RPEDR), Algorithm 2 (dimension selection), and Algorithm 3 (double RPEDR) in Sections 2, 3.3 and 3.4, respectively. Default settings (base regressor, projection distribution, $L, M, d, n_1/n$) and implementation details are summarized in Section 3 and Appendix A. The full simulation design (including all nine models, different values of $n$ and $p$) is reported in Section 4 and Appendix C. Real-data preprocessing, splits and evaluation are given in Appendix D. Complete proofs of our theoretical results appear in Appendix B.

## ACKNOWLEDGMENTS

We sincerely thank the Area Chair and the three reviewers for their comprehensive and insightful comments, which helped to improve the paper.

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

## A FURTHER PRACTICAL CONSIDERATIONS

### A.1 CHOICE OF BASE REGRESSION METHOD

The choice of base regression method $\hat{g} \in \mathcal{G}_{d,n_1}$ used in Algorithm 1 depends on two primary factors: first, the method should be computationally efficient, as it will be applied $L \cdot M$ times to different randomly projected datasets; second, the base regression method must capture at least some of the signal after a *good* projection has been applied. We will show that a global polynomial-based method is often effective, even when the true signal is not exactly polynomial. However, there are scenarios in which a more flexible nonparametric approach is needed.

We investigate four options in detail. In each case, we describe how $\hat{g} \in \mathcal{G}_{d,m}$ is defined based on a dataset $(z_1, y_1), \ldots, (z_m, y_m) \in (\mathbb{R}^d \times \mathbb{R})^m$ of size[1] $m > 1 + d(d+3)/2$ and a test point $z \in \mathbb{R}^d$.

1. Global linear least squares (LLS): $\hat{g}_{\mathrm{GL}}(z) = \hat{\alpha} + \hat{\beta}^\top z$, where

$$(\hat{\alpha}, \hat{\beta}) := \underset{(\alpha, \beta) \in \mathbb{R} \times \mathbb{R}^d}{\arg\min} \left\{ \sum_{i=1}^m (y_i - \alpha - \beta^\top z_i)^2 \right\}.$$

2. Global quadratic least squares (QLS): $\hat{g}_{\mathrm{GQ}}(z) := \hat{a} + \hat{b}^\top z + z^\top \hat{C} z$, where

$$(\hat{a}, \hat{b}, \hat{C}) := \underset{(a,b,C) \in \mathbb{R} \times \mathbb{R}^d \times \mathbb{S}_{d \times d}}{\arg\min} \left\{ \sum_{i=1}^m (y_i - a - b^\top z_i - z_i^\top C z_i)^2 \right\}, \tag{3}$$

   and $\mathbb{S}_{d \times d}$ denotes the set of $d \times d$ symmetric matrices.

3. Nadaraya–Watson: Fix a kernel $K$ and a bandwidth $h > 0$, then let

$$\hat{g}_{\mathrm{NW}}(z) = \frac{\sum_{i=1}^m y_i K(h^{-1} \|z_i - z\|)}{\sum_{i=1}^m K(h^{-1} \|z_i - z\|)}.$$

   In our experiments below, we take $K(t) = \frac{e^{-t^2/2}}{\sqrt{2\pi}}$ and $h = 0.1$.

4. Multivariate Adaptive Regression Splines (MARS): Let $\hat{g}_{\mathrm{MARS}}$ denote the estimator produced by the forward and backward passes (Algorithms 2 and 3) of Friedman (1991), which fits piecewise linear models to the data and automatically selects interactions between variables. We use the implementation of this method from the R-package `earth` (Milborrow, 2024). The maximum degree of interaction is taken to be 3.

While many alternative base regression methods are available, we focus on these four methods for their balance between flexibility and computational feasibility.

To demonstrate how these different options perform in practice, we carry out a further set of experiments using three different models:

- Model 1a: as used in Section 3.1 with $A_0 = (\mathbf{1}_2, \mathbf{0}_{p-2})^\top$.
- Model 2: Let $X = (X_1, \ldots, X_p)^\top \sim \mathrm{Unif}([-1, 1]^p)$ and $Y = 2\sin(2\pi X_3) + \epsilon$ with $\epsilon \sim N(0, 1/4)$ independent of $X$.
- Model 3[2]: Let $X = (X_1, \ldots, X_p)^\top \sim N_p(0, I)$ and

$$Y = \frac{X_6}{1/2 + (X_7 + 3/2)^2} + \epsilon$$

  with $\epsilon \sim N(0, 1/4)$ independent of $X$. Here $A_0 = (e_6, e_7)$ is 2-dimensional, where $e_j$ denotes the $j$th canonical Euclidean basis vector in $\mathbb{R}^p$.

---

[1]this sample size requirement is mild and ensures that we can apply the global quadratic method below – if it is not satisfied, then either $d$ should be reduced, or we are limited to using the linear least squares method which only requires $m > d$.

[2]Model 3 was used in the simulation studies in Li (1991); Xia et al. (2002); Liu et al. (2023).

In each case, we set $p = 20$ and $n = 200$, and apply Algorithm 1 with $d = 5$, $L = 200$, $M = 200$, $n_1 = \lceil 2n/3 \rceil$, and $Q = \frac{1}{2} Q_{\mathrm{N}}^{\otimes d} + \frac{1}{2} Q_{\mathrm{C}}^{\otimes d}$.

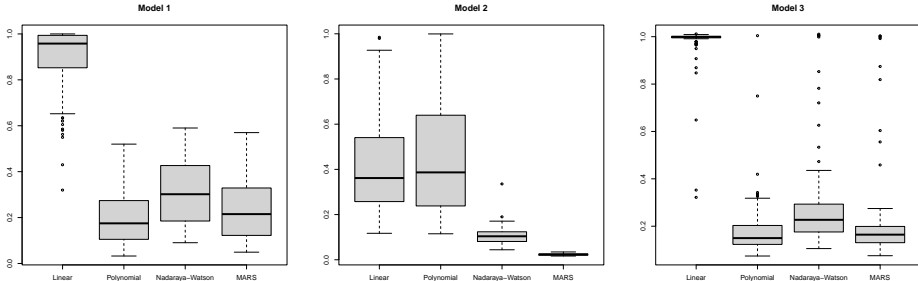

Figure 8: Boxplots of the sin-theta distance (see equation 2) between $U_{1:d_0}$ from the output of Algorithm 1 and the true direction $A_0$ over 100 repeats of the simulation. We present the results for Model 1a (left), Model 2 (middle) and Model 3 (right), for $p = 20$ and $n = 200$. The base method is linear least squares, quadratic least squares, Nadaraya–Watson or MARS.

Figure 8 exhibits some interesting behaviour. In Model 1a, the LLS is not effective because no linear signal exists in any $d$-dimensional projection of the data. For a similar reason, the linear and quadratic least squares methods do not perform well in Model 2. However, in Model 1a, the quadratic base method is very effective, as this model is correctly specified after applying the oracle projection $A_0$. MARS also performs well here, although at a higher computational cost. For Model 2, the flexibility of the nonparametric methods is required to find the signal, and MARS outperforms Nadaraya-Watson. The results for Model 3 show that, even when the base model is incorrectly specified, good performance is still achievable. Here the true regression function after the oracle projection is not polynomial. The linear base method captures the (approximately) linear signal in $X_6$, but misses the signal in $X_7$. However, the signal is sufficiently well-approximated by a quadratic function, allowing us to recover much of it using the quadratic base method; the slightly more computationally expensive MARS method performs similarly to the quadratic least squares.

Based on these results, MARS is recommended as a suitable default base regression method. It performs well across all examples presented here and across a wide range of settings in our large simulation study (Section 4 and Appendix C).

## A.2 CHOICE OF $L$, $M$ AND $d$

To further elucidate the effect of $L$ in practice, we conduct a numerical experiment using versions of Models 1a, 2 and 3 from the previous subsection with $p = 20$ and $p = 100$; see Figure 9. We set $n = 200$, with the other inputs to Algorithm 1 being $d = 5$, $M = 10p$, $n_1 = \lceil 2n/3 \rceil$, $Q = \frac{1}{2} Q_{\mathrm{N}}^{\otimes d} + \frac{1}{2} Q_{\mathrm{C}}^{\otimes d}$ for the projection distribution, and $\hat{g}$ set to MARS. The number of projection groups $L$ varies from 10 to 1000.

The empirical results in Figure 9 confirm that, in these examples, performance of our method improves as $L$ increases, in line with Theorem 1. In Model 1a, we observe greater variability across repeats – partly because, in this example, our algorithm captures only part of the true signal within the leading singular vector of $U$ in some repeats. As also discussed in Section 3.3 and Appendix C, nearly all of the signal is found in the first two singular vectors (which is not fully reflected by the sin-theta distance measure presented here). In fact, the extension in Section 3.4 is able to find the majority of the signal in a one-dimensional projection using a double application of our procedure.

We now turn to the choice of $M$ and $d$, which together determine the total number of projection directions considered within each of the $L$ groups in Algorithm 1. In contrast to the choice of $L$, there is a trade-off when choosing $d$ and $M$. For instance, if $M$ is taken to be small (e.g., less than 20 when $p = 20$), it is unlikely that a good projection appears within a block of size $M$. On the other hand, if $M$ is taken to be very large (e.g., greater than 1000 when $p = 20$), we may start to overfit– although this effect is often negligible due to the data resampling strategy taken in Algorithm 1. In our experiments, we see that the performance tends to level off rather than deteriorate as $M$ increases. Regarding the projection dimension $d$, if $d$ is too small–especially if it is smaller than (the

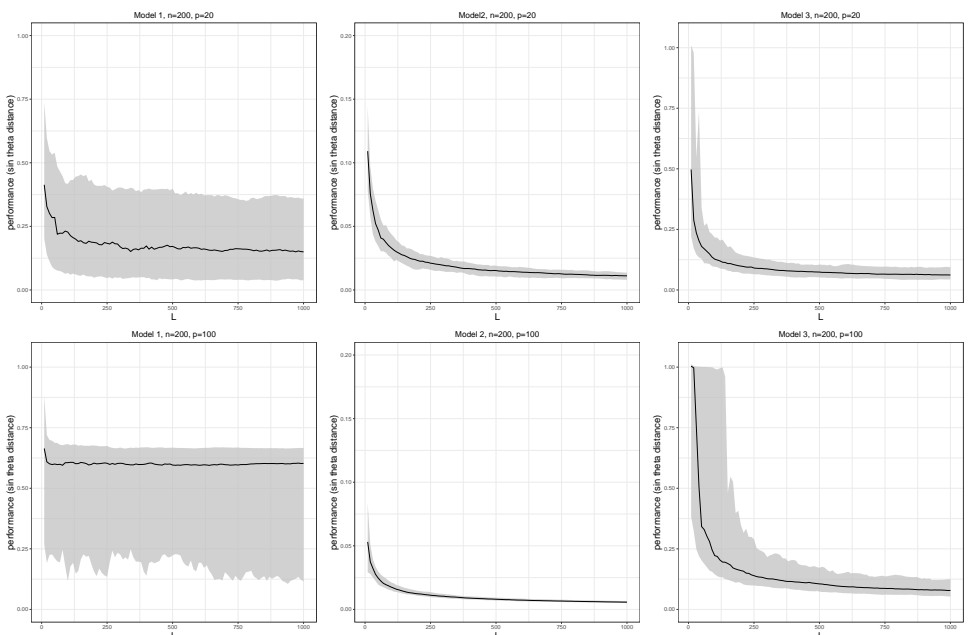

Figure 9: The median (solid line) of the sin-theta distance (see equation 2) between $U_{1:d_0}$ from the output of Algorithm 1 and the true direction $A_0$ over 100 repeats as $L$ varies. We present the results for Model 1a (left), Model 2 (middle) and Model 3 (right), for $p = 20$ (top row) and $p = 100$ (bottom row). The grey shaded region shows area between the 5%-95% quantiles for each value of $L$.

unknown) $d_0$–one or more relevant projection directions will always be missed, as the algorithm will consistently choose only a more dominant dimension reduction direction. Nevertheless, we aim to keep $d$ relatively small for computational considerations and the fact that the base regression methods may suffer from the curse of dimensionality.

To explore these aspects in practice, we repeat the experiments presented in Figure 9, but now fix $L$ at the default recommendation of 200, and vary $d \in \{1, 2, 5, 10, 20\}$ and $M \in [1000]$. The results are shown in Figure 10. When $p = 20$, taking $d = 1$ is not effective across all three models considered, and there is a clear advantage to taking $d > d_0$. To elucidate this point, consider an example where $d_0 = 1$. Setting $d = 1$ requires one of the $M$ projections in each group to be relatively highly correlated with the true projection direction, which is rare unless $M$ is very large. With a larger $d$, it is more likely that we find a projection with some signal, which then yields good performance in the final aggregation step. However, if $d$ is too large, the base method may begin to overfit–as seen, for example, Model 1a when $p = 20$ and $d = 10$.

Regarding $M$, when $p = 20$ and $d = 5$, the performance of our proposal improves rapidly as $M$ increases, and then levels off for $M$ between 100 and 200. When $p = 100$ in Model 1a, $M$ needs to be very large in order to capture all of the signal in the first singular vector of $U$; in fact, nearly all of the signal lies in the first two singular vectors even for moderately small values of $M$. For Model 3 with $p = 100$, the algorithm recovers one of the $d_0 = 2$ directions for small $M$ (for every $d$), but larger $d$ and $M$ are needed to capture the second direction. This explains the two plateaux seen in the bottom-right plot. Overall, we recommend $d = \min\{\lceil p^{1/2} \rceil, 10\}$ and $M = 10p$ as sensible default choices.

### A.3    CHOICE OF $n_1$

The purpose of this subsection is to present a sensitivity analysis regarding the sample-split ratio in Algorithm 1, since $n_1/n$ trades off selection stability against validation reliability. We use Models 1a, 2 and 3 from Appendix A.1, set $p \in \{20, 100\}$ and $n = 200$, and apply Algorithm 1 with $d = \min\{\lceil p^{1/2} \rceil, 10\}$, $L = 200$, $M = 10p$, and $Q = \frac{1}{2}Q_N^{\otimes d} + \frac{1}{2}Q_C^{\otimes d}$. We vary $n_1/n \in \{1/2, 2/3, 3/4, 4/5\}$. The results are presented in Figure 11, where we see that the perfor-

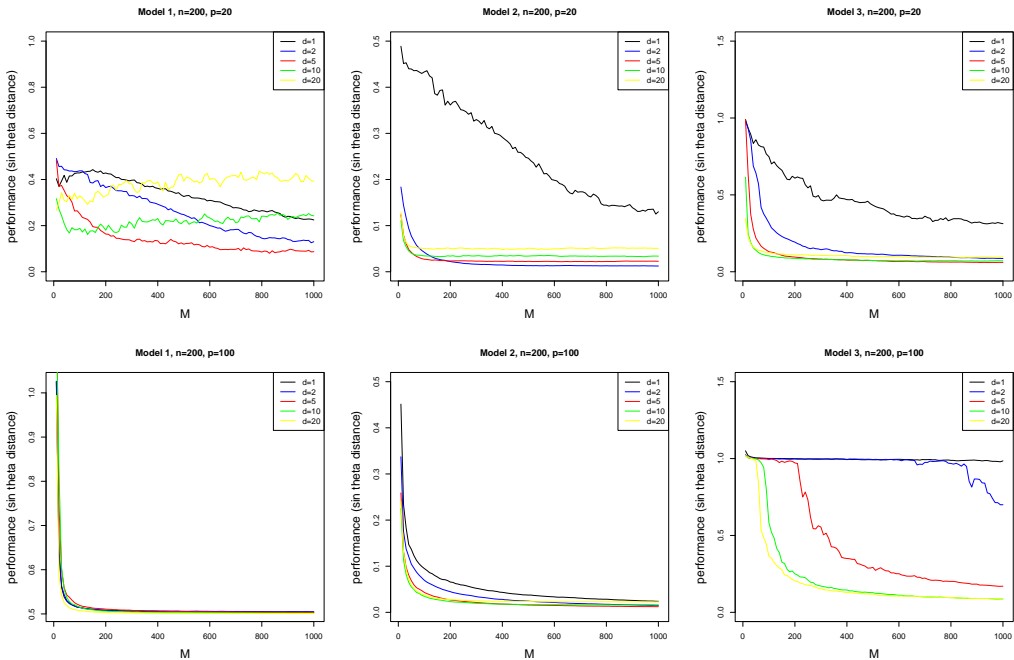

Figure 10: The median of the sin-theta distance (see equation 2) between $U_{1:d_0}$ from the output of Algorithm 1 and the true direction $A_0$ over 100 repeats as $M$ varies from 10 to 1000, with $d = 1$ (black), $d = 2$ (blue), $d = 5$ (red), $d = 10$ (green) and $d = 20$ (yellow). We present the results for Model 1a (left), Model 2 (middle) and Model 3 (right), for $p = 20$ (top row) and $p = 100$ (bottom row).

mance is very stable as $n_1$ varies over this range. Nevertheless, we use $n_1/n = 2/3$ throughout the experiments in this paper.

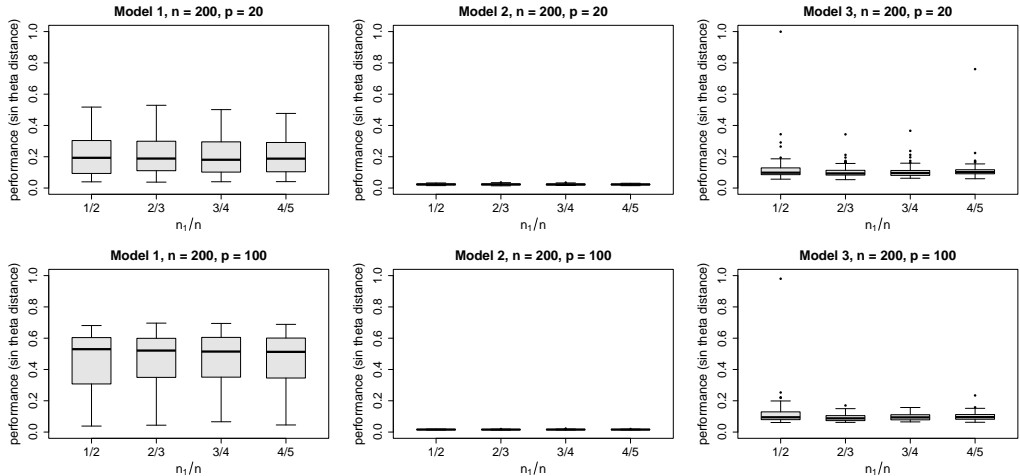

Figure 11: Boxplots of the sin-theta distance between $U_{1:d_0}$ from the output of Algorithm 1 and the true direction $A_0$ over 100 repeats as $n_1/n$ varies in $\{1/2, 2/3, 3/4, 4/5\}$. We present the results for Model 1a (left), Model 2 (middle) and Model 3 (right), for $p = 20$ (top row) and $p = 100$ (bottom row).

### A.4 COMPUTATIONAL CONSIDERATIONS

The dominant cost of Algorithm 1 comes from applying the base regression method $L \times M$ times; by contrast, projecting the data and computing the final SVD are relatively cheap if the ambient dimension $p$ is modest (tens to low hundreds). Notably, these $L \times M$ fits are highly parallelizable,

so multi-core execution substantially reduces wall-clock time. In practice, we parallelize only the outer loop over the $L$ groups (one group per worker), since these groups are independent of each other; within each worker, the $M$ candidates are evaluated sequentially, and the selected projections are aggregated afterwards.

In Appendix A.1 we have shown that global linear least squares (LLS) and global quadratic least squares (QLS) are effective in certain applications. We report below the illustrative runtimes for the setting $n = 200$, $p = 50$ (Model 1a). Specifically, we measure the wall-clock time per experiment (using 10 CPU cores) on a 3.20 GHz Intel i9-14900KF computer for MARS (default), QLS, and LLS. Table 1 summarizes the results; these numbers are illustrative and complement the full timings in Appendix C.3 (all measured on the same machine).

| Base regressor | Time (s) | Notes |
|---|---|---|
| MARS (default) | 56.133 | slow; flexible nonparametric default, robust to misspecification |
| QLS | 20.780 | faster; less strong but still often effective |
| LLS | 12.794 | fastest; effective only when the signal is close to linear |

Table 1: Runtime (seconds) per run for RPE employed with three base regressors on 10 cores; Model 1a with $n = 200$, $p = 50$; $L{=}200$, $M{=}10p$, $d{=}\lceil\sqrt{p}\rceil$, $Q = \frac{1}{2}Q_{\mathrm{N}}^{\otimes d} + \frac{1}{2}Q_{\mathrm{C}}^{\otimes d}$

An appealing direction is to first screen candidate projections within each group using a cheap base regressor (e.g., QLS), then apply MARS only to the top candidates (for instance, the top 5%) before selecting the best in the group. This two-stage strategy has the potential to preserve good performance at a much lower cost; a thorough evaluation is left to future work. Nevertheless, we report runtimes of our method with default settings in Appendix C.3.

### A.5 DEMONSTRATION OF DOUBLE RANDOM PROJECTION ENSEMBLE DIMENSION REDUCTION

We demonstrate the procedure of Algorithm 3 using Model 1a and a new model as follows:

- Model 4: Let $X = (X_1, \ldots, X_p)^\top \sim N_p(0, I)$ and

$$Y = \exp\Big(\frac{X_1 - X_2 + X_3}{3}\Big) + \epsilon$$

  with $\epsilon \sim N(0, 1/5)$ independent of $X$. Here $A_0 = \frac{1}{\sqrt{3}}(1, -1, 1, \mathbf{0}_{p-3})^\top$ is one-dimensional.

In both Models 1a and 4, the true projection $A_0$ is one-dimensional and therefore we seek to find the best one-dimensional projection. Figure 12 compares the leading singular vector obtained from Algorithm 1 with the output of Algorithm 3, using the recommended default inputs and $\check{d}_0 = 1$. Specifically, we apply Algorithm 1 with $L = 200$, $M = 10p$, $d = \min\{\lceil p^{1/2}\rceil, 10\}$, $n_1 = \lceil 2n/3\rceil$, the projection distribution $Q = \frac{1}{2}Q_{\mathrm{N}}^{\otimes d} + \frac{1}{2}Q_{\mathrm{C}}^{\otimes d}$, and $\hat{g} = \hat{g}_{\mathrm{MARS}}$. In both models, a second application yields substantially improved performance. For Model 1a, the first application (Algorithm 1) typically identifies the first two coordinates as important, but splits the signal across the first two singular vectors. The second application (Algorithm 3) then effectively combines these separate signals into a better estimate of the one-dimensional $A_0$. A similar and even more pronounced improvement is observed in Model 4.

When the initial application of Algorithms 1 and 2 correctly identifies the true dimension $d_0$, applying Algorithm 3 provides no further benefit. However, as long as $\check{d}_0 \geq d_0$, there is typically no disadvantage either–see, for example, the results for Model 3 presented in Figure 6 of the main text.

We now provide an extension of Theorem 1, which provides guarantees for the output of Algorithm 3. To set the scene for this new result, suppose we input $\check{d}_0 = d_0$ in Algorithm 3, and that Algorithm 2 suggests we should set $\hat{d}_0 > d_0$, then lines 5-7 of Algorithm 3 apply. Let $A_0^* \in \mathcal{A}_{\hat{d}_0 \times d_0}$ be the $Q$ matrix in a QR decomposition of $\hat{A}_0^\top A_0$. Intuitively, $A_0^*$ captures exactly the part of

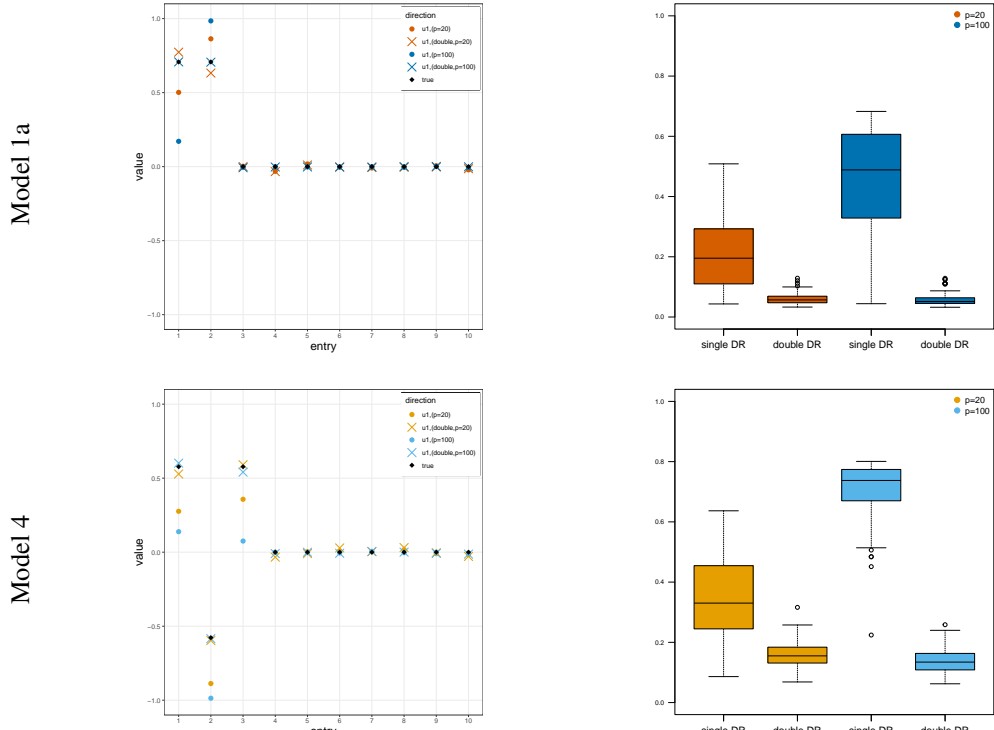

Figure 12: Demonstration of the double RPE dimension reduction approach in Algorithm 3. We compare a single application of Algorithm 1 with $\hat{d}_0 = 1$ to the double application in Algorithm 3 with $\check{d}_0 = 1$. Left: the first 10 entries of the estimated $A_0$ for a single run; right: the sin-theta distance over 100 repeats. We report results for $p \in \{20, 100\}$, with Model 1a (top) and Model 4 (bottom).

the true CMS $\mathcal{S}(A_0)$ that is preserved inside $\mathcal{S}(\hat{A}_0)$ after the first-stage projection. As in Theorem 1, we treat the $\{(z_1, y_1), \ldots, (z_n, y_n)\}$ as fixed pairs in $\mathbb{R}^{\hat{d}_0} \times \mathbb{R}$. From Algorithm 3, we obtain $\check{U} = \check{U}_{1:\hat{d}_0} = (\check{U}_1, \ldots, \check{U}_{\hat{d}_0})$ and $\check{D}$. Algorithm 3 then outputs $\check{A}_0 = \hat{A}_0(\check{U}_1, \ldots, \check{U}_{d_0}) = \hat{A}_0 \check{U}_{1:d_0}$. Let $\Pi_*^{\infty} := \mathbb{E}(\check{U} \check{D} \check{U}^T \mid \hat{A}_0) \in \mathbb{S}_{\hat{d}_0 \times \hat{d}_0}$.

Theorem 2 bounds the expected error of the output of Algorithm 3 conditionally on the initial output from the first application of Algorithm 1. The bound includes the *false negative* error arising from the first application, defined as

$$d_{\mathrm{FN}}(\hat{A}_0, A_0) := \left\| (I - \hat{A}_0 \hat{A}_0^\top) A_0 \right\|_F.$$

$d_{\mathrm{FN}}$ measures *false negatives* and quantifies the amount of the space spanned by $A_0$ that is *missed* by the projection $\hat{A}_0$. Indeed, we have $d_{\mathrm{FN}}(\hat{A}_0, A_0) = 0$ when $\mathcal{S}(A_0) \subseteq \mathcal{S}(\hat{A}_0)$. This should be interpreted as quantifying the amount of the true signal missed by the first application of Algorithm 1. The other terms in the bound in Theorem 2 should be interpreted similarly to those in Theorem 1, note, however, that since the dimension has been reduced from $p$ to $\hat{d}_0$ in the first application, the final term in the bound no longer depends on the ambient dimension $p$.

**Theorem 2.** *We have that*

$$\mathbb{E}\big[d_{\mathrm{F}}\big(\mathcal{S}(\check{A}_0), \mathcal{S}(A_0)\big) \mid \hat{A}_0\big]$$

$$\leq d_{\mathrm{FN}}(\hat{A}_0, A_0) + 2d_0^{1/2} \|\Pi_*^{\infty} - A_0^*(A_0^*)^\top\|_{\mathrm{op}} + \frac{2(2\pi)^{1/2} d_0^{1/2} d \cdot \hat{d}_0}{L^{1/2}}.$$

The proof of Theorem 2 is given in Appendix B.

# B    PROOFS OF OUR TECHNICAL RESULTS

The aim of this section is to prove Theorem 1 and Theorem 2.

*Proof of Theorem 1.*  Recall that the data $(x_1, y_1), \ldots, (x_n, y_n)$ are treated as $n$ fixed (i.e. non random) pairs in $\mathbb{R}^p \times \mathbb{R}$. From Algorithm 1 we have that $\hat{\Pi} = \frac{1}{L} \sum_{\ell=1}^L \mathbf{P}_{\ell,*} \mathbf{P}_{\ell,*}^\top$, and $\hat{A}_0^L \in \mathbb{R}^{p \times d_0}$ denotes the matrix whose columns are the $d_0$ leading eigenvectors of $\hat{\Pi}$, ordered according to the corresponding eigenvalues. Recall also that $\Pi^\infty = \mathbb{E}(\mathbf{P}_{\ell,*} \mathbf{P}_{\ell,*}^\top)$. Let $\Pi := A_0 A_0^\top \in \mathbb{R}^{p \times p}$, and for $j \in [d_0]$ write $a_j$ for the $j$th column of $A_0$, so that $A_0 = (a_1, \cdots, a_{d_0})$. Since $A_0^\top A_0 = I_{d_0 \times d_0}$, the $d_0$ largest eigenvalues of $\Pi$ are 1 and the remainder are 0. Indeed, $\mathrm{rank}\,(\Pi) = \mathrm{rank}\,(A_0) = d_0$, so $\Pi$ has $d_0$ nonzero eigenvalues. Moreover, $\Pi a_j = A_0 A_0^\top a_j = A_0 (A_0^\top a_j) = A_0 e_j = a_j$ for $j \in [d_0]$. Then, since $\Pi$ and $\hat{\Pi}$ are symmetric, by the Davis–Kahan Theorem (Yu et al., 2015, Theorem 2) we have

$$d_{\mathrm{F}}\big(\mathcal{S}(\hat{A}_0^L), \mathcal{S}(A_0)\big) \equiv \big\|\sin\Theta(\hat{A}_0^L, A_0)\big\|_F \leq 2 d_0^{1/2} \big\|\hat{\Pi} - \Pi\big\|_{\mathrm{op}} \tag{4}$$
$$\leq 2 d_0^{1/2}\Big(\|\hat{\Pi} - \Pi^\infty\|_{\mathrm{op}} + \|\Pi^\infty - \Pi\|_{\mathrm{op}}\Big).$$

The second term in equation 4 does not depend on $L$ or on the randomness in the projections. The remainder of the proof therefore involves controlling the expectation of the first term on the right hand side of equation 4. To that end, we first show that

$$\mathbb{P}\Big(\big\|\hat{\Pi} - \Pi^\infty\big\|_{\mathrm{op}} \geq t\Big) \leq p \cdot e^{-\frac{t^2 L}{8 d^2}} \tag{5}$$

for $t > 0$. To see this, for $\ell \in [L]$ let $\Pi_\ell := \mathbf{P}_{\ell,*} \mathbf{P}_{\ell,*}^\top$ and write $J_\ell := \frac{1}{L}(\Pi_\ell - \Pi^\infty)$. Then

$$\hat{\Pi} - \Pi^\infty = \frac{1}{L} \sum_{\ell=1}^L \mathbf{P}_{\ell,*} \mathbf{P}_{\ell,*}^\top - \Pi^\infty = \sum_{\ell=1}^L J_\ell.$$

Furthermore, $\mathbb{E}(J_\ell) = 0$, and $J_\ell$ (and thus also $J_\ell^2$) is symmetric. We now show that $J_\ell^2 \preccurlyeq \frac{d^2}{L^2} I_{p \times p}$, in other words $\frac{d^2}{L^2} I_{p \times p} - J_\ell^2$ is positive semidefinite almost surely, or equivalently that

$$\sup_{\{v : \|v\| = 1\}} v^\top J_\ell^2 v \leq \frac{d^2}{L^2}. \tag{6}$$

First, write $\mathbf{P}_{\ell,*} = (\mathbf{P}_{\ell,*,1}, \ldots, \mathbf{P}_{\ell,*,d})$. Since $\mathrm{diag}(\mathbf{P}_{\ell,*}^\top \mathbf{P}_{\ell,*}) = (1, \ldots, 1)^\top \in \mathbb{R}^d$, we have $\|\mathbf{P}_{\ell,*,j}\|^2 = 1$ for $j \in [d]$. Then, for $v \in \mathbb{R}^p$ with $\|v\|_2 = 1$, by Cauchy-Schwarz we have

$$v^\top \Pi_l v = \|\mathbf{P}_{\ell,*}^\top v\|^2 = \sum_{j=1}^d (\mathbf{P}_{\ell,*,j}^\top v)^2 \leq \sum_{j=1}^d \|\mathbf{P}_{\ell,*,j}\|^2 \|v\|^2 = d.$$

Therefore, $v^\top \Pi_l v \in [0, d]$. Moreover, by the linearity of expectation, we also have

$$v^\top \Pi^\infty v = v^\top \mathbb{E}(\Pi_\ell) v = \mathbb{E}(v^\top \Pi_\ell v) \in [0, d]$$

whenever $\|v\|_2 = 1$. It follows that

$$v^\top (\Pi_\ell - \Pi^\infty) v = v^\top \Pi_\ell v - v^\top \Pi^\infty v \in [-d, d], \qquad \text{for all } \|v\|_2 = 1.$$

Hence we can write $\Pi_\ell - \Pi^\infty = U_\ell D_\ell U_\ell^\top$, where $U_\ell \in \mathbb{R}^{p \times p}$ is orthonormal and $D_\ell$ is a diagonal $p \times p$ matrix with all diagonal entries between $-d$ and $d$ (Horn & Johnson, 1985, Corollary 2.5.11). We deduce that, for all $v \in \mathbb{R}^p$ with $\|v\|_2 = 1$, we have

$$v^\top J_\ell^2 v = \frac{1}{L^2} v^\top (\Pi_\ell - \Pi^\infty)^2 v = \frac{1}{L^2} v^\top U_\ell D_\ell^2 U_\ell^\top v \leq \frac{d^2}{L^2} \|U_\ell^\top v\|^2 = \frac{d^2}{L^2},$$

which establishes equation 6.

Finally, because each of the projections $\mathbf{P}_{\ell,*}$ is chosen from a disjoint group of independently generated random projections, we have that $\mathbf{P}_{\ell,*}$ for $\ell \in [L]$ are independent, and therefore $J_\ell$ for $\ell \in [L]$

are also independent. It then follows from the Matrix Hoeffding inequality (see, for example Tropp (2012, Theorem 1.3)), that

$$\mathbb{P}\Big(\big\|\hat{\Pi} - \Pi^\infty\big\|_{\mathrm{op}} \geq t\Big) \leq p \cdot \exp\Big(-\frac{t^2}{8\|\sum_{\ell=1}^{L} \frac{d^2 I_{p \times p}}{L^2}\|_{\mathrm{op}}}\Big) = p \cdot e^{-\frac{t^2 L}{8d^2}},$$

which establishes the bound in equation 5. To complete the proof we bound the expectation as follows:

$$\mathbb{E}\Big(\big\|\hat{\Pi} - \Pi^\infty\big\|_{\mathrm{op}}\Big) = \int_0^\infty \mathbb{P}\Big(\big\|\hat{\Pi} - \Pi^\infty\big\|_{\mathrm{op}} \geq t\Big) dt$$
$$\leq \int_0^\infty p \cdot e^{-\frac{t^2 L}{8d^2}} dt = \frac{d \cdot p\sqrt{2\pi}}{\sqrt{L}},$$

as required. □

*Proof of Theorem 2.* First, by triangle inequality we have

$$d_{\mathrm{F}}\big(\mathcal{S}(\check{A}_0), \mathcal{S}(A_0)\big) \leq d_{\mathrm{F}}\big(\mathcal{S}(\hat{A}_0 \check{U}_{1:d_0}), \mathcal{S}(\hat{A}_0 A_0^*)\big) + d_{\mathrm{F}}\big(\mathcal{S}(\hat{A}_0 A_0^*), \mathcal{S}(A_0)\big). \tag{7}$$

Now for any $B \in \mathbb{R}^{\hat{d}_0 \times \hat{d}_0}$

$$\big\|\hat{A}_0 B \hat{A}_0^\top\big\|_F^2 = \mathrm{tr}\big\{\hat{A}_0 B \hat{A}_0^\top \hat{A}_0 B^\top \hat{A}_0^\top\big\} = \mathrm{tr}\big\{B(\hat{A}_0^\top \hat{A}_0) B^\top (\hat{A}_0^\top \hat{A}_0)\big\}$$
$$= \mathrm{tr}\big\{BB^\top\big\} = \big\|B\big\|_F^2.$$

It follows that

$$d_{\mathrm{F}}\big(\mathcal{S}(\hat{A}_0 \check{U}_{1:d_0}), \mathcal{S}(\hat{A}_0 A_0^*)\big) = \frac{1}{\sqrt{2}}\big\|\hat{A}_0 \check{U}_{1:d_0} \check{U}_{1:d_0}^\top \hat{A}_0^\top - \hat{A}_0 A_0^* (A_0^*)^\top \hat{A}_0^\top\big\|_F$$
$$= \frac{1}{\sqrt{2}}\big\|\hat{A}_0 [\check{U}_{1:d_0} \check{U}_{1:d_0}^\top - A_0^* (A_0^*)^\top] \hat{A}_0^\top\big\|_F$$
$$= \frac{1}{\sqrt{2}}\big\|\check{U}_{1:d_0} \check{U}_{1:d_0}^\top - A_0^* (A_0^*)^\top\big\|_F$$
$$= d_{\mathrm{F}}\big(\mathcal{S}(\check{U}_{1:d_0}), \mathcal{S}(A_0^*)\big). \tag{8}$$

Further, by an application of Theorem 1 to the data projected by $\hat{A}_0$ and working conditionally on $\hat{A}_0$, we have

$$\mathbb{E}\big\{d_{\mathrm{F}}\big(\mathcal{S}(\check{U}_{1:d_0}), \mathcal{S}(A_0^*)\big) \mid \hat{A}_0\big\} \leq 2d_0^{1/2}\|\Pi_*^\infty - A_0^*(A_0^*)^\top\|_{\mathrm{op}} + \frac{2(2\pi)^{1/2} d_0^{1/2} d \cdot \hat{d}_0}{L^{1/2}}. \tag{9}$$

Now consider the second term on the right hand side of equation 7. For simplicity of exposition, let $P := \hat{A}_0 A_0^* (A_0^*)^\top \hat{A}_0^\top$, $Q := A_0 A_0^\top$. Both $P$ and $Q$ are symmetric, $P^2 = P$, $Q^2 = Q$, and $\mathrm{tr}(PQ) = \mathrm{tr}(QP)$. Further

$$\mathrm{tr}(Q) = \mathrm{tr}(A_0 A_0^\top) = \mathrm{tr}(A_0^\top A_0) = \mathrm{tr}(I_{d_0}) = d_0,$$

and similarly

$$\mathrm{tr}(P) = \mathrm{tr}\big\{\hat{A}_0 A_0^* (A_0^*)^\top \hat{A}_0^\top\big\} = \mathrm{tr}\big\{(A_0^*)^\top (\hat{A}_0^\top \hat{A}_0) A_0^*\big\}$$
$$= \mathrm{tr}\big\{(A_0^*)^\top A_0^*\big\} = \mathrm{tr}(I_{d_0}) = d_0.$$

It follows that

$$d_{\mathrm{F}}^2\big(\mathcal{S}(\hat{A}_0 A_0^*), \mathcal{S}(A_0)\big) = \frac{1}{2}\big\|P - Q\big\|_F^2$$
$$= \frac{1}{2}\mathrm{tr}\big\{(P - Q)^\top (P - Q)\big\}$$
$$= \frac{1}{2}\mathrm{tr}\big\{(P - Q)^2\big\} \tag{10}$$
$$= \frac{1}{2}\mathrm{tr}(P^2 - PQ - QP + Q^2)$$
$$= \frac{1}{2}\mathrm{tr}(P + Q - 2PQ) = d_0 - \mathrm{tr}(PQ).$$

Recall that $A_0^* \in \mathcal{A}_{\hat{d}_0 \times d_0}$ is given by the QR decomposition of $M := \hat{A}_0^\top A_0$, i.e.

$$M = \hat{A}_0^\top A_0 = A_0^* R, \tag{11}$$

where $R \in \mathbb{R}^{d_0 \times d_0}$ is some upper triangular matrix. Then solving equation 11 gives

$$R = (A_0^*)^\top \hat{A}_0^\top A_0.$$

Therefore

$$\mathrm{tr}(PQ) = \mathrm{tr}\big\{\hat{A}_0 A_0^* (A_0^*)^\top \hat{A}_0^\top A_0 A_0^\top\big\} = \mathrm{tr}\big\{(A_0^*)^\top \hat{A}_0^\top A_0 A_0^\top \hat{A}_0 A_0^*\big\} = \mathrm{tr}(RR^\top). \tag{12}$$

On the other hand, recall from the definition of false negative measure $d_{\mathrm{FN}}(\hat{A}_0, A_0)$ we have

$$
\begin{aligned}
d_{\mathrm{FN}}^2(\hat{A}_0, A_0) &= \big\|(I - \hat{A}_0 \hat{A}_0^\top) A_0\big\|_F^2 \\
&= \mathrm{tr}\big\{A_0^\top (I - \hat{A}_0 \hat{A}_0^\top)^2 A_0\big\} \\
&= \mathrm{tr}\big\{A_0^\top (I - \hat{A}_0 \hat{A}_0^\top) A_0\big\} \\
&= \mathrm{tr}\big\{A_0 A_0^\top (I - \hat{A}_0 \hat{A}_0^\top)\big\} \\
&= \mathrm{tr}\big\{A_0 A_0^\top\big\} - \mathrm{tr}\big\{A_0 A_0^\top \hat{A}_0 \hat{A}_0^\top\big\} \\
&= d_0 - \mathrm{tr}\big\{\hat{A}_0^\top A_0 A_0^\top \hat{A}_0\big\} \\
&= d_0 - \mathrm{tr}(MM^\top).
\end{aligned} \tag{13}
$$

Finally, we have

$$\mathrm{tr}(MM^\top) = \mathrm{tr}\big\{A_0^* RR^\top (A_0^*)^\top\big\} = \mathrm{tr}\big\{RR^\top (A_0^*)^\top A_0^*\big\} = \mathrm{tr}(RR^\top). \tag{14}$$

Combining equation 10, equation 12, equation 13, and equation 14, we have

$$d_{\mathrm{F}}\big(\mathcal{S}(\hat{A}_0 A_0^*), \mathcal{S}(A_0)\big) = d_{\mathrm{FN}}(\hat{A}_0, A_0) \tag{15}$$

Now taking conditional expectation with respect to $\hat{A}_0$ on both sides of equation 7, in combination with equation 9 and equation 15 completes the proof. □

## C   FULL SIMULATION RESULTS

In this section, we present the full results of our simulation study, which follows the design of experiments described in Section 4 of the main text. The models considered are Model 1 (with $q = 2$, introduced in Section 3.1), Model 2 and 3 (introduced in Appendix A.1), Model 4 (introduced in Appendix A.5), and five additional models defined as follows: let $X := (X_1, \ldots, X_p)^\top \sim N_p(0, I)$ and $\epsilon \sim N(0, 1/4)$, with $X$ and $\epsilon$ independent:

- Model 5: $Y = \frac{X_1 + X_2 + 5e^{-2(X_1 + X_2 + X_3)^2}}{2} + \epsilon$,
- Model 6: $Y = 5X_1 X_2 X_3 + \epsilon$
- Model 7: $Y = 4(X_1 - X_2 + X_3)\sin\big(\frac{\pi}{2}(X_1 + X_2)\big) + \epsilon$
- Model 8: $Y = X_1(X_1 + X_2 + 1) + \epsilon$
- Model 9: $Y = 10\cos(6X_1) + e^{X_2+1} + \epsilon$

Models 5, 6 and 7 were used in Liu et al. (2023), and Model 8 was used in Li (1991); Xia et al. (2002); Liu et al. (2023).

For each model, we consider the settings with $p \in \{20, 50, 100, 200, 500\}$ and $n \in \{50, 200, 500\}$. The full description of the algorithms considered, along with how tuning parameters were chosen, is given in Section 4.

## C.1 Dimension $d_0$ known

First, we report the complete results for the setting where the dimension $d_0$ of the central mean subspace $\mathcal{S}(A_0)$ is known, providing the full picture of this scenario described in the main text. To save space, rather than presenting boxplots here, we focus on the mean and standard error of the sin-theta distance (when $d_0$ is known) over 100 repeats of the experiments. The results are given in Table 2 (Models 1a-3), Table 3 (Models 4-6), and Table 4 (Models 7-9).

The broad message from the results is that our random projection based algorithms are competitive across all nine models, for different values of $n$ and $p$. The relative performance is broadly similar to the $p \in \{50, 500\}$, $n = 200$ cases presented in the main text. More specifically, we note the following. First, in the cases where $p \geq n$, SIR, pHd and DR methods are not applicable. Second, in relatively large $p$ settings (for example, $n < 10p$), our method shows substantial improvement over existing methods. Overall, our approach enjoys the best performance for Models 2, 3, 4, 8, and 9. For the remaining models, our random projection based method and drMARS perform similarly and typically outperform the other competitors.

| Setting | | | Competing Methods | | | | | | Our Methods | |
|---|---|---|---|---|---|---|---|---|---|---|
| Model | $p$ | $n$ | SIR | pHd | MAVE | DR | gKDR | drMARS | RPE | RPE2 |
| | 20 | 50 | $0.97_{0.05}$ | $0.81_{0.10}$ | $0.87_{0.14}$ | $0.97_{0.05}$ | $0.92_{0.09}$ | $0.13_{0.18}$ | $0.32_{0.14}$ | $\mathbf{0.12_{0.04}}$ |
| | | 200 | $0.97_{0.05}$ | $0.40_{0.08}$ | $0.24_{0.04}$ | $0.33_{0.06}$ | $0.29_{0.06}$ | $\mathbf{0.03_{0.01}}$ | $0.20_{0.11}$ | $0.06_{0.02}$ |
| | | 500 | $0.96_{0.07}$ | $0.24_{0.05}$ | $0.10_{0.02}$ | $0.19_{0.03}$ | $0.14_{0.03}$ | $\mathbf{0.01_{0.01}}$ | $0.13_{0.08}$ | $0.05_{0.02}$ |
| | 50 | 50 | / | / | $0.93_{0.09}$ | / | $0.97_{0.03}$ | $0.34_{0.28}$ | $0.46_{0.17}$ | $\mathbf{0.14_{0.08}}$ |
| | | 200 | $0.99_{0.02}$ | $0.74_{0.09}$ | $0.93_{0.08}$ | $0.65_{0.12}$ | $0.98_{0.04}$ | $\mathbf{0.04_{0.01}}$ | $0.32_{0.17}$ | $0.06_{0.02}$ |
| 1a | | 500 | $0.98_{0.04}$ | $0.43_{0.06}$ | $0.42_{0.12}$ | $0.34_{0.04}$ | $0.57_{0.20}$ | $\mathbf{0.02_{0.02}}$ | $0.28_{0.15}$ | $0.05_{0.02}$ |
| ($d_0 = 1$) | 100 | 50 | / | / | $0.95_{0.07}$ | / | $0.98_{0.02}$ | $0.33_{0.28}$ | $0.57_{0.17}$ | $\mathbf{0.29_{0.29}}$ |
| | | 200 | $1.00_{0.00}$ | $0.95_{0.03}$ | $0.97_{0.04}$ | $0.99_{0.01}$ | $0.99_{0.02}$ | $\mathbf{0.05_{0.02}}$ | $0.46_{0.17}$ | $0.06_{0.02}$ |
| | | 500 | $0.99_{0.01}$ | $0.72_{0.07}$ | $0.94_{0.08}$ | $0.54_{0.04}$ | $0.99_{0.02}$ | $\mathbf{0.01_{0.01}}$ | $0.38_{0.16}$ | $0.05_{0.02}$ |
| | 200 | 50 | / | / | $0.97_{0.06}$ | / | $0.98_{0.02}$ | $0.31_{0.27}$ | $0.64_{0.19}$ | $\mathbf{0.35_{0.30}}$ |
| | | 200 | / | / | $0.97_{0.04}$ | / | $0.99_{0.01}$ | $0.23_{0.21}$ | $0.55_{0.16}$ | $\mathbf{0.06_{0.02}}$ |
| | 500 | 50 | / | / | $0.98_{0.05}$ | / | $0.99_{0.01}$ | $\mathbf{0.34_{0.31}}$ | $0.83_{0.18}$ | $0.78_{0.26}$ |
| | | 200 | / | / | $0.98_{0.03}$ | / | $0.99_{0.01}$ | $0.22_{0.19}$ | $0.61_{0.15}$ | $\mathbf{0.08_{0.03}}$ |
| | 20 | 50 | $0.96_{0.06}$ | $0.98_{0.03}$ | $0.90_{0.09}$ | $0.97_{0.03}$ | $0.94_{0.06}$ | $0.45_{0.35}$ | $\mathbf{0.20_{0.26}}$ | $0.26_{0.33}$ |
| | | 200 | $0.88_{0.11}$ | $0.98_{0.02}$ | $0.78_{0.11}$ | $0.96_{0.05}$ | $0.88_{0.11}$ | $0.09_{0.10}$ | $\mathbf{0.02_{0.00}}$ | $0.02_{0.00}$ |
| | | 500 | $0.60_{0.10}$ | $0.99_{0.02}$ | $0.61_{0.10}$ | $0.95_{0.06}$ | $0.81_{0.12}$ | $0.05_{0.02}$ | $\mathbf{0.02_{0.00}}$ | $0.02_{0.00}$ |
| | 50 | 50 | / | / | $0.91_{0.07}$ | / | $0.98_{0.03}$ | $0.79_{0.29}$ | $\mathbf{0.21_{0.29}}$ | $0.43_{0.41}$ |
| | | 200 | $0.98_{0.02}$ | $0.99_{0.01}$ | $0.87_{0.05}$ | $0.98_{0.03}$ | $0.92_{0.06}$ | $0.09_{0.04}$ | $\mathbf{0.02_{0.00}}$ | $0.02_{0.00}$ |
| 2 | | 500 | $0.92_{0.07}$ | $0.99_{0.01}$ | $0.76_{0.06}$ | $0.98_{0.03}$ | $0.75_{0.07}$ | $0.06_{0.02}$ | $\mathbf{0.02_{0.00}}$ | $0.02_{0.00}$ |
| ($d_0 = 1$) | 100 | 50 | / | / | $0.95_{0.05}$ | / | $0.99_{0.01}$ | $0.77_{0.31}$ | $\mathbf{0.27_{0.36}}$ | $0.54_{0.43}$ |
| | | 200 | $0.99_{0.02}$ | $1.00_{0.01}$ | $0.91_{0.05}$ | $0.98_{0.02}$ | $0.96_{0.04}$ | $0.10_{0.04}$ | $\mathbf{0.02_{0.00}}$ | $0.02_{0.01}$ |
| | | 500 | $0.99_{0.01}$ | $1.00_{0.00}$ | $0.84_{0.05}$ | $0.99_{0.02}$ | $0.84_{0.05}$ | $0.06_{0.02}$ | $\mathbf{0.02_{0.00}}$ | $0.02_{0.01}$ |
| | 200 | 50 | / | / | $0.96_{0.05}$ | / | $1.00_{0.00}$ | $0.75_{0.31}$ | $0.26_{0.37}$ | $0.30_{0.38}$ |
| | | 200 | / | / | $0.92_{0.04}$ | / | $0.98_{0.02}$ | $0.27_{0.19}$ | $\mathbf{0.01_{0.00}}$ | $0.01_{0.00}$ |
| | 500 | 50 | / | / | $0.97_{0.04}$ | / | $1.00_{0.00}$ | $0.85_{0.24}$ | $0.33_{0.42}$ | $0.34_{0.42}$ |
| | | 200 | / | / | $0.95_{0.03}$ | / | $0.99_{0.01}$ | $0.33_{0.25}$ | $\mathbf{0.01_{0.00}}$ | $0.01_{0.00}$ |
| | 20 | 50 | $1.30_{0.07}$ | $1.27_{0.08}$ | $1.07_{0.10}$ | $1.29_{0.07}$ | $1.16_{0.11}$ | $0.92_{0.28}$ | $\mathbf{0.50_{0.31}}$ | $0.50_{0.30}$ |
| | | 200 | $0.85_{0.13}$ | $0.88_{0.12}$ | $0.72_{0.13}$ | $0.99_{0.11}$ | $0.89_{0.12}$ | $0.48_{0.33}$ | $\mathbf{0.10_{0.02}}$ | $0.10_{0.02}$ |
| | | 500 | $0.51_{0.08}$ | $0.55_{0.08}$ | $0.39_{0.07}$ | $0.67_{0.11}$ | $0.60_{0.11}$ | $0.27_{0.26}$ | $\mathbf{0.07_{0.01}}$ | $0.07_{0.01}$ |
| | 50 | 50 | / | / | $1.12_{0.10}$ | / | $1.32_{0.05}$ | $1.21_{0.17}$ | $0.66_{0.34}$ | $0.70_{0.33}$ |
| | | 200 | $1.27_{0.06}$ | $1.29_{0.06}$ | $1.07_{0.08}$ | $1.23_{0.07}$ | $1.03_{0.08}$ | $0.54_{0.27}$ | $\mathbf{0.10_{0.03}}$ | $0.10_{0.04}$ |
| 3 | | 500 | $0.87_{0.10}$ | $0.93_{0.09}$ | $0.80_{0.10}$ | $0.98_{0.09}$ | $0.79_{0.07}$ | $0.32_{0.25}$ | $\mathbf{0.06_{0.01}}$ | $0.06_{0.01}$ |
| ($d_0 = 2$) | 100 | 50 | / | / | $1.16_{0.11}$ | / | $1.37_{0.02}$ | $1.18_{0.19}$ | $\mathbf{0.71_{0.35}}$ | $0.80_{0.31}$ |
| | | 200 | $1.38_{0.05}$ | $1.38_{0.02}$ | $1.09_{0.08}$ | $1.29_{0.04}$ | $1.22_{0.04}$ | $0.63_{0.25}$ | $\mathbf{0.10_{0.05}}$ | $0.11_{0.05}$ |
| | | 500 | $1.22_{0.05}$ | $1.27_{0.06}$ | $1.06_{0.06}$ | $1.16_{0.06}$ | $0.94_{0.05}$ | $0.36_{0.27}$ | $\mathbf{0.06_{0.01}}$ | $0.07_{0.01}$ |
| | 200 | 50 | / | / | $1.19_{0.09}$ | / | $1.39_{0.01}$ | $1.21_{0.18}$ | $0.83_{0.32}$ | $0.83_{0.32}$ |
| | | 200 | / | / | $1.13_{0.07}$ | / | $1.32_{0.03}$ | $1.11_{0.19}$ | $\mathbf{0.13_{0.09}}$ | $0.13_{0.09}$ |
| | 500 | 50 | / | / | $1.22_{0.09}$ | / | $1.41_{0.00}$ | $1.21_{0.18}$ | $0.94_{0.23}$ | $0.93_{0.23}$ |
| | | 200 | / | / | $1.16_{0.07}$ | / | $1.38_{0.01}$ | $1.15_{0.19}$ | $\mathbf{0.25_{0.22}}$ | $0.25_{0.22}$ |

Table 2: The average sin-theta distance between $\hat{A}_0$ and $A_0$ over 100 repeats for Models 1a, 2 and 3, with $p \in \{20, 50, 100, 200, 500\}$ and $n \in \{50, 200, 500\}$. We compare two versions of our method with six competing approaches. For each setting, we also present 10 times the standard error for each method as a subscript. The "/" entries indicate that the method is not applicable.

| Setting | | | Competing Methods | | | | | | Our Methods | |
|---|---|---|---|---|---|---|---|---|---|---|
| Model | $p$ | $n$ | SIR | pHd | MAVE | DR | gKDR | drMARS | RPE | RPE2 |
| | | 50 | $0.90_{0.10}$ | $0.93_{0.08}$ | $\mathbf{0.51_{0.12}}$ | $0.85_{0.13}$ | $0.65_{0.13}$ | $\mathbf{0.50_{0.21}}$ | $0.53_{0.12}$ | $\mathbf{0.48_{0.19}}$ |
| | 20 | 200 | $0.30_{0.06}$ | $0.61_{0.11}$ | $0.29_{0.05}$ | $0.37_{0.07}$ | $0.38_{0.08}$ | $0.28_{0.22}$ | $0.34_{0.14}$ | $\mathbf{0.16_{0.04}}$ |
| | | 500 | $0.17_{0.03}$ | $0.38_{0.08}$ | $0.18_{0.03}$ | $0.23_{0.04}$ | $0.23_{0.04}$ | $0.14_{0.13}$ | $0.29_{0.12}$ | $\mathbf{0.11_{0.03}}$ |
| | | 50 | / | / | $\mathbf{0.55_{0.15}}$ | / | $0.86_{0.09}$ | $0.82_{0.20}$ | $0.66_{0.12}$ | $\mathbf{0.55_{0.19}}$ |
| | 50 | 200 | $0.67_{0.12}$ | $0.90_{0.08}$ | $0.43_{0.06}$ | $0.60_{0.07}$ | $0.49_{0.06}$ | $0.32_{0.10}$ | $0.61_{0.15}$ | $\mathbf{0.14_{0.04}}$ |
| 4 | | 500 | $0.30_{0.03}$ | $0.66_{0.09}$ | $0.30_{0.03}$ | $0.37_{0.04}$ | $0.34_{0.04}$ | $0.18_{0.09}$ | $0.53_{0.15}$ | $\mathbf{0.09_{0.03}}$ |
| | | 50 | / | / | $\mathbf{0.61_{0.16}}$ | / | $0.95_{0.04}$ | $0.82_{0.20}$ | $0.74_{0.10}$ | $\mathbf{0.62_{0.18}}$ |
| $(d_0 = 1)$ | 100 | 200 | $0.99_{0.01}$ | $0.98_{0.02}$ | $0.46_{0.06}$ | $0.77_{0.07}$ | $0.64_{0.06}$ | $0.41_{0.09}$ | $0.70_{0.10}$ | $\mathbf{0.14_{0.04}}$ |
| | | 500 | $0.52_{0.05}$ | $0.88_{0.06}$ | $0.39_{0.04}$ | $0.52_{0.04}$ | $0.41_{0.03}$ | $0.20_{0.10}$ | $0.63_{0.15}$ | $\mathbf{0.09_{0.03}}$ |
| | 200 | 50 | / | / | $\mathbf{0.69_{0.16}}$ | / | $0.98_{0.01}$ | $0.84_{0.20}$ | $0.80_{0.09}$ | $0.78_{0.17}$ |
| | | 200 | / | / | $0.48_{0.06}$ | / | $0.82_{0.06}$ | $0.85_{0.16}$ | $0.74_{0.08}$ | $\mathbf{0.17_{0.15}}$ |
| | 500 | 50 | / | / | $\mathbf{0.77_{0.16}}$ | / | $0.99_{0.00}$ | $0.85_{0.15}$ | $0.84_{0.08}$ | $0.85_{0.11}$ |
| | | 200 | / | / | $0.54_{0.06}$ | / | $0.94_{0.04}$ | $0.86_{0.15}$ | $0.79_{0.04}$ | $\mathbf{0.25_{0.24}}$ |
| | | 50 | $1.28_{0.07}$ | $1.33_{0.06}$ | $1.15_{0.09}$ | $1.25_{0.09}$ | $1.15_{0.08}$ | $1.06_{0.19}$ | $1.06_{0.09}$ | $\mathbf{1.03_{0.13}}$ |
| | 20 | 200 | $1.00_{0.06}$ | $1.07_{0.08}$ | $0.95_{0.12}$ | $1.01_{0.08}$ | $0.95_{0.09}$ | $\mathbf{0.73_{0.31}}$ | $1.00_{0.00}$ | $0.93_{0.22}$ |
| | | 500 | $0.85_{0.12}$ | $0.99_{0.05}$ | $0.64_{0.14}$ | $0.85_{0.12}$ | $0.74_{0.13}$ | $\mathbf{0.44_{0.33}}$ | $1.00_{0.00}$ | $0.66_{0.39}$ |
| | | 50 | / | / | $1.20_{0.10}$ | / | $1.30_{0.05}$ | $1.29_{0.12}$ | $\mathbf{1.12_{0.11}}$ | $1.13_{0.10}$ |
| | 50 | 200 | $1.24_{0.05}$ | $1.38_{0.03}$ | $1.15_{0.05}$ | $1.18_{0.05}$ | $1.09_{0.05}$ | $\mathbf{0.84_{0.29}}$ | $1.00_{0.00}$ | $0.97_{0.14}$ |
| 5 | | 500 | $1.04_{0.04}$ | $1.11_{0.07}$ | $1.06_{0.05}$ | $1.05_{0.04}$ | $0.95_{0.05}$ | $\mathbf{0.50_{0.30}}$ | $1.00_{0.00}$ | $0.70_{0.36}$ |
| | | 50 | / | / | $1.25_{0.08}$ | / | $1.36_{0.03}$ | $1.30_{0.12}$ | $\mathbf{1.15_{0.13}}$ | $1.19_{0.11}$ |
| $(d_0 = 2)$ | 100 | 200 | $1.40_{0.02}$ | $1.40_{0.01}$ | $1.18_{0.04}$ | $1.27_{0.03}$ | $1.20_{0.04}$ | $1.02_{0.21}$ | $1.00_{0.00}$ | $\mathbf{0.96_{0.15}}$ |
| | | 500 | $1.17_{0.03}$ | $1.39_{0.03}$ | $1.15_{0.03}$ | $1.15_{0.03}$ | $1.05_{0.04}$ | $\mathbf{0.53_{0.29}}$ | $1.00_{0.00}$ | $0.74_{0.34}$ |
| | 200 | 50 | / | / | $1.29_{0.07}$ | / | $1.39_{0.01}$ | $1.29_{0.10}$ | $\mathbf{1.21_{0.12}}$ | $\mathbf{1.21_{0.12}}$ |
| | | 200 | / | / | $1.20_{0.04}$ | / | $1.29_{0.03}$ | $1.30_{0.13}$ | $\mathbf{1.00_{0.00}}$ | $\mathbf{1.00_{0.00}}$ |
| | 500 | 50 | / | / | $1.31_{0.08}$ | / | $1.41_{0.00}$ | $1.27_{0.14}$ | $\mathbf{1.23_{0.13}}$ | $\mathbf{1.23_{0.13}}$ |
| | | 200 | / | / | $1.24_{0.04}$ | / | $1.36_{0.01}$ | $1.30_{0.11}$ | $\mathbf{1.00_{0.00}}$ | $\mathbf{1.00_{0.00}}$ |
| | | 50 | $1.59_{0.06}$ | $1.50_{0.08}$ | $1.38_{0.12}$ | $1.59_{0.07}$ | $1.52_{0.08}$ | $1.16_{0.46}$ | $\mathbf{1.14_{0.27}}$ | $1.20_{0.26}$ |
| | 20 | 200 | $1.58_{0.07}$ | $1.39_{0.09}$ | $1.26_{0.15}$ | $1.38_{0.10}$ | $1.16_{0.13}$ | $\mathbf{0.66_{0.56}}$ | $\mathbf{0.73_{0.28}}$ | $\mathbf{0.73_{0.29}}$ |
| | | 500 | $1.59_{0.06}$ | $1.36_{0.09}$ | $0.41_{0.05}$ | $0.98_{0.13}$ | $0.69_{0.10}$ | $0.49_{0.56}$ | $\mathbf{0.36_{0.12}}$ | $\mathbf{0.36_{0.13}}$ |
| | | 50 | / | / | $1.47_{0.11}$ | / | $1.64_{0.03}$ | $1.53_{0.16}$ | $\mathbf{1.36_{0.23}}$ | $1.44_{0.16}$ |
| | 50 | 200 | $1.68_{0.02}$ | $1.59_{0.04}$ | $1.55_{0.06}$ | $1.65_{0.03}$ | $1.65_{0.03}$ | $\mathbf{1.09_{0.51}}$ | $1.08_{0.35}$ | $1.18_{0.36}$ |
| 6 | | 500 | $1.67_{0.03}$ | $1.54_{0.05}$ | $1.53_{0.06}$ | $1.49_{0.07}$ | $1.47_{0.06}$ | $\mathbf{0.66_{0.65}}$ | $0.67_{0.31}$ | $\mathbf{0.66_{0.32}}$ |
| | | 50 | / | / | $1.54_{0.11}$ | / | $1.68_{0.02}$ | $\mathbf{1.50_{0.21}}$ | $\mathbf{1.51_{0.17}}$ | $1.56_{0.14}$ |
| $(d_0 = 3)$ | 100 | 200 | $1.71_{0.03}$ | $1.68_{0.02}$ | $1.56_{0.05}$ | $1.70_{0.01}$ | $1.69_{0.02}$ | $\mathbf{1.29_{0.46}}$ | $\mathbf{1.31_{0.23}}$ | $1.41_{0.21}$ |
| | | 500 | $1.70_{0.01}$ | $1.64_{0.02}$ | $1.63_{0.04}$ | $1.67_{0.03}$ | $1.69_{0.02}$ | $\mathbf{0.88_{0.65}}$ | $1.02_{0.34}$ | $1.12_{0.38}$ |
| | 200 | 50 | / | / | $1.57_{0.10}$ | / | $1.71_{0.01}$ | $\mathbf{1.48_{0.19}}$ | $1.52_{0.19}$ | $1.52_{0.19}$ |
| | | 200 | / | / | $1.59_{0.06}$ | / | $1.71_{0.01}$ | $1.57_{0.19}$ | $\mathbf{1.39_{0.28}}$ | $\mathbf{1.39_{0.28}}$ |
| | 500 | 50 | / | / | $1.64_{0.09}$ | / | $1.72_{0.00}$ | $\mathbf{1.51_{0.17}}$ | $1.61_{0.16}$ | $1.61_{0.16}$ |
| | | 200 | / | / | $1.60_{0.06}$ | / | $1.72_{0.00}$ | $1.57_{0.20}$ | $\mathbf{1.50_{0.19}}$ | $\mathbf{1.50_{0.19}}$ |

Table 3: The average sin-theta distance between $\hat{A}_0$ and $A_0$ over 100 repeats for Models 4-6, with $p \in \{20, 50, 100, 200, 500\}$ and $n \in \{50, 200, 500\}$. We compare two versions of our method with six competing approaches. For each setting, we also present 10 times the standard error for each method as a subscript. The "/" entries indicate that the method is not applicable.

| Setting | | | Competing Methods | | | | | | Our Methods | |
|---|---|---|---|---|---|---|---|---|---|---|
| Model | $p$ | $n$ | SIR | pHd | MAVE | DR | gKDR | drMARS | RPE | RPE2 |
| | 20 | 50 | $1.34_{0.05}$ | $1.29_{0.06}$ | $1.23_{0.11}$ | $1.33_{0.06}$ | $1.30_{0.08}$ | $\mathbf{1.09_{0.24}}$ | $1.13_{0.19}$ | $1.15_{0.18}$ |
| | | 200 | $1.34_{0.05}$ | $1.13_{0.11}$ | $1.05_{0.15}$ | $1.07_{0.08}$ | $1.02_{0.07}$ | $\mathbf{0.60_{0.23}}$ | $0.65_{0.19}$ | $\mathbf{0.61_{0.22}}$ |
| | | 500 | $1.33_{0.06}$ | $0.99_{0.13}$ | $0.54_{0.22}$ | $0.95_{0.09}$ | $0.78_{0.16}$ | $0.55_{0.16}$ | $\mathbf{0.55_{0.02}}$ | $0.54_{0.17}$ |
| | 50 | 50 | / | / | $1.30_{0.09}$ | / | $1.36_{0.03}$ | $1.28_{0.16}$ | $\mathbf{1.22_{0.18}}$ | $1.26_{0.17}$ |
| | | 200 | $1.39_{0.02}$ | $1.31_{0.05}$ | $1.29_{0.07}$ | $1.33_{0.06}$ | $1.37_{0.04}$ | $\mathbf{0.67_{0.29}}$ | $0.79_{0.22}$ | $0.78_{0.27}$ |
| 7 | | 500 | $1.39_{0.02}$ | $1.21_{0.09}$ | $1.20_{0.09}$ | $1.10_{0.05}$ | $1.20_{0.09}$ | $0.56_{0.20}$ | $0.57_{0.01}$ | $\mathbf{0.53_{0.11}}$ |
| | 100 | 50 | / | / | $1.35_{0.07}$ | / | $1.39_{0.02}$ | $\mathbf{1.25_{0.17}}$ | $1.31_{0.13}$ | $1.33_{0.11}$ |
| $(d_0 = 2)$ | | 200 | $1.40_{0.01}$ | $1.39_{0.02}$ | $1.32_{0.06}$ | $1.40_{0.01}$ | $1.39_{0.02}$ | $\mathbf{0.87_{0.31}}$ | $0.95_{0.26}$ | $1.01_{0.26}$ |
| | | 500 | $1.40_{0.01}$ | $1.33_{0.04}$ | $1.33_{0.06}$ | $1.30_{0.06}$ | $1.39_{0.02}$ | $\mathbf{0.53_{0.17}}$ | $0.58_{0.01}$ | $0.56_{0.08}$ |
| | 200 | 50 | / | / | $1.38_{0.06}$ | / | $1.40_{0.01}$ | $\mathbf{1.27_{0.17}}$ | $1.33_{0.14}$ | $1.33_{0.14}$ |
| | | 200 | / | / | $1.34_{0.06}$ | / | $1.40_{0.01}$ | $1.18_{0.21}$ | $\mathbf{1.00_{0.24}}$ | $0.99_{0.25}$ |
| | 500 | 50 | / | / | $1.39_{0.06}$ | / | $1.41_{0.00}$ | $\mathbf{1.23_{0.18}}$ | $1.35_{0.12}$ | $1.35_{0.12}$ |
| | | 200 | / | / | $1.36_{0.04}$ | / | $1.41_{0.00}$ | $1.17_{0.20}$ | $\mathbf{1.03_{0.26}}$ | $\mathbf{1.03_{0.26}}$ |
| | 20 | 50 | $1.30_{0.08}$ | $1.21_{0.07}$ | $1.02_{0.11}$ | $1.29_{0.07}$ | $1.16_{0.13}$ | $0.38_{0.30}$ | $\mathbf{0.21_{0.15}}$ | $\mathbf{0.20_{0.12}}$ |
| | | 200 | $1.03_{0.15}$ | $1.03_{0.07}$ | $0.71_{0.12}$ | $0.92_{0.13}$ | $0.83_{0.13}$ | $0.20_{0.13}$ | $\mathbf{0.08_{0.02}}$ | $\mathbf{0.08_{0.02}}$ |
| | | 500 | $0.68_{0.13}$ | $0.88_{0.19}$ | $0.38_{0.07}$ | $0.68_{0.14}$ | $0.58_{0.12}$ | $0.07_{0.05}$ | $\mathbf{0.07_{0.01}}$ | $\mathbf{0.07_{0.01}}$ |
| | 50 | 50 | / | / | $1.11_{0.11}$ | / | $1.32_{0.05}$ | $0.73_{0.30}$ | $\mathbf{0.24_{0.24}}$ | $0.27_{0.24}$ |
| | | 200 | $1.34_{0.05}$ | $1.20_{0.04}$ | $1.02_{0.06}$ | $1.16_{0.06}$ | $1.13_{0.09}$ | $0.26_{0.16}$ | $\mathbf{0.08_{0.01}}$ | $\mathbf{0.08_{0.01}}$ |
| 8 | | 500 | $1.06_{0.11}$ | $1.07_{0.02}$ | $0.82_{0.08}$ | $0.94_{0.10}$ | $0.83_{0.08}$ | $0.14_{0.10}$ | $\mathbf{0.07_{0.01}}$ | $\mathbf{0.07_{0.01}}$ |
| | 100 | 50 | / | / | $1.15_{0.10}$ | / | $1.37_{0.02}$ | $0.81_{0.27}$ | $\mathbf{0.33_{0.31}}$ | $0.35_{0.29}$ |
| $(d_0 = 2)$ | | 200 | $1.40_{0.01}$ | $1.36_{0.02}$ | $1.06_{0.06}$ | $1.32_{0.04}$ | $1.28_{0.05}$ | $0.36_{0.14}$ | $\mathbf{0.07_{0.01}}$ | $0.08_{0.02}$ |
| | | 500 | $1.34_{0.04}$ | $1.19_{0.04}$ | $0.99_{0.05}$ | $1.10_{0.05}$ | $1.06_{0.07}$ | $0.17_{0.12}$ | $\mathbf{0.07_{0.01}}$ | $\mathbf{0.07_{0.01}}$ |
| | 200 | 50 | / | / | $1.18_{0.11}$ | / | $1.39_{0.01}$ | $0.79_{0.31}$ | $\mathbf{0.54_{0.38}}$ | $\mathbf{0.54_{0.38}}$ |
| | | 200 | / | / | $1.08_{0.06}$ | / | $1.37_{0.02}$ | $0.88_{0.21}$ | $\mathbf{0.10_{0.02}}$ | $\mathbf{0.10_{0.02}}$ |
| | 500 | 50 | / | / | $1.21_{0.09}$ | / | $1.41_{0.00}$ | $0.83_{0.30}$ | $\mathbf{0.74_{0.35}}$ | $\mathbf{0.74_{0.35}}$ |
| | | 200 | / | / | $1.11_{0.07}$ | / | $1.40_{0.01}$ | $0.89_{0.19}$ | $\mathbf{0.18_{0.14}}$ | $\mathbf{0.18_{0.14}}$ |
| | 20 | 50 | $1.30_{0.07}$ | $1.31_{0.06}$ | $1.21_{0.09}$ | $1.32_{0.07}$ | $1.22_{0.07}$ | $\mathbf{0.95_{0.32}}$ | $1.00_{0.15}$ | $1.02_{0.16}$ |
| | | 200 | $1.06_{0.05}$ | $1.17_{0.08}$ | $1.12_{0.06}$ | $1.18_{0.09}$ | $1.17_{0.07}$ | $0.31_{0.43}$ | $\mathbf{0.18_{0.21}}$ | $0.18_{0.22}$ |
| | | 500 | $1.00_{0.04}$ | $1.08_{0.05}$ | $1.03_{0.04}$ | $1.06_{0.04}$ | $1.08_{0.05}$ | $0.14_{0.31}$ | $\mathbf{0.07_{0.02}}$ | $\mathbf{0.07_{0.02}}$ |
| | 50 | 50 | / | / | $1.25_{0.08}$ | / | $1.33_{0.04}$ | $1.24_{0.15}$ | $\mathbf{1.07_{0.16}}$ | $1.12_{0.17}$ |
| | | 200 | $1.30_{0.06}$ | $1.33_{0.04}$ | $1.22_{0.04}$ | $1.30_{0.05}$ | $1.18_{0.04}$ | $0.69_{0.43}$ | $\mathbf{0.27_{0.30}}$ | $0.28_{0.30}$ |
| 9 | | 500 | $1.09_{0.03}$ | $1.21_{0.05}$ | $1.15_{0.03}$ | $1.18_{0.06}$ | $1.15_{0.04}$ | $0.47_{0.48}$ | $\mathbf{0.09_{0.06}}$ | $\mathbf{0.09_{0.06}}$ |
| $(d_0 = 2)$ | 100 | 50 | / | / | $1.28_{0.07}$ | / | $1.37_{0.02}$ | $1.24_{0.16}$ | $\mathbf{1.07_{0.18}}$ | $1.13_{0.15}$ |
| | | 200 | $1.40_{0.03}$ | $1.39_{0.02}$ | $1.23_{0.05}$ | $1.34_{0.03}$ | $1.27_{0.05}$ | $0.89_{0.36}$ | $\mathbf{0.35_{0.35}}$ | $0.37_{0.36}$ |
| | | 500 | $1.26_{0.05}$ | $1.33_{0.04}$ | $1.22_{0.03}$ | $1.26_{0.04}$ | $1.15_{0.02}$ | $0.53_{0.48}$ | $\mathbf{0.11_{0.18}}$ | $\mathbf{0.11_{0.18}}$ |
| | 200 | 50 | / | / | $1.30_{0.07}$ | / | $1.39_{0.01}$ | $1.22_{0.22}$ | $1.09_{0.16}$ | $\mathbf{1.09_{0.15}}$ |
| | | 200 | / | / | $1.26_{0.05}$ | / | $1.35_{0.02}$ | $1.25_{0.20}$ | $\mathbf{0.34_{0.39}}$ | $\mathbf{0.34_{0.39}}$ |
| | 500 | 50 | / | / | $1.32_{0.07}$ | / | $1.41_{0.00}$ | $1.24_{0.22}$ | $\mathbf{1.13_{0.20}}$ | $\mathbf{1.13_{0.20}}$ |
| | | 200 | / | / | $1.28_{0.05}$ | / | $1.39_{0.01}$ | $1.22_{0.25}$ | $\mathbf{0.28_{0.37}}$ | $\mathbf{0.28_{0.37}}$ |

Table 4: The average sin-theta distance between $\hat{A}_0$ and $A_0$ over 100 repeats for Models 7-9, with $p \in \{20, 50, 100, 200, 500\}$ and $n \in \{50, 200, 500\}$. We compare two versions of our method with six competing approaches. For each setting, we also present 10 times the standard error for each method as a subscript. The "/" entries indicate that the method is not applicable.

## C.2 Dimension $d_0$ unknown

In this subsection, our primary goal is to evaluate the performance of Algorithm 1 when the dimension of the final projection, $\hat{d}_0$, is chosen via Algorithm 2. We therefore focus on a single application of our random projection ensemble algorithm. The double dimension reduction technique in Algorithm 3 is not considered here, as it requires a prespecified target dimension. For comparison, we also include the results from the competing methods, with their projection dimension selected by the corresponding recommended default approaches–specifically, the *marginal dimension test* from the dr R package for SIR and pHd; *generalised cross-validation*, which is available as part of the mave package (for MAVE); and the corresponding GitHub links referenced in the previous subsection (for gKDR and drMARS – for these two, we set the maximum projection dimension to be $\lceil \sqrt{p} \rceil$). The DR method is only applicable with a prespecified $d_0$, and it is therefore excluded from this subsection.

In the experiments below, we reuse the Models 1a-9 described earlier. Our proposal, labelled **RPE** in the boxplots, follows the recommendations from Section 3 for Algorithm 1, with $\hat{d}_0$ selected by Algorithm 2 using $R = 10000$. Competing methods utilize their respective default values for tuning parameters.

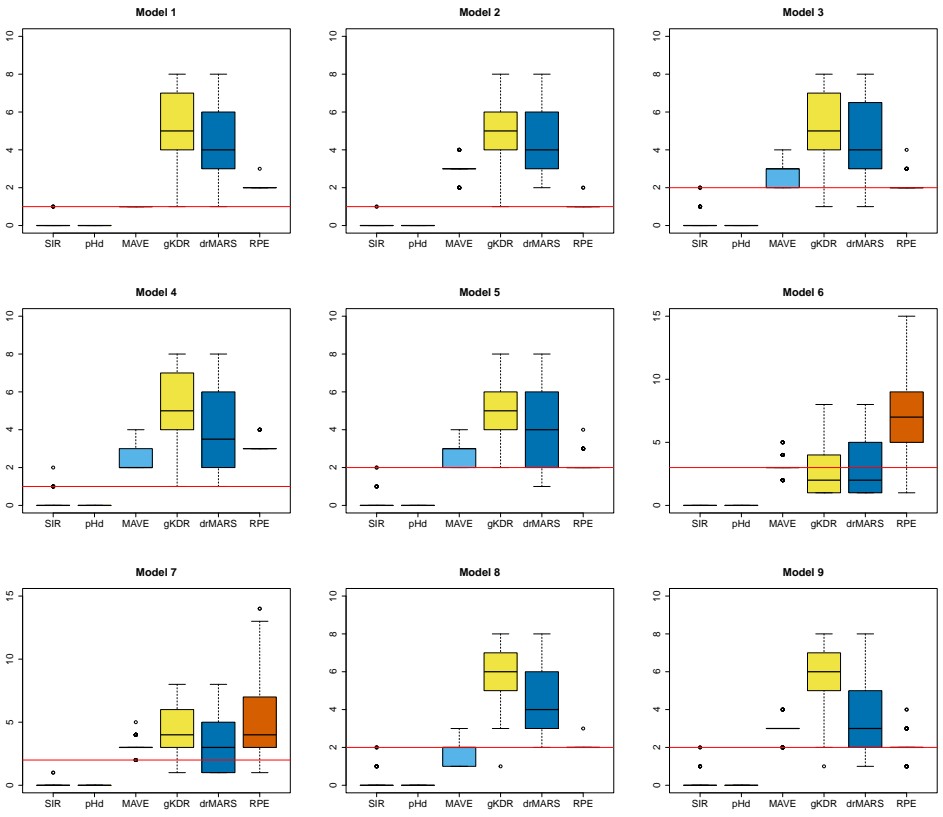

Figure 13: Boxplots of $\hat{d}_0$ from Algorithm 2 alongside competing methods over 100 repeats for models 1a-9. We present the results for $n = 200$ and $p = 50$. The red horizontal lines indicate the true dimension $d_0$.

The results in Figure 13 show that, with the exception of SIR and pHd, most methods including ours tend to be somewhat liberal in choosing $\hat{d}_0$, in the sense that they typically select $\hat{d}_0 > d_0$. By contrast, SIR and pHd typically select $\hat{d}_0 = 0$ in these examples, indicating no signal is detected – as mentioned above, both methods appear to suffer from the curse of dimensionality.

To further assess the performance when $d_0$ is unknown, we consider two complementary metrics. We introduce the *false positives* measure

$$d_{\mathrm{FP}}(\hat{A}_0, A_0) := \left\| (I - A_0 A_0^\top)\hat{A}_0 \right\|_F \tag{16}$$

and we recall the *false negatives* measure from Appendix A.5:

$$d_{\mathrm{FN}}(\hat{A}_0, A_0) = \left\| (I - \hat{A}_0 \hat{A}_0^\top)A_0 \right\|_F \tag{17}$$

In both definitions, $\hat{A}_0$ is a $p \times \hat{d}_0$ projection, where $\hat{d}_0$ need not equal $d_0$. The metric $d_{\mathrm{FP}}$ measures *false positives*, in the sense that it quantifies the extent to which $\hat{A}_0$ contains regions of the ambient $p$-dimensional space orthogonal to $A_0$. In particular, if $\mathcal{S}(\hat{A}_0) \subseteq \mathcal{S}(A_0)$, then $d_{\mathrm{FP}}(\hat{A}_0, A_0) = 0$. In contrast, $d_{\mathrm{FN}}$ measures *false negatives* and quantifies the amount of the space spanned by $A_0$ that is *missed* by the projection $\hat{A}_0$. Indeed, we have $d_{\mathrm{FN}}(\hat{A}_0, A_0) = 0$ when $\mathcal{S}(A_0) \subseteq \mathcal{S}(\hat{A}_0)$.

Note, however, that a small value of $d_{\mathrm{FP}}$ or $d_{\mathrm{FN}}$ on its own does not guarantee good performance: selecting no directions yields $d_{\mathrm{FP}} = 0$ but a large $d_{\mathrm{FN}}$, while selecting the full $p$-dimensional ambient space yields $d_{\mathrm{FN}} = 0$ but a large $d_{\mathrm{FP}}$. In some of our simulations, SIR and pHd effectively behave like the former, returning only a nearly trivial subspace and thus having small $d_{\mathrm{FP}}$ but very large $d_{\mathrm{FN}}$. Broadly speaking, increasing $\hat{d}_0$ typically decreases $d_{\mathrm{FN}}$ at the expense of increasing $d_{\mathrm{FP}}$. Only when $\hat{d}_0 = d_0$ can both $d_{\mathrm{FP}}$ and $d_{\mathrm{FN}}$ simultaneously be (or be close to) zero. Finally, we also have the identity

$$2d_{\mathrm{F}}^2(\hat{A}_0, A_0) = d_{\mathrm{FP}}^2(\hat{A}_0, A_0) + d_{\mathrm{FN}}^2(\hat{A}_0, A_0).$$

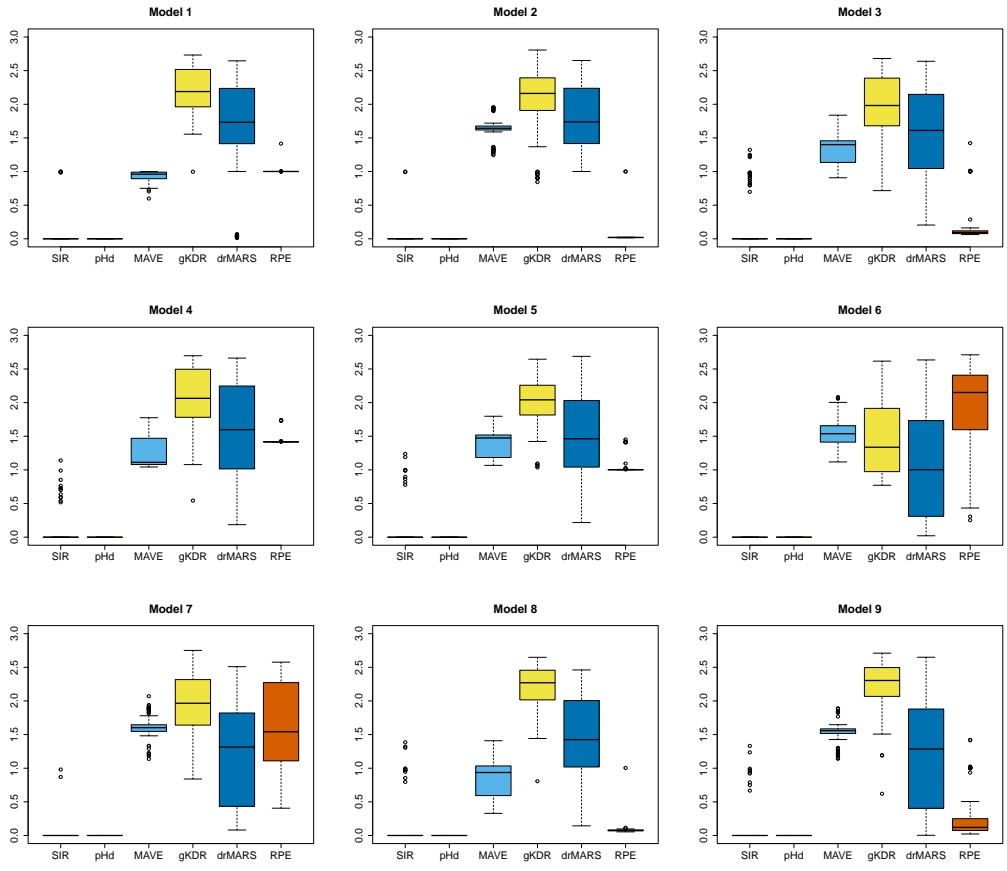

Figure 14: Boxplots of $d_{\mathrm{FP}}(\hat{A}_0, A_0)$ for the different methods when $d_0$ is unknown, for models 1a-9 over 100 simulations, with $n = 200$, $p = 50$. Our RPE method uses Algorithm 1, with $\hat{d}_0$ chosen via Algorithm 2. For competing methods, $\hat{d}_0$ is chosen by the corresponding approaches described at the beginning of this subsection.

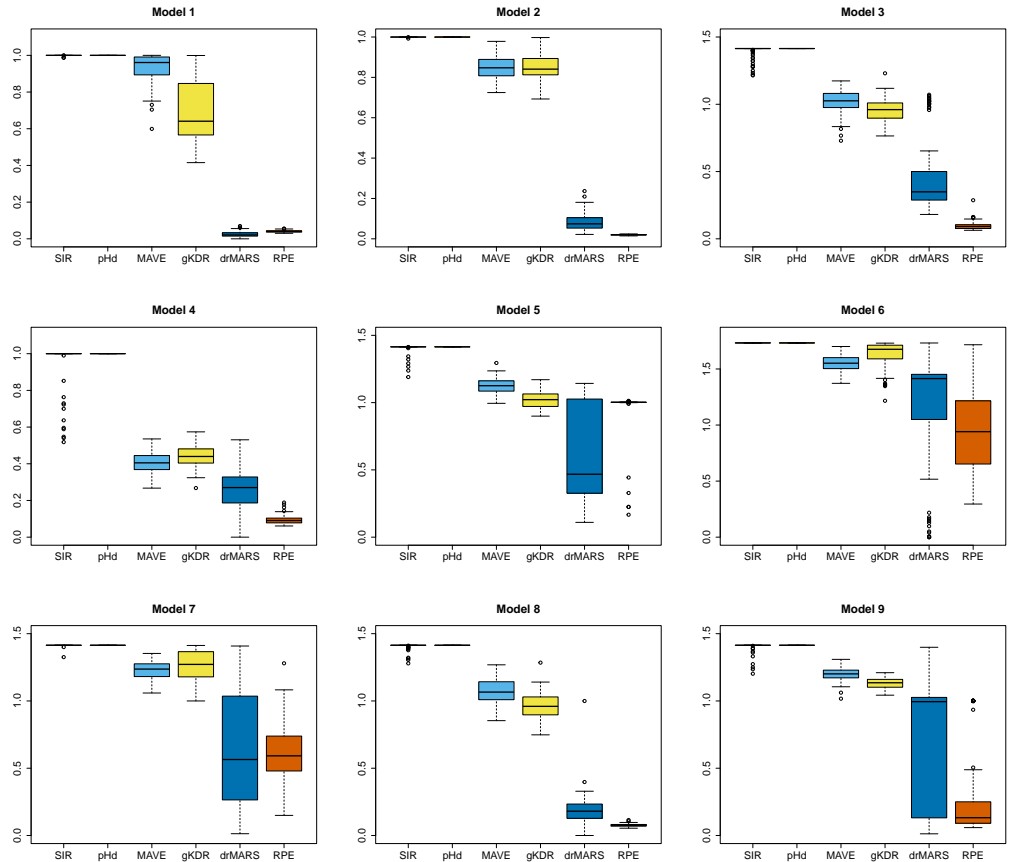

Figure 15: Boxplots of $d_{\mathrm{FN}}(\hat{A}_0, A_0)$ for the different methods when $d_0$ is unknown, for models 1a-9 over 100 simulations, with $n = 200$, $p = 50$. Our RPE method uses Algorithm 1, with $\hat{d}_0$ chosen via Algorithm 2. For competing methods, $\hat{d}_0$ is chosen by the corresponding approaches described at the beginning of this subsection.

Figures 14 and 15 present the results of $d_{\mathrm{FP}}$ and $d_{\mathrm{FN}}$, respectively, for different methods across the nine models. When $\hat{d}_0$ is frequently selected equal $d_0$, namely Models 3, 8 and 9, we observe excellent performance in terms of both false positives and false negatives. In Models 1 and 4, our algorithm tends to identify the one-dimensional signal direction by selecting the individual coordinate axes that contribute to the true projection. In these cases, we perform well in terms of the false-negative metric, indicating that nearly all of the signal is captured. However, this comes at the cost of higher false positives, though our approach remains competitive with the other methods considered. For Models 2 and 6, we often take $\hat{d}_0$ larger than $d_0$, thereby including some noise directions alongside the signals. Nevertheless, our approach still excels in terms of false negatives, outperforming the other methods considered–especially in Model 2, where the competing methods fail to capture the signal directions–although in Model 6 this advantage comes at the cost of higher false positives. Finally, in Models 5 and 7, our algorithm tends to select the $e_1$ and $e_2$ directions separately while missing $e_3$, and thus also fails to combine these signals in the optimal way. As a result, neither false-positive nor false-negative is close to zero. Nevertheless, our method remains competitive with the other algorithms considered in these examples.

Tables 5 and 6 show the full results for the false-positive and false-negative metrics for additional experiments corresponding to those in Section C.2. We present the results for $p \in \{20, 50, 100, 200, 500\}$ with $n = 200$. Results for $n \in \{50, 500\}$ are broadly similar and are omitted for brevity.

| Setting | | Competing Methods | | | | | Our Methods |
|---|---|---|---|---|---|---|---|
| Model | $p$ | SIR | pHd | MAVE | gKDR | drMARS | RPE |
| 1a ($d_0 = 1$) | 20 | $0.06_{0.23}$ | $0.07_{0.17}$ | $0.24_{0.04}$ | $1.48_{0.60}$ | $1.15_{0.76}$ | $1.00_{0.00}$ |
| | 50 | $0.04_{0.20}$ | $0.00_{0.00}$ | $0.93_{0.08}$ | $2.22_{0.33}$ | $1.54_{0.80}$ | $1.00_{0.04}$ |
| | 100 | $0.00_{0.00}$ | $0.00_{0.00}$ | $0.97_{0.04}$ | $2.87_{0.26}$ | $1.82_{0.84}$ | $1.07_{0.21}$ |
| | 200 | / | / | $0.97_{0.04}$ | $2.84_{0.26}$ | $1.75_{0.90}$ | $1.00_{0.00}$ |
| | 500 | / | / | $0.98_{0.03}$ | $2.76_{0.26}$ | $1.73_{0.92}$ | $1.01_{0.07}$ |
| 2 ($d_0 = 1$) | 20 | $0.88_{0.11}$ | $0.98_{0.02}$ | $1.59_{0.23}$ | $1.46_{0.44}$ | $1.47_{0.45}$ | $0.02_{0.00}$ |
| | 50 | $0.98_{0.02}$ | $0.99_{0.01}$ | $1.63_{0.15}$ | $2.08_{0.52}$ | $1.82_{0.57}$ | $0.04_{0.14}$ |
| | 100 | $0.99_{0.02}$ | $1.00_{0.01}$ | $1.62_{0.14}$ | $2.59_{0.41}$ | $1.86_{0.75}$ | $0.12_{0.31}$ |
| | 200 | / | / | $1.63_{0.15}$ | $2.76_{0.39}$ | $2.19_{0.58}$ | $0.01_{0.00}$ |
| | 500 | / | / | $1.59_{0.16}$ | $2.70_{0.33}$ | $2.07_{0.56}$ | $0.01_{0.00}$ |
| 3 ($d_0 = 2$) | 20 | $0.46_{0.25}$ | $0.04_{0.15}$ | $0.92_{0.30}$ | $1.23_{0.52}$ | $1.06_{0.63}$ | $0.10_{0.02}$ |
| | 50 | $0.19_{0.40}$ | $0.00_{0.00}$ | $1.34_{0.21}$ | $1.94_{0.46}$ | $1.50_{0.73}$ | $0.19_{0.29}$ |
| | 100 | $0.00_{0.00}$ | $0.00_{0.00}$ | $1.35_{0.20}$ | $2.55_{0.41}$ | $1.54_{0.84}$ | $0.35_{0.47}$ |
| | 200 | / | / | $1.40_{0.19}$ | $2.62_{0.31}$ | $1.86_{0.64}$ | $0.12_{0.10}$ |
| | 500 | / | / | $1.42_{0.18}$ | $2.75_{0.26}$ | $1.88_{0.65}$ | $0.07_{0.06}$ |
| 4 ($d_0 = 1$) | 20 | $0.32_{0.19}$ | $0.00_{0.00}$ | $1.06_{0.35}$ | $1.32_{0.61}$ | $1.09_{0.67}$ | $1.42_{0.00}$ |
| | 50 | $0.09_{0.25}$ | $0.00_{0.00}$ | $1.27_{0.21}$ | $2.10_{0.49}$ | $1.50_{0.81}$ | $1.44_{0.09}$ |
| | 100 | $0.00_{0.00}$ | $0.00_{0.00}$ | $1.28_{0.20}$ | $2.49_{0.45}$ | $1.72_{0.83}$ | $1.54_{0.18}$ |
| | 200 | / | / | $1.28_{0.20}$ | $2.54_{0.34}$ | $1.77_{0.70}$ | $1.40_{0.07}$ |
| | 500 | / | / | $1.27_{0.18}$ | $2.66_{0.33}$ | $1.82_{0.71}$ | $1.36_{0.11}$ |
| 5 ($d_0 = 2$) | 20 | $0.35_{0.21}$ | $0.01_{0.10}$ | $0.98_{0.34}$ | $1.45_{0.47}$ | $0.97_{0.55}$ | $1.01_{0.01}$ |
| | 50 | $0.09_{0.28}$ | $0.00_{0.00}$ | $1.40_{0.19}$ | $2.06_{0.39}$ | $1.47_{0.67}$ | $1.05_{0.13}$ |
| | 100 | $0.00_{0.00}$ | $0.00_{0.00}$ | $1.41_{0.18}$ | $2.52_{0.40}$ | $1.55_{0.72}$ | $1.19_{0.25}$ |
| | 200 | / | / | $1.44_{0.17}$ | $2.61_{0.39}$ | $1.88_{0.65}$ | $1.00_{0.04}$ |
| | 500 | / | / | $1.46_{0.18}$ | $2.67_{0.33}$ | $1.84_{0.66}$ | $0.98_{0.07}$ |
| 6 ($d_0 = 3$) | 20 | $0.03_{0.15}$ | $0.00_{0.00}$ | $1.26_{0.29}$ | $0.85_{0.37}$ | $0.59_{0.57}$ | $1.01_{0.46}$ |
| | 50 | $0.00_{0.00}$ | $0.00_{0.00}$ | $1.55_{0.23}$ | $1.47_{0.56}$ | $1.10_{0.79}$ | $1.97_{0.56}$ |
| | 100 | $0.00_{0.00}$ | $0.00_{0.00}$ | $1.56_{0.22}$ | $2.55_{0.50}$ | $1.36_{0.73}$ | $2.47_{0.62}$ |
| | 200 | / | / | $1.61_{0.22}$ | $2.87_{0.30}$ | $1.59_{0.67}$ | $0.64_{0.37}$ |
| | 500 | / | / | $1.52_{0.19}$ | $2.91_{0.31}$ | $1.58_{0.74}$ | $0.65_{0.37}$ |
| 7 ($d_0 = 2$) | 20 | $0.02_{0.13}$ | $0.00_{0.00}$ | $1.31_{0.32}$ | $1.00_{0.53}$ | $0.87_{0.57}$ | $0.99_{0.44}$ |
| | 50 | $0.02_{0.13}$ | $0.00_{0.00}$ | $1.61_{0.17}$ | $1.91_{0.55}$ | $1.23_{0.77}$ | $1.63_{0.65}$ |
| | 100 | $0.00_{0.00}$ | $0.00_{0.00}$ | $1.63_{0.16}$ | $2.56_{0.41}$ | $1.52_{0.82}$ | $2.26_{0.79}$ |
| | 200 | / | / | $1.63_{0.15}$ | $2.60_{0.39}$ | $1.64_{0.68}$ | $0.64_{0.27}$ |
| | 500 | / | / | $1.62_{0.14}$ | $2.71_{0.33}$ | $1.90_{0.60}$ | $0.61_{0.26}$ |
| 8 ($d_0 = 2$) | 20 | $0.37_{0.34}$ | $0.11_{0.17}$ | $0.48_{0.24}$ | $1.56_{0.40}$ | $0.99_{0.63}$ | $0.08_{0.02}$ |
| | 50 | $0.12_{0.35}$ | $0.00_{0.00}$ | $0.82_{0.25}$ | $2.23_{0.35}$ | $1.42_{0.71}$ | $0.09_{0.09}$ |
| | 100 | $0.00_{0.00}$ | $0.00_{0.00}$ | $0.90_{0.22}$ | $2.65_{0.35}$ | $1.62_{0.80}$ | $0.18_{0.31}$ |
| | 200 | / | / | $0.97_{0.22}$ | $2.70_{0.29}$ | $1.83_{0.64}$ | $0.10_{0.02}$ |
| | 500 | / | / | $1.04_{0.20}$ | $2.73_{0.23}$ | $1.89_{0.65}$ | $0.11_{0.06}$ |
| 9 ($d_0 = 2$) | 20 | $0.29_{0.28}$ | $0.00_{0.00}$ | $1.45_{0.24}$ | $1.63_{0.40}$ | $0.86_{0.64}$ | $0.11_{0.14}$ |
| | 50 | $0.10_{0.29}$ | $0.00_{0.00}$ | $1.53_{0.16}$ | $2.23_{0.38}$ | $1.26_{0.80}$ | $0.28_{0.36}$ |
| | 100 | $0.00_{0.00}$ | $0.00_{0.00}$ | $1.52_{0.16}$ | $2.67_{0.39}$ | $1.74_{0.81}$ | $0.41_{0.51}$ |
| | 200 | / | / | $1.53_{0.17}$ | $2.52_{0.45}$ | $1.90_{0.68}$ | $0.06_{0.04}$ |
| | 500 | / | / | $1.51_{0.18}$ | $2.51_{0.39}$ | $2.00_{0.67}$ | $0.04_{0.02}$ |

Table 5: The average of $d_{\text{FP}}(\mathcal{S}(\hat{A}_0), \mathcal{S}(A_0))$ over 100 repeats of the experiment for Models 1a-9, with $p \in \{20, 50, 100, 200, 500\}$ and $n = 200$. For each setting, we also present 10 times the standard error for each method as a subscript. The "/" entries indicate that the method is not applicable.

| Setting | | Competing Methods | | | | | Our Methods |
|---|---|---|---|---|---|---|---|
| Model | $p$ | SIR | pHd | MAVE | gKDR | drMARS | RPE |
| | 20 | $0.99_{0.04}$ | $0.88_{0.25}$ | $0.24_{0.04}$ | $0.25_{0.06}$ | $0.02_{0.02}$ | $0.05_{0.01}$ |
| | 50 | $1.00_{0.00}$ | $1.00_{0.00}$ | $0.93_{0.08}$ | $0.69_{0.15}$ | $0.02_{0.01}$ | $0.04_{0.01}$ |
| 1a ($d_0 = 1$) | 100 | $1.00_{0.00}$ | $1.00_{0.00}$ | $0.97_{0.04}$ | $0.83_{0.07}$ | $0.03_{0.02}$ | $0.04_{0.00}$ |
| | 200 | / | / | $0.97_{0.04}$ | $0.94_{0.03}$ | $0.12_{0.15}$ | $0.05_{0.01}$ |
| | 500 | / | / | $0.98_{0.03}$ | $0.98_{0.01}$ | $0.12_{0.10}$ | $0.06_{0.01}$ |
| | 20 | $0.88_{0.11}$ | $0.98_{0.02}$ | $0.71_{0.10}$ | $0.81_{0.12}$ | $0.07_{0.04}$ | $0.02_{0.00}$ |
| | 50 | $0.98_{0.02}$ | $0.99_{0.01}$ | $0.85_{0.06}$ | $0.85_{0.06}$ | $0.08_{0.04}$ | $0.02_{0.00}$ |
| 2 ($d_0 = 1$) | 100 | $0.99_{0.02}$ | $1.00_{0.01}$ | $0.89_{0.05}$ | $0.91_{0.04}$ | $0.09_{0.04}$ | $0.02_{0.00}$ |
| | 200 | / | / | $0.90_{0.05}$ | $0.94_{0.02}$ | $0.14_{0.08}$ | $0.01_{0.00}$ |
| | 500 | / | / | $0.93_{0.04}$ | $0.98_{0.01}$ | $0.15_{0.14}$ | $0.01_{0.00}$ |
| | 20 | $1.07_{0.16}$ | $1.40_{0.07}$ | $0.69_{0.12}$ | $0.88_{0.17}$ | $0.43_{0.34}$ | $0.10_{0.02}$ |
| | 50 | $1.40_{0.05}$ | $1.41_{0.00}$ | $1.02_{0.09}$ | $0.96_{0.08}$ | $0.48_{0.28}$ | $0.10_{0.03}$ |
| 3 ($d_0 = 2$) | 100 | $1.41_{0.00}$ | $1.41_{0.00}$ | $1.05_{0.09}$ | $1.12_{0.06}$ | $0.56_{0.30}$ | $0.10_{0.09}$ |
| | 200 | / | / | $1.08_{0.08}$ | $1.26_{0.03}$ | $0.93_{0.31}$ | $0.15_{0.18}$ |
| | 500 | / | / | $1.13_{0.07}$ | $1.34_{0.01}$ | $0.97_{0.30}$ | $0.64_{0.43}$ |
| | 20 | $0.35_{0.19}$ | $1.00_{0.00}$ | $0.27_{0.05}$ | $0.35_{0.07}$ | $0.18_{0.08}$ | $0.11_{0.03}$ |
| | 50 | $0.96_{0.11}$ | $1.00_{0.00}$ | $0.40_{0.05}$ | $0.44_{0.05}$ | $0.26_{0.10}$ | $0.09_{0.02}$ |
| 4 ($d_0 = 1$) | 100 | $1.00_{0.00}$ | $1.00_{0.00}$ | $0.43_{0.06}$ | $0.58_{0.05}$ | $0.32_{0.12}$ | $0.09_{0.02}$ |
| | 200 | / | / | $0.46_{0.06}$ | $0.75_{0.03}$ | $0.71_{0.22}$ | $0.11_{0.12}$ |
| | 500 | / | / | $0.51_{0.06}$ | $0.89_{0.02}$ | $0.68_{0.25}$ | $0.19_{0.21}$ |
| | 20 | $1.11_{0.13}$ | $1.41_{0.00}$ | $0.94_{0.14}$ | $0.88_{0.14}$ | $0.55_{0.40}$ | $0.93_{0.23}$ |
| | 50 | $1.41_{0.04}$ | $1.41_{0.00}$ | $1.13_{0.06}$ | $1.02_{0.06}$ | $0.61_{0.34}$ | $0.97_{0.16}$ |
| 5 ($d_0 = 2$) | 100 | $1.41_{0.00}$ | $1.41_{0.00}$ | $1.16_{0.05}$ | $1.15_{0.04}$ | $0.89_{0.31}$ | $0.95_{0.17}$ |
| | 200 | / | / | $1.18_{0.06}$ | $1.25_{0.02}$ | $1.16_{0.29}$ | $1.01_{0.03}$ |
| | 500 | / | / | $1.22_{0.05}$ | $1.34_{0.01}$ | $1.15_{0.24}$ | $1.02_{0.05}$ |
| | 20 | $1.73_{0.02}$ | $1.73_{0.00}$ | $1.26_{0.17}$ | $1.37_{0.22}$ | $1.04_{0.56}$ | $0.63_{0.25}$ |
| | 50 | $1.71_{0.00}$ | $1.73_{0.00}$ | $1.55_{0.07}$ | $1.63_{0.11}$ | $1.22_{0.48}$ | $0.93_{0.35}$ |
| 6 ($d_0 = 3$) | 100 | $1.73_{0.00}$ | $1.73_{0.00}$ | $1.57_{0.07}$ | $1.63_{0.05}$ | $1.28_{0.50}$ | $1.15_{0.28}$ |
| | 200 | / | / | $1.58_{0.08}$ | $1.67_{0.02}$ | $1.56_{0.24}$ | $1.55_{0.17}$ |
| | 500 | / | / | $1.61_{0.07}$ | $1.70_{0.01}$ | $1.58_{0.20}$ | $1.59_{0.15}$ |
| | 20 | $1.41_{0.00}$ | $1.41_{0.00}$ | $0.98_{0.15}$ | $1.01_{0.11}$ | $0.61_{0.39}$ | $0.56_{0.20}$ |
| | 50 | $1.41_{0.01}$ | $1.41_{0.00}$ | $1.23_{0.06}$ | $1.27_{0.11}$ | $0.61_{0.40}$ | $0.62_{0.23}$ |
| 7 ($d_0 = 2$) | 100 | $1.41_{0.00}$ | $1.41_{0.00}$ | $1.28_{0.08}$ | $1.32_{0.05}$ | $0.81_{0.37}$ | $0.80_{0.26}$ |
| | 200 | / | / | $1.30_{0.07}$ | $1.37_{0.02}$ | $1.08_{0.30}$ | $1.07_{0.24}$ |
| | 500 | / | / | $1.34_{0.06}$ | $1.40_{0.01}$ | $0.98_{0.32}$ | $1.12_{0.22}$ |
| | 20 | $1.25_{0.16}$ | $1.31_{0.16}$ | $0.84_{0.19}$ | $0.73_{0.14}$ | $0.18_{0.18}$ | $0.08_{0.02}$ |
| | 50 | $1.41_{0.02}$ | $1.41_{0.00}$ | $1.07_{0.09}$ | $0.96_{0.10}$ | $0.19_{0.11}$ | $0.08_{0.01}$ |
| 8 ($d_0 = 2$) | 100 | $1.41_{0.00}$ | $1.41_{0.00}$ | $1.11_{0.09}$ | $1.15_{0.06}$ | $0.25_{0.10}$ | $0.07_{0.01}$ |
| | 200 | / | / | $1.11_{0.10}$ | $1.30_{0.03}$ | $0.60_{0.26}$ | $0.10_{0.02}$ |
| | 500 | / | / | $1.14_{0.09}$ | $1.37_{0.01}$ | $0.63_{0.27}$ | $0.35_{0.37}$ |
| | 20 | $1.24_{0.15}$ | $1.41_{0.08}$ | $1.08_{0.07}$ | $1.11_{0.10}$ | $0.31_{0.44}$ | $0.26_{0.34}$ |
| | 50 | $1.40_{0.04}$ | $1.41_{0.00}$ | $1.20_{0.05}$ | $1.13_{0.04}$ | $0.68_{0.44}$ | $0.29_{0.33}$ |
| 9 ($d_0 = 2$) | 100 | $1.41_{0.00}$ | $1.41_{0.00}$ | $1.21_{0.05}$ | $1.22_{0.04}$ | $0.85_{0.37}$ | $0.34_{0.36}$ |
| | 200 | / | / | $1.24_{0.05}$ | $1.31_{0.02}$ | $1.14_{0.28}$ | $0.45_{0.45}$ |
| | 500 | / | / | $1.26_{0.06}$ | $1.36_{0.01}$ | $1.08_{0.34}$ | $0.44_{0.46}$ |

Table 6: The average of $d_{\mathrm{FN}}(\mathcal{S}(\hat{A}_0), \mathcal{S}(A_0))$ over 100 repeats of the experiment for Models 1a-9, with $p \in \{20, 50, 100, 200, 500\}$ and $n = 200$. For each setting, we also present 10 times the standard error for each method as a subscript. The "/" entries indicate that the method is not applicable.

## C.3 COMPUTATIONAL COST

Table 7 reports the wall-clock runtimes in seconds for one run of Model 1a across different values of $n$ and $p$. We compare all competing methods considered and our method (label 'RPE') under three compute budgets (single core, 10 cores, 20 cores), all measured on the same 3.20 GHz Intel i9-

14900KF computer. The multi-core columns exhibit that our method achieves roughly $8\times$ speedup at 10 cores and $10$–$12\times$ at 20 cores.

| Setting | | Competing Methods | | | | | | Our Methods | | |
|---|---|---|---|---|---|---|---|---|---|---|
| $p$ | $n$ | SIR | pHd | MAVE | DR | gKDR | drMARS | RPE | RPE(10 cores) | RPE(20 cores) |
| | 50 | 0.007 | 0.007 | 0.051 | 0.009 | 0.015 | 0.138 | 104.819 | 14.940 | 11.677 |
| 20 | 200 | 0.007 | 0.005 | 0.408 | 0.007 | 0.078 | 0.211 | 135.390 | 18.826 | 14.208 |
| | 500 | 0.008 | 0.006 | 0.961 | 0.008 | 0.381 | 0.276 | 177.824 | 24.095 | 17.888 |
| | 50 | / | / | 0.047 | / | 0.026 | 0.185 | 311.972 | 40.821 | 29.769 |
| 50 | 200 | 0.011 | 0.007 | 0.827 | 0.013 | 0.224 | 0.478 | 422.228 | 54.199 | 38.174 |
| | 500 | 0.014 | 0.009 | 2.988 | 0.020 | 1.006 | 0.635 | 594.113 | 74.270 | 50.941 |
| | 50 | / | / | 0.047 | / | 0.109 | 0.188 | 702.826 | 89.105 | 62.658 |
| 100 | 200 | 0.032 | 0.016 | 0.820 | 0.018 | 0.778 | 0.841 | 970.345 | 121.098 | 82.853 |
| | 500 | 0.033 | 0.020 | 7.945 | 0.031 | 3.327 | 1.162 | 1477.162 | 181.603 | 116.395 |

Table 7: Runtime (seconds) for one run of Model 1a, with $p \in \{20, 50, 100\}$ and $n \in \{50, 200, 500\}$. The "/" entries denote that the method is not applicable.

While the classical SDR methods are faster in wall-clock time, many become inapplicable when $p \geq n$ and are typically ineffective in higher-dimensional regimes. By contrast, our random projection-based method delivers substantial improvements across settings and, in several scenarios, is the only approach achieving non-trivial performance (see Tables 2–4, and also the real data analysis in Appendix D). Thus, the additional computation is justified when accuracy matters, and parallel execution keeps runtimes manageable under the recommended defaults.

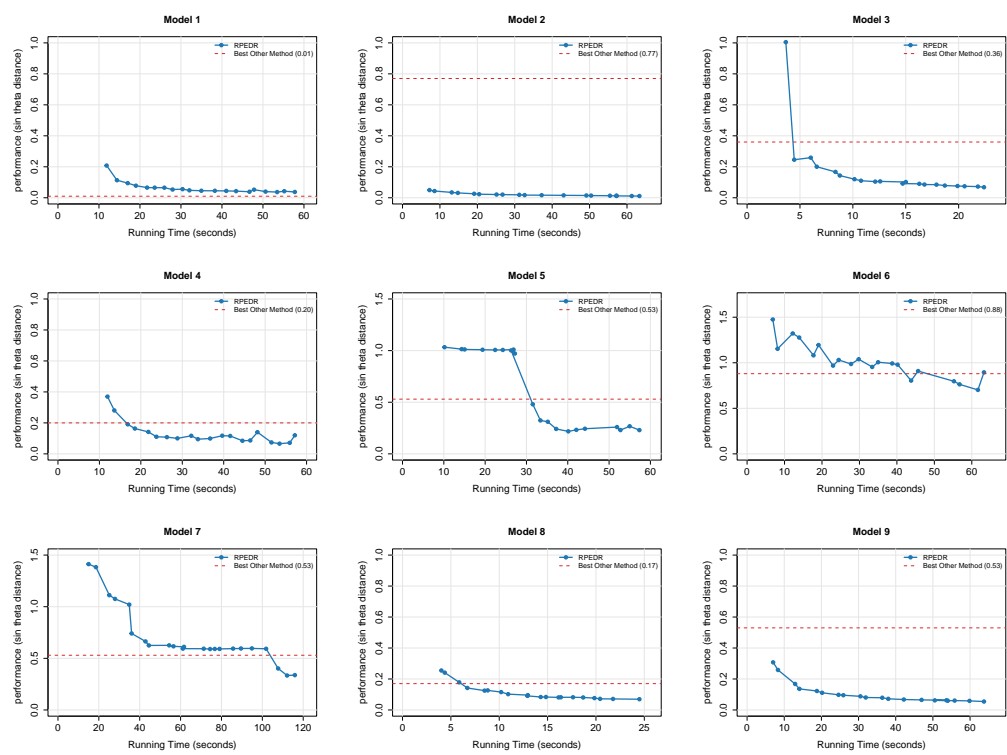

Figure 16: Performance in terms of sin theta distance versus running time of RPE-based method (20 cores of the same machine) across nine models, with $n = 500$ and $p = 100$. In models 1, 4, and 7, RPE2 is used. In Models 3 and 8, single application of RPE with QLS as the base method is used. In the other models, a single application of RPE with the MARS default is used.

In cases where computational resources may be limited, we can significantly reduce the computational cost and still remain competitive with (and often superior to) existing approaches. For some

examples, we could slightly reduce $M$ and $L$, use simpler projection distributions, and / or adopt a less computationally intensive base method. To demonstrate this, in Figure 16 we present the runtimes for some additional experiments with $Q = Q_{\mathrm{C}}^{\otimes d}$ (as opposed to $Q = \frac{1}{2}Q_{\mathrm{N}}^{\otimes d} + \frac{1}{2}Q_{\mathrm{C}}^{\otimes d}$, since sparsity in projections is natural in high dimensional settings), $M = 5p$ (rather than $10p$) and $L$ varying from 10 to 200. We see that, while the performance improves as $L$ (and therefore the runtime) increases, we can typically outperform (or come very close to) all existing methods with relatively low computational resources (smaller $L$); the more computationally intensive default settings used in Table 7 are aimed at achieving near-optimal performance. Notably, when using larger values of $L$ in combination with $Q = Q_{\mathrm{C}}^{\otimes d}$ in these experiments, our RPE-based methods outperform drMARS in Models 5-7, whereas under the default settings in Tables 3 and 4 drMARS was slightly better.

## D  REAL DATA APPLICATION

In this section, we compare the performance of our methods with competing approaches using real-world data. We make use of the following four datasets from the UC Irvine Machine Learning Repository:

1. **Superconductivity**[3] (Hamidieh, 2018): This dataset consists of measurements on 81 physical and atomic covariates for 21,263 superconductors. The response variable is the critical temperature of the superconductor. We use all $p = 81$ covariates and randomly select a subsample of $n = 1000$ observations for training, and use the remainder for testing.

2. **Communities and Crime**[4] (Redmond & Baveja, 2002): This dataset contains 1,994 observations on 122 predictive variables relating to socioeconomic and police data. The response variable of interest is the per capita rate of violent crimes. For simplicity, we remove variables with missing values, leaving $p = 105$ predictive covariates in our experiment, and take a random sample of $n = 1329$ observations for training (the remainder for testing).

3. **Residential Building**[5] (Rafiei & Adeli, 2016): This dataset includes construction cost, sales prices, project variables, and economic variables corresponding to real estate single-family residential apartments in Tehran, Iran. There are 372 observations and 105 variables in total. We use all $p = 105$ predictive variables and take a random sample of $n = 248$ observations for training (the remainder for testing). There are two options for the response variable–sales price and construction cost–and we include both in our experiments.

4. **Geographical Origin of Music**[6] (Zhou et al., 2014): This dataset consists of 1,059 observations on 116 variables relating to audio features of a piece of music. We use the latitude of the piece's origin as the response variable. In this case, all $p = 116$ variables are used, and a subsample of $n = 706$ observations is taken for training (the remainder for testing).

In these experiments, since the true projection $A_0$ is unknown, we measure the performance of different techniques based on regression prediction accuracy after projecting the data. More precisely, we fixed the projection dimension $\hat{d}_0$ to either 3 or 10, and then apply the methods described in Section C.1 using only the training data. The interpretation is that each method selects its $\hat{d}_0$ most important directions. We then apply MARS[7] to the projected training data after each method. The performance is evaluated using the root mean squared error (RMSE) on the corresponding projected test set. The results are presented in Table 8, where our methods exhibit competitive performance across all four datasets considered here.

---

[3]https://archive.ics.uci.edu/dataset/464/superconductivty+data
[4]https://archive.ics.uci.edu/dataset/183/communities+and+crime
[5]https://archive.ics.uci.edu/dataset/437/residential+building+data+set
[6]https://archive.ics.uci.edu/dataset/315/geographical+original+of+music
[7]We use cross-validation to determine the degree, with the maximum order of interactions set to four.

| Dataset | No proj | $\hat{d}_0$ | Dimension reduction technique | | | | | | Our Methods | |
|---|---|---|---|---|---|---|---|---|---|---|
| | | | SIR | pHd | MAVE | DR | gKDR | drMARS | RPE | RPE2 |
| 1 | 17.4 | 3 | **16.5** | 30.2 | 19.3 | 29.9 | 18.1 | 18.8 | 18.5 | 17.0 |
| | | 10 | 17.3 | 27.8 | 18.0 | 22.2 | 17.9 | 20.4 | **17.0** | **17.0** |
| 2 | 0.180 | 3 | 0.127 | 0.212 | 0.140 | 0.184 | 0.132 | 0.138 | 0.134 | **0.125** |
| | | 10 | 0.137 | 0.187 | 0.135 | 0.136 | 0.132 | 0.140 | 0.134 | **0.126** |
| 3(sales) | 150 | 3 | 295 | 1198 | 218 | / | 386 | **158** | 175 | 175 |
| | | 10 | 295 | 699 | 295 | / | 429 | 211 | **125** | **125** |
| 3(cons.) | 32.1 | 3 | 70.4 | 203 | 47.6 | / | 161 | 136 | 80.0 | **27.6** |
| | | 10 | 68.4 | 301 | 42.4 | / | 47.1 | 29.6 | **24.0** | **24.0** |
| 4 | 25.7 | 3 | 16.9 | 18.1 | 16.7 | / | 17.7 | 17.0 | 16.8 | **15.8** |
| | | 10 | 17.9 | 17.5 | **15.8** | / | 17.0 | 19.4 | 16.2 | **15.8** |

Table 8: The RMSE on the test set for four real-data experiments. As a baseline, we also include the result of applying MARS directly to the original data (i.e., no projection is applied, denoted as "No proj"). For datasets 3 and 4, the DR method yielded very large RMSEs, so we omitted the results (denoted by "/").

## DECLARATION OF LLM USAGE

We used large language models solely to aid and polish writing (grammar, clarity, and phrasing) and code presentation (readability and consistency), without altering the logic. LLMs were not used for research ideation, retrieval, or discovery. The authors take full responsibility for the content of this paper.

