# OpenReview forum: "Random-projection ensemble dimension reduction"
_ICLR.cc/2026/Conference — ICLR 2026 Poster_

### Official Review · Reviewer_FpME · 2025-10-30

**Soundness:** 3
**Presentation:** 3
**Contribution:** 2
**Rating:** 6
**Confidence:** 3

**Summary:**

The paper proposes Random-Projection Ensemble Dimension Reduction (RPE), a flexible sufficient dimension reduction (SDR) framework. It samples many low-dimensional random projections, selects the best in each group by held-out regression performance, and aggregates selected projections via the average of outer products followed by SVD; the singular values also drive a data-dependent estimate of the target dimension (Algorithm 2) and a “double pass” refinement (Algorithm 3) when further compression is desired. The authors give a finite-sample bound that decays as L^(-1/2)in the number of groups (Theorem 1), extensive simulations (nine models), and real-data results claiming competitive or superior performance versus SIR, pHd, MAVE, DR, gKDR, and drMARS.

**Strengths:**

1. The method cleanly combines random projections, held-out selection, and SVD-based aggregation, making it plug-and-play with different projection families and base regressors while yielding a stable subspace estimator (with an optional double-pass refinement and a built-in dimension-selection procedure).
2. The finite-sample analysis isolates an “infinite-simulation” term and a sampling error that shrinks as L^(-1/2), clarifying how performance improves with the number of projection groups.
3. Extensive simulations spanning multiple p,n settings and diverse baselines, plus real-data evaluations, show competitive or superior accuracy; the algorithms and defaults are presented clearly enough to reproduce results.

**Weaknesses:**

1. The Contributions section is overly long and diffuses the main claims. The current text mixes method mechanics into the contributions. The Contributions should state only the new ideas.
2. The paper discusses how to choose L,M, and d, but provides no guidance for n_1 (the within-group training split used for projection selection).
3. Some arguments in the proof are confusing.

**Questions:**

1. Related work is missing several recent SDR directions that fit your setting. Such as BC (Huang, H. et.al. (2024)), KSDR (Liu, B. et.al. (2024)) and neural networks based SDR (Xu, S. et.al. (2025)).
2. The introduction foregrounds the high-dimensional p>n regime, but the main text only highlights a p<n case (e.g., n=200,p=50). Please surface at least one representative p>n setting (e.g., n=50,p=100) in the main paper—moving the results from the appendix if needed. Otherwise, the empirical narrative does not match the paper’s stated scope.
3. In the proof of Theorem 1, you state that {P_(l,*) }_(l=1)^Lare independent, but marginally they are not: each P_(l,*) is a data-dependent function of the same sample. If the intended claim is conditional independence given the observed dataset D, please state and use it explicitly. Moreover, all concentration steps should be written with P(⋅|D)and E[⋅|D].
4. The paper does not report the projection Frobenius distance d_F in unknown-d_0 settings; please include this results.
5. Please add a sensitivity study varying n_1/n∈{0.5,0.67,0.8}, since n_1 trades off selection variance versus validation reliability.

Reference:
1. Huang, H. H., Yu, F., & Zhang, T. (2024). Robust sufficient dimension reduction via α-distance covariance. Journal of Nonparametric Statistics, 1-16.
2. Liu, B., & Xue, L. (2024). Sparse kernel sufficient dimension reduction. Journal of Nonparametric Statistics, 1-24.
3. Xu, S., & Yu, Z. (2025, April). Neural Networks Perform Sufficient Dimension Reduction. In Proceedings of the AAAI Conference on Artificial Intelligence (Vol. 39, No. 20, pp. 21806-21814).

---

> ### Author Response · Authors · 2025-11-20
> **Author Responses - Part I**
>
> Thank you for taking the time to carefully review our work and for providing a detailed and concise summary of our main contributions. We have worked hard to provide a comprehensive, point-by-point response to your comments and questions below, and we have now posted an updated version of our paper. In the meantime, we warmly welcome any further comments or questions you may have.
>
> ### **Weaknesses**
>
> **1. The Contributions section is overly long and diffuses the main claims. The current text mixes method mechanics into the contributions. The Contributions should state only the new ideas.**
>
> This is a great suggestion suggestion -- thanks! In our revision, we have updated the introduction to better outline our main contributions. We now write
>
> **Our contributions**
>
> 1. We propose *random projection ensemble dimension reduction (RPEDR)* (Algorithm 1), a novel framework that is distinct from existing SDR methods. This is supplemented by two further algorithms: Algorithm 2 selects an appropriate projection dimension and Algorithm 3 provides a further refinement in some cases (see the diagram in Figure 2).
>
> 2. We systematically study the key tuning choices in our framework (e.g., projection distributions, number of random projections) and provide default recommendations.
>
> 3. We establish theoretical results for our proposal (Theorem 1 and Theorem 2), and demonstrate its strong performance through extensive simulated and real data experiments.
>
> We have also shortened the paragraph starting "Our proposed framework...", in particular removing the details of the method mechanics. This paragraph now reads:
>
> *Our proposed framework is highly flexible and allows user-specified choices of the random projection distribution and the base regression method, depending on the problem at hand -- these choices, along with other practical aspects, are investigated in Section 3 and Appendix A. In particular, Theorem 1 in Section 3.2 shows that the expected estimation error converges at a rate no slower than $L^{-1/2}$, where $L$ is the number of projection groups. This rate is observed in empirical investigations.*
>
> We hope you agree that this helps to distill our main contributions -- thank you again for the suggestion!
>
> **2. The paper discusses how to choose $L,M$, and $d$, but provides no guidance for $n\_1$ (the within-group training split used for projection selection).**
>
> Thanks for raising this point -- we have responded in detail to your related Question 5 below.
>
> **3. Some arguments in the proof are confusing.**
>
> We hope we have clarified this point in our response to your 3rd Question below. Please do let us know if there are any other points of confusion.

---

> ### Author Response · Authors · 2025-11-20
> **Author Responses - Part II**
>
> ### **Questions**
>
> **1. Related work is missing several recent SDR directions that fit your setting. Such as BC (Huang, H. et.al. (2024)), KSDR (Liu, B. et.al. (2024)) and neural networks based SDR (Xu, S. et.al. (2025)).**
>
> Thanks for making us aware of these works, which we now cite in the first paragraph of related work section -- we apologise for omitting them from our original submission.
>
> **2. The introduction foregrounds the high-dimensional $p>n$ regime, but the main text only highlights a $p<n$ case (e.g., $n=200,p=50$). Please surface at least one representative $p>n$ setting (e.g., $n=50,p=100$) in the main paper moving the results from the appendix if needed. Otherwise, the empirical narrative does not match the paper’s stated scope.**
>
> Thanks for raising this point and for encouraging us to further explore cases where $p > n$. We are pleased to report that we've conducted several further simulation experiments with $p \in \lbrace200, 500\rbrace$ and $n \in \lbrace50, 200\rbrace$,  and our RPE-based method continues to show superior performance over competitors in these more challenging settings; see the summary table in Response Part III below. In the revised manuscript, in addition to extending Tables 2, 3 and 4 to include these results, based on your recommendation, we have included boxplots of the results for $n=200$ and $p = 500$ in the main text -- see Figure 7.
>
> **3. In the proof of Theorem 1, you state that $\lbrace P\_{l, \ast}\rbrace \_{l=1}^L$ are independent, but marginally they are not: each $P\_{l, \ast}$ is a data-dependent function of the same sample. If the intended claim is conditional independence given the observed dataset $D$, please state and use it explicitly. Moreover, all concentration steps should be written with $P(\cdot|D)$and $E[\cdot|D]$.**
>
> You are correct that the chosen projections depend on the data and are therefore not marginally independent when the training data is treated as random. In Theorem 1, however, the the training data are considered fixed, so the only randomness comes from the random projections, under which the $\lbrace P\_{l, \ast}\rbrace \_{l=1}^L$ are independent. We now recall this fact explicitly at the beginning of the proof to avoid confusion.
>
> **4. The paper does not report the projection Frobenius distance $d\_F$ in unknown-$d\_0$ settings; please include this results.**
>
> Thanks for this suggestion. We didn't report the measure $d\_{\mathrm{F}}$ in this case because we have
> $$
> 2d\_{\mathrm{F}}^2(\hat{A}\_0, A\_0) = d\_{\mathrm{FP}}^2(\hat{A}\_0, A\_0) + d\_{\mathrm{FN}}^2(\hat{A}\_0, A\_0).  \qquad (1)
> $$
>
> We feel that separately reporting $d\_{\mathrm{FP}}$ and $d\_{\mathrm{FN}}$ provides a clearer picture of the tradeoffs between the false positives and false negatives that the different methods incur when $d\_0$ is unknown, though we would be happy to report $d\_\mathrm{F}$ as well if you feel it would add to the current results.
>
> To see (1), write
> \begin{align*}
> 2d\_{\mathrm{F}}^2(\hat{A}\_0, A\_0)
> &= \bigl|| \hat{A}\_0 \hat{A}\_0^\top - A\_0 A\_0^\top \bigr||\_F^2 \\\\
> &= \text{tr} \bigl\lbrace (\hat{A}\_0 \hat{A}\_0^\top - A\_0 A_0^\top)^\top (\hat{A}\_0 \hat{A}\_0^\top - A\_0 A\_0^\top) \bigr\rbrace \\\\
> &= \text{tr} \bigl\lbrace \hat{A}\_0 (\hat{A}\_0^\top \hat{A}\_0) \hat{A}\_0^\top - \hat{A}\_0 \hat{A}\_0^\top A\_0 A\_0^\top - A\_0 A\_0^\top\hat{A}\_0 \hat{A}\_0^\top + A\_0 (A\_0^\top A\_0) A\_0^\top \bigr\rbrace \\\\
> &= \text{tr} \bigl\lbrace \hat{A}\_0^\top \hat{A}\_0 \bigr\rbrace + \text{tr}\bigl\lbrace A\_0^\top A\_0 \bigr\rbrace - 2\text{tr} \bigl\lbrace A\_0 A\_0^\top\hat{A}\_0 \hat{A}\_0^\top \bigr\rbrace \\\\
> &= \hat{d}\_0 + d\_0 - 2\text{tr} \bigl\lbrace A\_0 A\_0^\top\hat{A}\_0 \hat{A}\_0^\top \bigr\rbrace.
> \end{align*}
>
> On the other hand,
> \begin{align*}
> d\_{\text{FP}}^2 (\hat{A}\_0, A\_0)
> &= \bigl || (I - A\_0 A\_0^\top) \hat{A}\_0 \bigr ||\_F^2 \\\\
> &= \text{tr} \bigl\lbrace \hat{A}\_0^\top (I - A\_0 A\_0^\top)^2 \hat{A}\_0 \bigr\rbrace \\\\
> &= \text{tr} \bigl\lbrace \hat{A}\_0^\top (I - A\_0 A\_0^\top) \hat{A}\_0 \bigr\rbrace \\\\
> &= \text{tr} \bigl\lbrace \hat{A}\_0^\top \hat{A}\_0 \bigr\rbrace - \text{tr} \bigl\lbrace \hat{A}\_0^\top A\_0 A\_0^\top \hat{A}\_0 \bigr\rbrace \\\\
> &= \hat{d}\_0 - \text{tr}\bigl\lbrace A\_0 A\_0^\top\hat{A}\_0 \hat{A}\_0^\top \bigr\rbrace,
> \end{align*}
> similarly, we can derive
> $$
> d\_{\text{FN}}^2 = d\_0 - \text{tr}\bigl\lbrace A\_0 A\_0^\top\hat{A}\_0 \hat{A}\_0^\top\bigr\rbrace.
> $$
> The above three equations complete our claim.
>
> **5. Please add a sensitivity study varying $n_1/n \in \lbrace0.5,0.67,0.8\rbrace$, since $n_1$ trades off selection variance versus validation reliability.**
>
> Thank you for this suggestion, which we have taken on board. In our revision, we have included additional experiments in a new Appendix A.3. As you can see, the performance of our algorithm is relatively stable as $n\_1/n$ varies across the range you suggested.

---

> ### Author Response · Authors · 2025-11-20
> **Author Responses - Part III**
>
> For your convenience, we also report the new simulation results as follows (large $p$ settings):
> | Model | $p$ | $n$ | MAVE         | gKDR         | drMARS            | RPE              | RPE2             |
> |-------|-----|-----|-----------------|----------------|-----------------------|---------------------|---------------------|
> | 1     | 200 | 50  | $0.97_{0.06}$   | $0.98_{0.02}$  | $\mathbf{0.31_{0.27}}$ | $0.64_{0.19}$       | $\mathbf{0.35_{0.30}}$ |
> | 1     | 200 | 200 | $0.97_{0.04}$   | $0.99_{0.01}$  | $0.23_{0.21}$         | $0.55_{0.16}$       | $\mathbf{0.06_{0.02}}$ |
> | 1     | 500 | 50  | $0.98_{0.05}$   | $0.99_{0.01}$  | $\mathbf{0.34_{0.31}}$ | $0.83_{0.18}$       | $0.78_{0.26}$       |
> | 1     | 500 | 200 | $0.98_{0.03}$   | $0.99_{0.01}$  | $0.22_{0.19}$         | $0.61_{0.15}$       | $\mathbf{0.08_{0.03}}$ |
> | 2     | 200 | 50  | $0.96_{0.05}$   | $1.00_{0.00}$  | $0.75_{0.31}$         | $\mathbf{0.26_{0.37}}$ | $\mathbf{0.30_{0.38}}$ |
> | 2     | 200 | 200 | $0.92_{0.04}$   | $0.98_{0.02}$  | $0.27_{0.19}$         | $\mathbf{0.01_{0.00}}$ | $\mathbf{0.01_{0.00}}$ |
> | 2     | 500 | 50  | $0.97_{0.04}$   | $1.00_{0.00}$  | $0.85_{0.24}$         | $\mathbf{0.33_{0.42}}$ | $\mathbf{0.34_{0.42}}$ |
> | 2     | 500 | 200 | $0.95_{0.03}$   | $0.99_{0.01}$  | $0.33_{0.25}$         | $\mathbf{0.01_{0.00}}$ | $\mathbf{0.01_{0.00}}$ |
> | 3     | 200 | 50  | $1.19_{0.09}$   | $1.39_{0.01}$  | $1.21_{0.18}$         | $\mathbf{0.83_{0.32}}$ | $\mathbf{0.83_{0.32}}$ |
> | 3     | 200 | 200 | $1.13_{0.07}$   | $1.32_{0.03}$  | $1.11_{0.19}$         | $\mathbf{0.13_{0.09}}$ | $\mathbf{0.13_{0.09}}$ |
> | 3     | 500 | 50  | $1.22_{0.09}$   | $1.41_{0.00}$  | $1.21_{0.18}$         | $\mathbf{0.94_{0.23}}$ | $\mathbf{0.93_{0.23}}$ |
> | 3     | 500 | 200 | $1.16_{0.07}$   | $1.38_{0.01}$  | $1.15_{0.19}$         | $\mathbf{0.25_{0.22}}$ | $\mathbf{0.25_{0.22}}$ |
> | 4     | 200 | 50  | $\mathbf{0.69_{0.16}}$ | $0.98_{0.01}$ | $0.84_{0.20}$ | $0.80_{0.09}$       | $0.78_{0.17}$       |
> | 4     | 200 | 200 | $0.48_{0.06}$   | $0.82_{0.06}$  | $0.85_{0.16}$         | $0.74_{0.08}$       | $\mathbf{0.17_{0.15}}$ |
> | 4     | 500 | 50  | $\mathbf{0.77_{0.16}}$ | $0.99_{0.00}$ | $0.85_{0.15}$ | $0.84_{0.08}$       | $0.85_{0.11}$       |
> | 4     | 500 | 200 | $0.54_{0.06}$   | $0.94_{0.04}$  | $0.86_{0.15}$         | $0.79_{0.04}$       | $\mathbf{0.25_{0.24}}$ |
> | 5     | 200 | 50  | $1.29_{0.07}$   | $1.39_{0.01}$  | $1.29_{0.10}$         | $\mathbf{1.21_{0.12}}$ | $\mathbf{1.21_{0.12}}$ |
> | 5     | 200 | 200 | $1.20_{0.04}$   | $1.29_{0.03}$  | $1.30_{0.13}$         | $\mathbf{1.00_{0.00}}$ | $\mathbf{1.00_{0.00}}$ |
> | 5     | 500 | 50  | $1.31_{0.08}$   | $1.41_{0.00}$  | $1.27_{0.14}$         | $\mathbf{1.23_{0.13}}$ | $\mathbf{1.23_{0.13}}$ |
> | 5     | 500 | 200 | $1.24_{0.04}$   | $1.36_{0.01}$  | $1.30_{0.11}$         | $\mathbf{1.00_{0.00}}$ | $\mathbf{1.00_{0.00}}$ |
> | 6     | 200 | 50  | $1.57_{0.10}$   | $1.71_{0.01}$  | $\mathbf{1.48_{0.19}}$ | $1.52_{0.19}$       | $1.52_{0.19}$       |
> | 6     | 200 | 200 | $1.59_{0.06}$   | $1.71_{0.01}$  | $1.57_{0.19}$         | $\mathbf{1.39_{0.28}}$ | $\mathbf{1.39_{0.28}}$ |
> | 6     | 500 | 50  | $1.64_{0.09}$   | $1.72_{0.00}$  | $\mathbf{1.51_{0.17}}$ | $1.61_{0.16}$       | $1.61_{0.16}$       |
> | 6     | 500 | 200 | $1.60_{0.06}$   | $1.72_{0.00}$  | $1.57_{0.20}$         | $\mathbf{1.50_{0.19}}$ | $\mathbf{1.50_{0.19}}$ |
> | 7     | 200 | 50  | $1.38_{0.06}$   | $1.40_{0.01}$  | $\mathbf{1.27_{0.17}}$ | $1.33_{0.14}$       | $1.33_{0.14}$       |
> | 7     | 200 | 200 | $1.34_{0.06}$   | $1.40_{0.01}$  | $1.18_{0.21}$         | $\mathbf{1.00_{0.24}}$ | $\mathbf{0.99_{0.25}}$ |
> | 7     | 500 | 50  | $1.39_{0.06}$   | $1.41_{0.00}$  | $\mathbf{1.23_{0.18}}$ | $1.35_{0.12}$       | $1.35_{0.12}$       |
> | 7     | 500 | 200 | $1.36_{0.04}$   | $1.41_{0.00}$  | $1.17_{0.20}$         | $\mathbf{1.03_{0.26}}$ | $\mathbf{1.03_{0.26}}$ |
> | 8     | 200 | 50  | $1.18_{0.11}$   | $1.39_{0.01}$  | $0.79_{0.31}$         | $\mathbf{0.54_{0.38}}$ | $\mathbf{0.54_{0.38}}$ |
> | 8     | 200 | 200 | $1.08_{0.06}$   | $1.37_{0.02}$  | $0.88_{0.21}$         | $\mathbf{0.10_{0.02}}$ | $\mathbf{0.10_{0.02}}$ |
> | 8     | 500 | 50  | $1.21_{0.09}$   | $1.41_{0.00}$  | $0.83_{0.30}$         | $\mathbf{0.74_{0.35}}$ | $\mathbf{0.74_{0.35}}$ |
> | 8     | 500 | 200 | $1.11_{0.07}$   | $1.40_{0.01}$  | $0.89_{0.19}$         | $\mathbf{0.18_{0.14}}$ | $\mathbf{0.18_{0.14}}$ |
> | 9     | 200 | 50  | $1.30_{0.07}$   | $1.39_{0.01}$  | $1.22_{0.22}$         | $\mathbf{1.09_{0.16}}$ | $\mathbf{1.09_{0.15}}$ |
> | 9     | 200 | 200 | $1.26_{0.05}$   | $1.35_{0.02}$  | $1.25_{0.20}$         | $\mathbf{0.34_{0.39}}$ | $\mathbf{0.34_{0.39}}$ |
> | 9     | 500 | 50  | $1.32_{0.07}$   | $1.41_{0.00}$  | $1.24_{0.22}$         | $\mathbf{1.13_{0.20}}$ | $\mathbf{1.13_{0.20}}$ |
> | 9     | 500 | 200 | $1.28_{0.05}$   | $1.39_{0.01}$  | $1.22_{0.25}$         | $\mathbf{0.28_{0.37}}$ | $\mathbf{0.28_{0.37}}$ |

---

> ### Author Response · Authors · 2025-11-28
> **Author Responses – Thanks!**
>
> Dear Reviewer FpME,
>
> We would like to sincerely thank you again for the time and effort you invested in reviewing our paper, and for the many constructive and detailed suggestions you provided. We carefully considered all of your comments and have incorporated them into our revised manuscript where possible.
>
> We are very grateful for your thoughtful feedback, which has helped us to significantly improve the quality and clarity of the work.

---

### Official Review · Reviewer_MBR7 · 2025-11-01

**Soundness:** 3
**Presentation:** 3
**Contribution:** 2
**Rating:** 4
**Confidence:** 3

**Summary:**

The main goal of this paper is to accurately estimate the central mean subspace $S(A_0).$ To achieve that, paper develops random-projection ensemble dimension reduction (RPEDR) framework for high-dimensional regression problems. RPEDR first generates independent random projections and selects the optimal projection based on mean-squared error in each group. After averaging outer products of selected projection matrices to get $\hat{\Pi}$, final estimated dimension reduction directions and relative importance of each direction are obtained from SVD of $\hat{\Pi}$. The paper then derives the expected estimation error
bound of proposed method to show the effects of multiple parameters. Finally, extensive empirical results present the effectiveness of new method.

**Strengths:**

1. The proposed method is parallelizable, making it applicable to distributed systems.
2. The paper not just shows empirical results but also provides theoretical bound for estimation error.

**Weaknesses:**

1. The superiority of proposed method is not so promising. According to experiment results in Table 2-7, drMARS sometimes shows better performances and needs less runtime. On the other hand, proposed RPE2 and RPE are not always optimal and consume much more runtime. This weakens the contribution of this work.
2. This framework has multiple parameters, including L, M and d, and each needs special tuning, which makes this framework less practical.
3. The computation cost of proposed method may bring concerns because of the long runtime.
4. Considering the main goal of this work is dimension reduction, the reviews about this topic is not clear enough to show existing methods and their relationships with proposed method.
5. Lack of comparison of theoretical bounds. Since paper derives the error bound, it's necessary to compare with existing bounds.

**Questions:**

As the main assumption in this work, the existence and uniqueness of the CMS, a more detailed description about it is needed, like the exact conditions about this assumption.

---

> ### Author Response · Authors · 2025-11-20
> **Author Responses - Part I**
>
> Thank you very much for taking the time to carefully review our submission, and for accurately outlining our main contributions.  Below we provide point-by-point responses to your comments and questions.  We have worked hard to provide a comprehensive initial response and have now posted an updated draft of our paper. In the meantime, we welcome any further comments or questions you may have.
>
> ### **Weaknesses**
> **1. The superiority of proposed method is not so promising. According to experiment results in Table 2-7, drMARS sometimes shows better performances and needs less runtime. On the other hand, proposed RPE2 and RPE are not always optimal and consume much more runtime. This weakens the contribution of this work.**
>
> Thank you for raising this point. As you highlight, our method is not *always* optimal across all scenarios, and drMARS sometimes performs slightly better, in particular in some settings in models 5-7. However, we'd like to stress that models 3, 5, 6, 7 and 8 are taken from the drMARS paper, and we used these to provide as fair as possible an empirical comparison with drMARS as well as the other methods (Models 3 and 8 are also used in many other existing works). There are settings, in particular models 2, 3, 8 and 9, where we perform much better than any other methods, including drMARS.
>
> Another strength of our method can be seen in its robustness across all settings we consider, in the sense that RPE (and RPE2) never `fails' to find any signal, while all other competitors suffer from finding only trivial signal is some cases -- e.g., drMARS fails to detect any signal across most settings in Model 2.
>
> Moreover, based on the other reviewers' suggestions, we have included additional simulation experiments with $p \in \lbrace 200, 500\rbrace$ and $n \in \lbrace50, 200\rbrace$, specifically to target cases where $p \geq n$. In these new experiments our proposals perform very well! See the new Figure 7 and the additional results in Tables 2-4.
>
> Regarding computational cost, as you highlight, our method is typically slower than the competitors, as we show in Appendix C.3. We recognise the importance of the trade-offs between computational complexity and predictive performance, but in much of our paper we focus on the latter, as we see our main contribution as introducing a new and flexible framework.  There are many ways in which we can reduce the computational cost of our approach if that is a concern -- see our detailed response to your third point below.
>
> **2. This framework has multiple parameters, including L, M and d, and each needs special tuning, which makes this framework less practical.**
>
> We worked hard in our paper to fully investigate the effect of the choices of these parameters, and to provide practical default recommendations. Indeed, we suggest setting $L = 200$, $M = 10p$ and $d = \lceil \sqrt{p} \rceil$. As is shown in Appendix A.2, our Algorithm exhibits relatively stable performance with small changes to these parameters. As discussed in response to your other points, in some cases we can achieve similar performance with smaller $L$, $M$ and $d$, which can significantly reduce the computational cost without compromising the predictive performance.
>
> **3. The computation cost of proposed method may bring concerns because of the long runtime.**
>
> Thank you for raising this important point. If computational resources are limited, there are many ways in which we can improve the computational cost of the recommended default version of our method. For instance, as discussed in Section A.3 using a simpler (e.g linear) base regression method significantly reduces the computational cost, though this may come at the expense of poorer performance in cases where there is no linear true signal (e.g. as is the case in models 1 and 3 in Section A.1). For these models, a quadratic least squares method is effective (see Figure 7) at modest computational cost.  Other ways to reduce the overall computational cost include taking $L$ and/or $M$ to be smaller -- see the new Figure 16 in Appendix C.3, which shows that in many cases we can obtain competitive results with the existing methods with modest computational resources, even in a relatively `large-scale' setting of $n=500$, $p=100$. Notably, RPE-based methods also show better performance than drMARS for model 5-7 when we set a large $L$, different from the results given by Table 3-4 when we just used the default settings, but further illustrating the strength of our method. In fact, the plots in Figure 16 were produced by setting $M=5p$, and set a grid for different values of $L$. We also now include further discussion of these points in Section C.3 of the revised manuscript.

---

> ### Author Response · Authors · 2025-11-20
> **Author Responses - Part II**
>
> **4. Considering the main goal of this work is dimension reduction, the reviews about this topic is not clear enough to show existing methods and their relationships with proposed method.**
>
> Thank you for raising this point.  First, we have expanded the related work section to include additional existing works based on the suggestion of one of the other reviewers.
>
> Regarding the broader SDR literature and the relationship between our method and existing SDR approaches, we view our proposal as a completely new way to tackle the problem, and a direct one-to-one comparison with existing approaches is not always possible.
>
> We do not feel it is appropriate for us to provide a detailed survey of the existing works, due to space constraints, and instead we have provided references to survey papers on the topic. However, if you feel there are particular details missing, or that specific related works should be added to our references, we would very much welcome concrete suggestions on what to include.
>
> **5. Lack of comparison of theoretical bounds. Since paper derives the error bound, it's necessary to compare with existing bounds.**
>
> Thanks for this point. We feel that our Theorem 1 is somewhat different to existing error bounds in the SDR literature. To clarify the distinction, we note that our result treats the training data as fixed (or equivalently, it can be viewed as being conditional on the training data), and provides a bound on the expected estimation error with respect to the randomness in the random projections in Algorithm 1.  Note that our result holds under no assumptions on the random projection distribution on $\mathcal{Q}_{p\times d}$, or on the configuration of the training data.
>
> In contrast, existing bounds in the literature are probabilisitic / asymptotic bounds with respect to the randomness of the data generating process. Such bounds typically use different error metrics from ours, are sometimes asymptotic ($n \rightarrow \infty$), and depend on method-specific tuning parameters and assumptions on the covariate distribution. For instance, drMARS gives a bound on the projection matrix $B$ (Liu et al., 2023, Theorem 4): there exists a $d \times d$ rotation matrix $Q$ such that $\bigl \|\tilde{B} - BQ \bigr\| = O_P(m_*^{1/2}n^{-1/2}+ \tilde{p}(m_*))$, where the rate depends on the number of spline basis functions $m_*$ selected, and an approximation bias term $\tilde{p}(\cdot)$. Similarly, gKDR establishes consistency rates for an RKHS-based matrix $M_n(x)$, whose eigenspace can be seen as an estimation of the CMS. The bounds in (Fukumizu & Leng, 2014, Theorem 3-4) are expressed in terms of the kernel-dependent smoothness constants $L_m$ and $\alpha_m$, and depends on a carefully chosen regularisation $\epsilon_n$. These rates are not comparable with ours in terms of the number of projection groups $L$.
>
> Since the randomness, error metrics and tuning parameters are different, we do not view a direct comparison of these bounds as particularly informative for our setting. We hope this clarification helps to address your concern.
>
> References:
>
> 1. Kenji Fukumizu and Chenlei Leng. Gradient-based kernel dimension reduction for regression. *Journal of the American Statistical Association*, 109(505):359–370, 2014.
> 2. Yu Liu, Degui Li, and Yingcun Xia. Dimension reduction and MARS. *Journal of Machine Learning Research*, 24(309):1–30, 2023.

---

> ### Author Response · Authors · 2025-11-20
> **Author responses - Part III**
>
> ### **Questions**
> **As the main assumption in this work, the existence and uniqueness of the CMS, a more detailed description about it is needed, like the exact conditions about this assumption.**
>
> Thanks for this point. We believe this assumption is mild. Indeed, it essentially amounts to an identifiability condition that ensures we have a well-defined target of estimation, and it is widely used in the sufficient dimension reduction literature. One simple sufficient condition is that the domain of the marginal distribution of the covariates $X$ is open and convex -- see Cook and Li (2002, Definition 2 and the discussion afterwards). Further, for a detailed description of when we can ensure the existence and uniqueness of the CMS, see Cook (1996, Lemma 1) and Chiaromonte \& Cook, (1997, Section 2.1).
>
> To provide some intuition, a well-known example where there does **not** exist a unique CMS is the following from Cook (1998):  Let $X = (X\_1,X\_2)$ be distributed according to Haar measure (i.e. uniformly distributed) on the unit circle $\lbrace x = (x\_1,x\_2) \in \mathbb{R}^2: ||x|| = \sqrt{x\_1^2 + x\_2^2} = 1\rbrace$.  Then suppose the "true" regression model is
> $$
> Y = X\_1^2 + \epsilon
> $$
> where $\epsilon$ is independent of $X$, with mean $0$. In this case, the space spanned by the columns of $A\_0 = (1,0)^\top$ is a (mean) dimension reduction subspace, since $Y \perp X | X\_1$.  However, since $X\_2^2 = 1 - X\_1^2$ almost surely on the unit circle, we can also write
> $$
> Y = 1 - X\_2^2 + \epsilon,
> $$
> so the space spanned by the columns of $A\_0' = (0,1)^T$ is also a (mean) dimension reduction subspace. A general definition of the central (mean) subspace is the intersection of all (mean) dimension reduction subspaces. In this example, that intersection is the trivial subspace $\lbrace 0 \rbrace$, which is **not** a mean dimension reduction subspace, as it would require $\mathbb{E} [Y | X]$ to be constant. Therefore, a non-trivial CMS does not exist in this example. This also gives some intuition for why conditions such as an open, convex support for the marginal distribution of $X$ ensure existence / uniqueness of the CMS.
>
> In our revised paper, we now write:
>
> *For the purposes of this paper, we assume the existence and uniqueness of the CMS. This assumption is standard and mild (see, for example, Cook, 1996, Lemma 1), and allows us to focus on our main goal of estimating the space $\mathcal{S}(A\_0)$ based on a dataset of $n$ independent and identically distributed pairs $(X\_1, Y\_1), \ldots, (X\_n, Y\_n) \sim P$.*
>
> to direct the reader to explicit conditions under which this assumption holds. Note that the condition stated there guarantees the existence and uniqueness of the *central subspace (CS)*, and the existence and uniqueness of the CMS can be guaranteed in essentially the same way (see Cook and Li, 2002). We hope this helps to clarify your concern.
>
> References:
>
> 1. R Dennis Cook. Graphics for regressions with a binary response. *Journal of the American Statistical Association*, 91(435):983–992, 1996.
>
> 2. Francesca Chiaromonte and R Dennis Cook. On Foundations of Regression Graphics. Technical Report 616, University of Minnesota, School of Statistics, 1997.
>
> 3. R Dennis Cook and Bing Li. Dimension reduction for conditional mean in regression. *The Annals of Statistics*, 30(2):455–474, 2002.

---

> ### Author Response · Authors · 2025-11-28
> **Author Responses – Thanks!**
>
> Dear Reviewer MBR7,
>
> We would like to sincerely thank you again for the time and effort you invested in reviewing our paper, and for the many constructive and detailed suggestions you provided. We carefully considered all of your comments and have incorporated them into our revised manuscript where possible.
>
> We are very grateful for your thoughtful feedback, which has helped us to significantly improve the quality and clarity of the work.

---

### Official Review · Reviewer_8gwY · 2025-11-01

**Soundness:** 3
**Presentation:** 3
**Contribution:** 3
**Rating:** 6
**Confidence:** 2

**Summary:**

The paper introduces Random Projection Ensemble Dimension Reduction (RPEDR). The method projects data onto multiple low-dimensional random subspaces, evaluates each projection within its group using a base regression model, selects the best projection in each group, and aggregates the winners via SVD. This procedure adaptively identifies informative directions, yielding stable estimates at modest computational cost. The authors also study design choices, projection distributions and the number of projections, and provide theory showing that the expected estimation error decreases monotonically with the number of projection groups.

**Strengths:**

* Proposes a novel, flexible, and theoretically principled framework for dimension reduction in high-dimensional regression via random-projection ensembles.
* Adaptively selects informative projection directions, delivering more stable estimates.
* Provides theoretical guarantees showing that the expected estimation error decreases monotonically as the number of projection groups increases.

**Weaknesses:**

I am not deeply familiar with the SDR literature; my expertise lies more in random-projection methods for computational efficiency. Consequently, I cannot fully determine whether the comparisons with existing SDR approaches are sufficiently comprehensive or representative of the current state of the art. The following comments are therefore offered as provisional observations:

* Theorem 1 relies on the assumption $L = \infty$, which limits its practical value for guiding the selection of $L$ in real applications. Could similar theoretical guarantees be derived for finite $L$?

* The introduction frames the problem as high-dimensional ($n<p$), yet Appendices analyzes parameter choices primarily for $n>p$. This inconsistency weakens empirical support for the stated high-dimensional motivation. Additional experiments or discussion for the $n<p$ regime would improve credibility.

* Algorithm 3 is described as an enhanced version of Algorithm 1, but it is not clear whether Theorem 1, or a corresponding variant, extends to this setting. Clarifying the theoretical validity of the two-stage procedure would strengthen the methodological contribution.

* Algorithm 3 fixes $L = 200$ without justification. Is this choice empirically optimal, or merely heuristic? How sensitive is performance to $L$ across different $n, p, d$?

* While the paper repeatedly emphasizes the flexibility and efficiency of the proposed method, it lacks an explicit characterization of computational complexity,
The authors are encouraged to clearly specify the time and memory complexity under typical parameter settings and provide a comparison with established methods like MAVE and drMARS to better illustrate the computational advantages and trade-offs of the proposed approach.

* Although Appendix D mentions real-data experiments, the main text presents only simulation results. It is recommended to include at least one illustrative real-data application n the main paper to enhance the work’s practical relevance and overall impact.

**Questions:**

1. In Algorithm 2 (line 6), how is the threshold $T_{\ell} > 1/2$ chosen? Could alternative thresholds (other than $1/2$) be used?

2. In Algorithm 3, $L = 200$ is used while many experiments set $n = 200$, which may suggest nontrivial computational cost. Can a smaller $L$ be employed without materially degrading performance? If so, is there guidance or a sensitivity analysis for selecting $L$ as a function of (n,p,d)?

3. Beyond Gaussian and Cauchy projections, are other projection families, e.g., sub-Gaussian or Hadamard-structured transforms, compatible with the framework?

4. I recommend adding a  framework diagram to illustrate the workflow and the relationships among Algorithms 1–3.

---

> ### Author Response · Authors · 2025-11-20
> **Author Responses - Part I**
>
> Thank you very much for taking the time to carefully review our submission — we are glad that you found our proposal novel and recognised the strengths of our work. Below we provide detailed responses to the constructive points you raised about the paper. We have now uploaded an updated draft of the paper. In the meantime, we warmly welcome any further comments or questions you may have.
>
> ### **Weaknesses**
>
> **1. Theorem 1 relies on the assumption $L = \infty$, which limits its practical value for guiding the selection of  in real applications. Could similar theoretical guarantees be derived for finite $L$?**
>
> Thank you for raising this point, though there appears to be some confusion.  Theorem 1 does in fact provide guarantees for finite $L$, and it does not assume $L=\infty$.  The result provides a bound on the expected sin theta distance between the output of Algorithm 1 and the true projection $A_0$, which holds for any finite $L$. The quantity corresponding to $L=\infty$ in the statement is $\Pi^{\infty}$, which is the expected value of $\hat{\Pi}$, or the infinite-simulation limit of $\Pi$, so the theorem can be viewed as bounding the difference between what one obtains with a fixed, finite $L$, and this idealised $L=\infty$ case.
>
> **2. The introduction frames the problem as high-dimensional ($n < p$), yet Appendices analyzes parameter choices primarily for $n > p$ . This inconsistency weakens empirical support for the stated high-dimensional motivation. Additional experiments or discussion for the $n <p$ regime would improve credibility.**
>
> Thanks for raising this point and for encouraging us to further explore cases where $p > n$. We are pleased to report that we've conducted several further simulation experiments with $p \in \lbrace200, 500\rbrace$ and $n \in \lbrace50, 200\rbrace$, and that our proposals remain superior in these more challenging regimes, while using the same default settings suggested in Section 3 and A. In fact, our method shows even more pronounced gains over competitors in these settings! See the summary table in Response Part III. In the revised manuscript, in addition to extending Tables 2, 3 and 4 to include these results, based on the recommendation of reviewer FpME, we have included boxplots of the results for $n=200$ and $p = 500$ in the main text -- see Figure 7. We hope these help to address your concern.
>
> **3. Algorithm 3 is described as an enhanced version of Algorithm 1, but it is not clear whether Theorem 1, or a corresponding variant, extends to this setting. Clarifying the theoretical validity of the two-stage procedure would strengthen the methodological contribution.**
>
> Thank you for this suggestion -- we're pleased to report that we have managed to prove a new result about the output of Algorithm 3.  This included in Appendix A.5 as follows:
>
> We now provide an extension of Theorem 1, which provides guarantees for the output of Algorithm 3. To set the scene for this new result suppose we input $\check{d}_0 = d_0$ in Algorithm 3, and that Algorithm 2 suggests we should set $\hat{d}_0 > d_0$, then lines 5-7 of Algorithm 3 apply. Let $A_0^{\ast} \in \mathcal{A}\_{\hat{d}\_0 \times d\_0}$ be the $Q$ matrix in a QR decomposition of $\hat{A}_0^{\top} A_0$. Intuitively, $A_0^{\ast}$ captures exactly the part of the true CMS $\mathcal{S}(A_0)$ that is preserved inside $\mathcal{S}(\hat A_0)$ after the first-stage projection. As in Theorem 1, we treat the $\{ (z_1, y_1), \ldots, (z_n, y_n)\}$ as fixed pairs in $\mathbb{R}^{\hat{d}_0} \times \mathbb{R}$, From Algorithm 3, we will obtain $\check{U} = \check{U}\_{1:\hat{d}\_0} = (\check{U}\_1, \ldots, \check{U}\_{\hat{d}_0})$ and $\check{D}$. Algorithm 3 then outputs $\check{A}\_0 = \hat{A}\_0 (\check{U}\_1, \ldots, \check{U}\_{d\_0}) = \hat{A}\_0 \check{U}\_{1:d\_0}$.  Let $\Pi \_{\ast} \^{\infty} := \mathbb{E}(\check{U} \check{D} \check{U}^\top \mid \hat{A}\_0) \in \mathbb{S}\_{\hat{d}\_0 \times \hat{d}\_0}$.
>
> Theorem 2 bounds the expected error of the output of Algorithm 3 conditionally on the initial output from the first application of Algorithm 1 and 2.  The bound includes the *false negative* error arising from the first application, defined as
> $$
> d\_{\mathrm{FN}}(\hat{A}\_0, A\_0) := \bigl || (I - \hat{A}\_0 \hat{A}\_0^{\top})A\_0 \bigr ||_F.
> $$
>
> $d\_{\mathrm{FN}}$ measures false negatives and quantifies the amount of the space spanned by $A\_0$ that is missed by the projection $\hat{A}\_0$. Indeed, we have $d\_{\mathrm{FN}}(\hat{A}\_0, A\_0) = 0$ when $\mathcal{S}(A\_0) \subseteq \mathcal{S}(\hat{A}\_0)$. This should be interpreted as quantifying the amount of the true signal missed by the first application of Algorithm 1.  The other terms in the bound in Theorem 2 should be interpreted similarly to the bounds appearing in Theorem 1, note however that, since the dimension has been reduced from $p$ to $\hat{d}\_0$ in the first application, the final term in the bound no longer depends on the ambient dimension $p$.

---

> ### Author Response · Authors · 2025-11-20
> **Author Responses - Part II**
>
> **Theorem 2**
>
> We have that
> $$
> \mathbb{E} \bigl[d\_{\text{F}}\bigl(\mathcal{S}(\check{A}\_0),  \mathcal{S}(A\_0)\bigr) \bigm| \hat{A}\_0 \bigr ] \leq d\_{\text{FN}} (\hat{A}\_0, A\_0) + 2d\_0^{1/2} \bigl || \Pi\_{\ast}^{\infty} - A\_0^{\ast}(A\_0^*)^{\top} \bigr ||\_{\mathrm{op}} +  \frac{2 (2\pi)^{1/2} d\_0^{1/2} d \cdot \hat{d}\_0}{L^{1/2}}.
> $$
> The proof of Theorem 2 is given in Appendix B.
>
> **Proof of Theorem 2**
>
> First, by triangle inequality we have
> $$
> d\_{\text{F}}\bigl(\mathcal{S}(\check{A}_0), \mathcal{S}({A}\_0)\bigr) \leq d\_{\text{F}} \bigl( \mathcal{S}(\hat{A}\_0 \check{U}\_{1:d\_0}), \mathcal{S}(\hat{A}\_0 A\_0^\{\ast}) \bigr) + d\_{\text{F}}(\mathcal{S}(\hat{A}\_0 A\_0^{\ast}), \mathcal{S}(A\_0) \bigr). \qquad (1)
> $$
>
> Now for any $B \in \mathbb{R}^{\hat{d}\_0 \times \hat{d}\_0}$, we have
> $$
> \bigl || \hat{A}\_0 B \hat{A}\_0^\top \bigr ||\_F^2  = \text{tr} \bigl \lbrace \hat{A}\_0 B \hat{A}\_0^\top \hat{A}_0 B^\top \hat{A}\_0^\top \bigr \rbrace  = \text{tr} \bigl \lbrace B (\hat{A}\_0^\top \hat{A}\_0) B^\top (\hat{A}\_0^\top \hat{A}\_0) \bigr \rbrace
> = \text{tr}\bigl \lbrace B B^\top \bigr \rbrace = \bigl || B \bigr ||_F^2.
> $$
>
> It follows that
>
> \begin{align*}
> d\_{\text{F}} \bigl( \mathcal{S}(\hat{A}\_0 \check{U}\_{1:d\_0}), \mathcal{S}(\hat{A}\_0 A\_0^{\ast}) \bigr)
> &= \frac{1}{\sqrt{2}} \bigl || \hat{A}\_0 \check{U}\_{1:d\_0} \check{U}\_{1:d\_0}^\top \hat{A}\_0^\top - \hat{A}\_0 A\_0^{\ast} (A\_0^{\ast})^\top \hat{A}\_0^\top \bigr ||\_F \\\\
> &= \frac{1}{\sqrt{2}} \bigl || \check{U}\_{1:d\_0} \check{U}\_{1:d\_0}^\top - A\_0^{\ast} (A\_0^{\ast})^\top \bigr ||_F \\\\
> &= d\_{\text{F}} \bigl( \mathcal{S}(\check{U}\_{1:d\_0}), \mathcal{S}(A\_0^{\ast}) \bigr). \qquad \qquad (2)
> \end{align*}
>
> Further, by an application of Theorem 1 to the data projected by $\hat{A}\_0$ and working conditionally on $\hat{A}\_0$, we have
> $$
> \mathbb{E} \bigl\lbrace d\_{\text{F}} \bigl( \mathcal{S}(\check{U}\_{1:d\_0}), \mathcal{S}(A\_0^{\ast}) \bigr) \bigm| \hat{A}\_0 \bigr\rbrace \leq 2d\_0^{1/2}||\Pi_{\ast}^{\infty} - A\_0^{\ast}(A\_0^{\ast})^{\top}||_{\mathrm{op}} +  \frac{2 (2\pi)^{1/2} d\_0^{1/2} d \cdot \hat{d}\_0}{L^{1/2}}. \qquad (3)
> $$
>
> Now consider the second term on the right hand side of (1). For simplicity of exposition, let $P:= \hat{A}\_0 A\_0^{\ast} (A\_0^{\ast})^\top \hat{A}\_0^\top$, $Q:= A\_0 A\_0^\top$. Both $P$ and $Q$ are symmetric, $P^2 = P$, $Q^2 = Q$, and $\text{tr}(PQ) = \text{tr}(QP)$. Further
> $$
> \text{tr} (Q) = \text{tr}( A\_0 A\_0^\top ) = \text{tr} (A\_0^\top A\_0 ) = \text{tr} (I\_{d\_0} ) = d\_0,
> $$
>
> and similarly
> $$
> \text{tr} (P) = \text{tr} \bigl \lbrace \hat{A}\_0 A\_0^{\ast} (A\_0^{\ast})^\top \hat{A}\_0^\top \bigr \rbrace  = \text{tr} \bigl \lbrace(A\_0^{\ast})^\top (\hat{A}\_0^\top \hat{A}\_0) A\_0^{\ast}\bigr\\rbrace = d\_0.
> $$
>
> It follows that
> \begin{align*}
> d\_\text{F}^2 \bigl( \mathcal{S}(\hat{A}\_0 A\_0^{\ast}), \mathcal{S} (A\_0) \bigr)
> &= \frac{1}{2} \bigl || P - Q \bigr ||_F^2 \\\\
> &= \frac{1}{2} \text{tr} ( P^2 - PQ -QP + Q^2 ) \\\\
> &= \frac{1}{2} \text{tr} (P + Q -2 PQ ) = d\_0 - \text{tr}( PQ ). \qquad (4)
> \end{align*}
>
> Recall that $A\_0^{\ast} \in \mathcal{A}\_{\hat{d}\_0 \times d\_0}$ is given by the QR decomposition of $M := \hat{A}\_0^\top A\_0$, with
> $$
> M = \hat{A}\_0^\top A\_0 = A\_0^{\ast} R, \qquad (5)
> $$
> where $R \in \mathbb{R}^{d\_0 \times d\_0}$ is some upper triangular matrix. Then solving (5) gives
> $$
> R = (A\_0^{\ast})^\top \hat{A}\_0^\top A\_0.
> $$
>
> Therefore
> $$
> \text{tr}(PQ)
> = \text{tr} \bigl \lbrace \hat{A}\_0 A\_0^{\ast} (A\_0^{\ast})^\top \hat{A}\_0^\top A\_0 A\_0^\top \bigr \rbrace
> = \text{tr} \bigl \lbrace(A\_0^{\ast})^\top \hat{A}\_0^\top A\_0 A\_0^\top \hat{A}\_0 A\_0^{\ast} \bigr \rbrace
> = \text{tr} ( R R^\top ). \qquad (6)
> $$
>
> On the other hand, recall from the definition of false negative measure $d\_{\text{FN}} (\hat{A}\_0, A\_0)$:
> \begin{align*}
> d\_{\text{FN}}^2 (\hat{A}\_0, A\_0)
> &= \bigl || (I - \hat{A}\_0 \hat{A}\_0^{\top}) A\_0 \bigr ||_F^2 = \text{tr} \bigl\lbrace A\_0^\top (I - \hat{A}\_0 \hat{A}\_0^{\top})^2 A\_0 \bigr\rbrace \\\\
> &= \text{tr} \bigl\lbrace A\_0^\top (I - \hat{A}\_0 \hat{A}\_0^{\top}) A\_0 \bigr\rbrace \\\\
> &= \text{tr} \bigl\lbrace A\_0 A\_0^\top \bigr\rbrace - \text{tr} \bigl\lbrace A\_0 A\_0^\top \hat{A}\_0 \hat{A}\_0^\top \bigr\rbrace \\\\
> & = d\_0 - \text{tr}( MM^\top). \qquad (7)
> \end{align*}
>
> Finally, we have
> $$
> \text{tr}(M M^\top )
> = \text{tr} \bigl\lbrace A\_0^{\ast} R R^\top (A\_0^{\ast})^\top \bigr\rbrace
> = \text{tr} \bigl\lbrace R R^\top (A\_0^{\ast})^\top A\_0^{\ast} \bigr\rbrace
> = \text{tr} (R R^\top). \qquad(8)
> $$
> So, by (4), (6), (7), and (8), we have
> $$
> d\_\text{F} \bigl( \mathcal{S}(\hat{A}\_0 A\_0^{\ast}), \mathcal{S} (A\_0) \bigr) = d\_{\text{FN}} (\hat{A}\_0, A\_0) \qquad(9)
> $$
> Now taking conditional expectation on $\hat{A}_0$ of both sides of (1), in combination with (3) and (9) completes the proof.

---

> ### Author Response · Authors · 2025-11-20
> **Author Responses - Part III**
>
> For your convenience, we also report the new simulation results as follows (large $p$ settings):
> | Model | $p$ | $n$ | MAVE         | gKDR         | drMARS            | RPE              | RPE2             |
> |-------|-----|-----|-----------------|----------------|-----------------------|---------------------|---------------------|
> | 1     | 200 | 50  | $0.97_{0.06}$   | $0.98_{0.02}$  | $\mathbf{0.31_{0.27}}$ | $0.64_{0.19}$       | $\mathbf{0.35_{0.30}}$ |
> | 1     | 200 | 200 | $0.97_{0.04}$   | $0.99_{0.01}$  | $0.23_{0.21}$         | $0.55_{0.16}$       | $\mathbf{0.06_{0.02}}$ |
> | 1     | 500 | 50  | $0.98_{0.05}$   | $0.99_{0.01}$  | $\mathbf{0.34_{0.31}}$ | $0.83_{0.18}$       | $0.78_{0.26}$       |
> | 1     | 500 | 200 | $0.98_{0.03}$   | $0.99_{0.01}$  | $0.22_{0.19}$         | $0.61_{0.15}$       | $\mathbf{0.08_{0.03}}$ |
> | 2     | 200 | 50  | $0.96_{0.05}$   | $1.00_{0.00}$  | $0.75_{0.31}$         | $\mathbf{0.26_{0.37}}$ | $\mathbf{0.30_{0.38}}$ |
> | 2     | 200 | 200 | $0.92_{0.04}$   | $0.98_{0.02}$  | $0.27_{0.19}$         | $\mathbf{0.01_{0.00}}$ | $\mathbf{0.01_{0.00}}$ |
> | 2     | 500 | 50  | $0.97_{0.04}$   | $1.00_{0.00}$  | $0.85_{0.24}$         | $\mathbf{0.33_{0.42}}$ | $\mathbf{0.34_{0.42}}$ |
> | 2     | 500 | 200 | $0.95_{0.03}$   | $0.99_{0.01}$  | $0.33_{0.25}$         | $\mathbf{0.01_{0.00}}$ | $\mathbf{0.01_{0.00}}$ |
> | 3     | 200 | 50  | $1.19_{0.09}$   | $1.39_{0.01}$  | $1.21_{0.18}$         | $\mathbf{0.83_{0.32}}$ | $\mathbf{0.83_{0.32}}$ |
> | 3     | 200 | 200 | $1.13_{0.07}$   | $1.32_{0.03}$  | $1.11_{0.19}$         | $\mathbf{0.13_{0.09}}$ | $\mathbf{0.13_{0.09}}$ |
> | 3     | 500 | 50  | $1.22_{0.09}$   | $1.41_{0.00}$  | $1.21_{0.18}$         | $\mathbf{0.94_{0.23}}$ | $\mathbf{0.93_{0.23}}$ |
> | 3     | 500 | 200 | $1.16_{0.07}$   | $1.38_{0.01}$  | $1.15_{0.19}$         | $\mathbf{0.25_{0.22}}$ | $\mathbf{0.25_{0.22}}$ |
> | 4     | 200 | 50  | $\mathbf{0.69_{0.16}}$ | $0.98_{0.01}$ | $0.84_{0.20}$ | $0.80_{0.09}$       | $0.78_{0.17}$       |
> | 4     | 200 | 200 | $0.48_{0.06}$   | $0.82_{0.06}$  | $0.85_{0.16}$         | $0.74_{0.08}$       | $\mathbf{0.17_{0.15}}$ |
> | 4     | 500 | 50  | $\mathbf{0.77_{0.16}}$ | $0.99_{0.00}$ | $0.85_{0.15}$ | $0.84_{0.08}$       | $0.85_{0.11}$       |
> | 4     | 500 | 200 | $0.54_{0.06}$   | $0.94_{0.04}$  | $0.86_{0.15}$         | $0.79_{0.04}$       | $\mathbf{0.25_{0.24}}$ |
> | 5     | 200 | 50  | $1.29_{0.07}$   | $1.39_{0.01}$  | $1.29_{0.10}$         | $\mathbf{1.21_{0.12}}$ | $\mathbf{1.21_{0.12}}$ |
> | 5     | 200 | 200 | $1.20_{0.04}$   | $1.29_{0.03}$  | $1.30_{0.13}$         | $\mathbf{1.00_{0.00}}$ | $\mathbf{1.00_{0.00}}$ |
> | 5     | 500 | 50  | $1.31_{0.08}$   | $1.41_{0.00}$  | $1.27_{0.14}$         | $\mathbf{1.23_{0.13}}$ | $\mathbf{1.23_{0.13}}$ |
> | 5     | 500 | 200 | $1.24_{0.04}$   | $1.36_{0.01}$  | $1.30_{0.11}$         | $\mathbf{1.00_{0.00}}$ | $\mathbf{1.00_{0.00}}$ |
> | 6     | 200 | 50  | $1.57_{0.10}$   | $1.71_{0.01}$  | $\mathbf{1.48_{0.19}}$ | $1.52_{0.19}$       | $1.52_{0.19}$       |
> | 6     | 200 | 200 | $1.59_{0.06}$   | $1.71_{0.01}$  | $1.57_{0.19}$         | $\mathbf{1.39_{0.28}}$ | $\mathbf{1.39_{0.28}}$ |
> | 6     | 500 | 50  | $1.64_{0.09}$   | $1.72_{0.00}$  | $\mathbf{1.51_{0.17}}$ | $1.61_{0.16}$       | $1.61_{0.16}$       |
> | 6     | 500 | 200 | $1.60_{0.06}$   | $1.72_{0.00}$  | $1.57_{0.20}$         | $\mathbf{1.50_{0.19}}$ | $\mathbf{1.50_{0.19}}$ |
> | 7     | 200 | 50  | $1.38_{0.06}$   | $1.40_{0.01}$  | $\mathbf{1.27_{0.17}}$ | $1.33_{0.14}$       | $1.33_{0.14}$       |
> | 7     | 200 | 200 | $1.34_{0.06}$   | $1.40_{0.01}$  | $1.18_{0.21}$         | $\mathbf{1.00_{0.24}}$ | $\mathbf{0.99_{0.25}}$ |
> | 7     | 500 | 50  | $1.39_{0.06}$   | $1.41_{0.00}$  | $\mathbf{1.23_{0.18}}$ | $1.35_{0.12}$       | $1.35_{0.12}$       |
> | 7     | 500 | 200 | $1.36_{0.04}$   | $1.41_{0.00}$  | $1.17_{0.20}$         | $\mathbf{1.03_{0.26}}$ | $\mathbf{1.03_{0.26}}$ |
> | 8     | 200 | 50  | $1.18_{0.11}$   | $1.39_{0.01}$  | $0.79_{0.31}$         | $\mathbf{0.54_{0.38}}$ | $\mathbf{0.54_{0.38}}$ |
> | 8     | 200 | 200 | $1.08_{0.06}$   | $1.37_{0.02}$  | $0.88_{0.21}$         | $\mathbf{0.10_{0.02}}$ | $\mathbf{0.10_{0.02}}$ |
> | 8     | 500 | 50  | $1.21_{0.09}$   | $1.41_{0.00}$  | $0.83_{0.30}$         | $\mathbf{0.74_{0.35}}$ | $\mathbf{0.74_{0.35}}$ |
> | 8     | 500 | 200 | $1.11_{0.07}$   | $1.40_{0.01}$  | $0.89_{0.19}$         | $\mathbf{0.18_{0.14}}$ | $\mathbf{0.18_{0.14}}$ |
> | 9     | 200 | 50  | $1.30_{0.07}$   | $1.39_{0.01}$  | $1.22_{0.22}$         | $\mathbf{1.09_{0.16}}$ | $\mathbf{1.09_{0.15}}$ |
> | 9     | 200 | 200 | $1.26_{0.05}$   | $1.35_{0.02}$  | $1.25_{0.20}$         | $\mathbf{0.34_{0.39}}$ | $\mathbf{0.34_{0.39}}$ |
> | 9     | 500 | 50  | $1.32_{0.07}$   | $1.41_{0.00}$  | $1.24_{0.22}$         | $\mathbf{1.13_{0.20}}$ | $\mathbf{1.13_{0.20}}$ |
> | 9     | 500 | 200 | $1.28_{0.05}$   | $1.39_{0.01}$  | $1.22_{0.25}$         | $\mathbf{0.28_{0.37}}$ | $\mathbf{0.28_{0.37}}$ |

---

> ### Author Response · Authors · 2025-11-20
> **Author Responses - Part IV**
>
> **4. Algorithm 3 fixes $L=200$ without justification. Is this choice empirically optimal, or merely heuristic? How sensitive is performance to $L$ across different $n$, $p$, $d$?**
>
> We set $L = 200$ in Algorithm 3 based on our experiments in Section A.2 for Algorithm 1 (which lead to our default recommendation to set $L = 200$ in Algorithm 1), and on the close relationship between Algorithms 1 and 3. Indeed, one way of viewing Algorithm 3 is that it re-applies Algorithm 1 a second time to the original output gained from Algorithm 1. This is also why we set $M = 10\hat{d}\_0$ and $d = \lceil \hat{d}\_0 \rceil$ in line 6 of Algorithm 3. We opted to simplify the presentation of Algorithm 3 by fixing these choices based on our experiments for Algorithm 1. There is, however, flexibility in the choices of these parameters, and alternatives may lead to a different trade-off between computational and statistical performance. While we have not provided a full exploration of these choices specifically for Algorithm 3, we'd expect that the sensitivity to $n$, $p$ and $d$ to follow similar patterns to those we observe for Algorithm 1.
>
> **5. While the paper repeatedly emphasizes the flexibility and efficiency of the proposed method, it lacks an explicit characterization of computational complexity, The authors are encouraged to clearly specify the time and memory complexity under typical parameter settings and provide a comparison with established methods like MAVE and drMARS to better illustrate the computational advantages and trade-offs of the proposed approach.**
>
> Thank you for raising this point -- we recognise the importance of the trade-offs between computational complexity and predictive performance.  In much our paper we focus on latter, and provide default recommendations for the inputs to our main algorithms based on this.  We appreciate that this means our proposal can be somewhat slow computationally in comparison to existing methods -- see appendix C.3.
>
> There are many ways in which we can improve the computational cost of the recommended default version of our method. For instance, as discussed in Section A.3 using a simpler (e.g linear) base regression method significantly reduces the computational cost, though this may come at the expense of poorer performance in cases where there is no linear true signal (e.g. as is the case in models 1 and 3 in Section A.1). For these models, a quadratic least squares method is effective (see Figure 7) at modest computational cost.  Other ways to reduce the overall computational cost include taking $L$ and/or $M$ to be smaller -- see the **new Figure 16** in Appendix C.3, which shows that in many cases we can obtain competitive results with the existing methods with modest computational resources, even in a relatively `large-scale' setting of $n=500$, $p=100$. Notably, RPE-based methods also show better performance than drMARS for model 5-7 when we set a large $L$, different from the results given by Table 3-4 when we just used the default settings, but further illustrating the strength of our method. In fact, the plots in Figure 16 were produced by setting $M=5p$, and set a grid for different values of $L$. We also now include further discussion of these points in Section C.3 of the revised manuscript.
>
> The main contribution of our paper is to introduce a new and flexible framework for dimension reduction based on an ensemble of random projections. While the consideration of the tradeoffs between computational costs and predictive performance is important, we feel that including a full and thorough investigation of these in the main manuscript is beyond the scope of the work and may detract from our primary contribution.
>
> **6. Although Appendix D mentions real-data experiments, the main text presents only simulation results. It is recommended to include at least one illustrative real-data application in the main paper to enhance the work’s practical relevance and overall impact.**
>
> Thank you for your recommendation to include the real-data experiments in the main text, with which we agree in principle.  However, since the oracle projection $A_0$ is unknown in these real data examples, we rely on regression performance after projection as a measure of the quality of the different methods, which is somewhat indirect for our main goal of estimating $\mathcal{S} (A_0)$. In addition, we have already added substantial material to the main text (workflow diagram, revised contributions, new high-dimensional experiments, and description direct to the theoretical and numerical results in the appendices) based on other helpful suggestions, so, for space reasons, we feel it is more appropriate to keep the real-data analysis in the appendix. We hope this has addressed your concern.

---

> ### Author Response · Authors · 2025-11-20
> **Author Responses - Part V**
>
> ### **Questions**
>
> **1. In Algorithm 2 (line 6), how is the threshold $T_\ell > 1/2$ chosen? Could alternative thresholds (other than $1/2$) be used?**
>
> Our recommendation to use $T_{\ell} > 1/2$ was to provide a concrete automatic suggestion for selecting the output projection dimension.  Other thresholds could certainly be used, and increasing the threshold would lead to a lower dimensional output projection -- this is desirable as we would like to reduce the dimension as much as possible, but if the threshold is too high we may of course miss important projection directions (i.e. increased false negatives) . On the other hand, a lower threshold would lead to a higher output dimension and may ultimately lead to "noise" directions being included (i.e increased false positives).
>
> As demonstrated in the numerical experiments for unknown $d_0$ in Appendix C.2 and Figure 13, our suggestion leads to a good balance between false positives and false negatives.
>
> In practice, depending on how involved the practitioner is in the analysis,  one may also choose to inspect the output of Algorithm 2 (as in Figure 5) and make an informed choice from such plots -- this is similar for example, to how one would choose the number of principal components to retain in PCA.
>
> **2. In Algorithm 3, $L=200$ is used while many experiments set $n=200$, which may suggest nontrivial computational cost. Can a smaller $L$ be employed without materially degrading performance? If so, is there guidance or a sensitivity analysis for selecting $L$ as a function of (n,p,d)?**
>
> Thank you for raising this point — we hope that our responses to your comments in weaknesses 4 and 5 above have addressed this concern. We would of course be very happy to answer any further questions you may have.
>
> **3. Beyond Gaussian and Cauchy projections, are other projection families, e.g., sub-Gaussian or Hadamard-structured transforms, compatible with the framework?**
>
> This is great point! In fact, any projection distribution can be used within our framework. As with many aspects of the method, there are tradeoffs between predictive performance, computational cost and possibly interpretability.  For instance, one could use "sparse" binary random projections (i.e., with a small proportion of non-zero entries in $\lbrace+1,-1\rbrace$). These would be computationally cheap (relative to our default recommendation), and may yield sparse output projections, which aid interpretability. However, in cases where the true $A_0$ is dense, such an approach would likely be less effective.
>
> **4. I recommend adding a framework diagram to illustrate the workflow and the relationships among Algorithms 1–3.**
>
> This is a great suggestion — thank you! We have now included a new diagram (Figure 2) in the introduction to illustrate the workflow and the relationships among Algorithms 1–3.

---

> ### Author Response · Authors · 2025-11-28
> **Author Responses – Thanks!**
>
> Dear Reviewer 8gwY,
>
> We would like to sincerely thank you again for the time and effort you invested in reviewing our paper, and for the many constructive and detailed suggestions you provided. We carefully considered all of your comments and have incorporated them into our revised manuscript where possible.
>
> We are very grateful for your thoughtful feedback, which has helped us to significantly improve the quality and clarity of the work.

---

### Author Response · Authors · 2025-11-27
**Summary of revisions**

We sincerely thank all reviewers for the comprehensive and insightful feedback, and for the time spent reviewing. The feedback has been incorporated to improve our paper.

Specifically, we have added or edited the following in our revised paper:

 - **New Theorem**: We are pleased to provide a new result about the output of Algorithm 3, which is shown in **Appendix A.5** as **Theorem 2**, the proof is given in **Appendix B**. The bound includes a new term $d_{\text{FN}}$, which we also used as a complementary metric for the method in Appendix C.2. This term vanishes if none of the true signal is missed by the first application of Algorithm 1. Thanks for the constructive suggestion by Reviewer [8gwY].

- **New Experiments**: We are pleased to report that we have conducted several further simulation experiments with $p \in \lbrace200, 500\rbrace$ and $n \in \lbrace50, 200\rbrace$, and our method shows even more pronounced gains over competitors in these settings! We have extended **Tables 2, 3 and 4 in Appendix C.1** to include these results. **Tables 5 and 6 in Appendix C.2** were also updated. We also added the boxplots of the results for $n=200$, $p=500$ in the main text - see **Figure 7**. Thanks for the constructive suggestion by Reviewers [8gwY, MBR7, FpME].

- **New contribution subsection**: we have updated the introduction to better outline our main contributions, now the contribution subsection only states the new ideas, and removes the details of the method mechanics. Thanks for the constructive suggestion by Reviewer [FpME].

- **New workflow diagram**: we have included a new diagram (**Figure 2**) in the introduction to illustrate the workflow and the relationships among Algorithms 1–3. Thanks for the constructive suggestion by Reviewer [8gwY].

- **Updated related work**: we have added three new related works in the related work section in the introduction. Thanks for the constructive suggestion by Reviewer [FpME].

- **New practical considerations related to the choice of $n_1 / n$**: we have added a sensitivity study varying $n_1 / n \in \lbrace 1/2, 2/3, 3/4, 4/5 \rbrace$ in the new **Appendix A.3**. The performance of our algorithm is relatively stable as $n_1 / n$ varies across the range. Thanks for the constructive suggestion by Reviewer [FpME].

- **New important relationship between $d\_{\text{F}}$ and the complementary metrics $d\_{\text{FN}}$ and $d\_{\text{FP}}$**: we have added the important identity
$$
2d\_{\mathrm{F}}^2(\hat{A}\_0, A\_0) = d\_{\mathrm{FP}}^2(\hat{A}\_0, A\_0) + d\_{\mathrm{FN}}^2(\hat{A}\_0, A\_0).
$$
in **Appendix C.2**. Therefore, we separately report $d\_{\text{FN}}$ and $d\_{\text{FP}}$ instead of $d\_\text{F}$ to provide a clearer picture of the tradeoffs between the false positives and false negatives that the methods incur when $d_0$ is unknown. Some discussion of the bahaviour of $d\_{\text{FN}}$ and $d\_{\text{FP}}$ are also added. Thanks for the constructive suggestion by Reviewer [FpME].

- **New discussion related to computational considerations**: we have added discussion for the case where computational resources are limited in **Appendix C.3**. There are various ways we can significantly reduce the computational cost. The new **Figure 16** shows that in many cases we can obtain the best results compared to existing methods with modest computational resources. Thanks for the constructive suggestion by Reviewers [8gwY, MBR7].

We greatly appreciate all the feedback; this has truly allowed us to make important improvement to our work.

---

### Meta-Review · Area_Chair_ed6p · 2026-01-06

**Summary:**

This paper proposes a sufficient dimension reduction (SDR) method using random projection together with ensemble. All reviewers found the method novel, the theory solid, and the experiments convincing. Although there were quite a few concerns from the reviewers, most have been addressed by the authors' rebuttal, see details below. I recommend to accept this paper although there is one reviewer with rating 4, as I believe the method is very neat: it is not common to see a simple idea that works surprisingly well on this very old topic in 2026. As agreed by the reviewers, the method is pretty different from existing SDR method from many aspects. That being said, there are some remaining issues where I strongly encourage the authors either to further address them in the camera-ready version or discussion them as future work:

1. The proposed method has the highest computational complexity among the discussed method. Further effort for scalability is expected.
2. There are multiple tuning parameters, $L,M,d,n_1$, which make the method hard to use in practice especially to scientific practitioners without strong background in SDR. Existing theory primarily focuses on $L$, but further theoretical study on other parameters may give insights about tuning parameters in theory, and further comprehensive empirical studies may provide better, easy-to-use default choices to general audience.

**Reviewer Concerns:**

Reviewer 8gwY:
1. Assumption for L (misunderstanding, addressed)
2. High-dimensional setup: p < n (addressed by additional experiments)
3. Theorem for Algorithm 3 (addressed by new theorem in Appendix A.5)
4. Lack of computational complexity (addressed by new results in Appendix C.3)
5. Lack of real data analysis (addressed by further explanation)

Reviewer MBR7:
1. Superiority of the proposed method (addressed by further explanation)
2. Tuning parameters (somewhat addressed by showing robustness and giving default choices)
3. Long runtime (somewhat mitigated)
4. Lack of literature review (addressed by expanding related work section)
5. Lack of comparison of theoretical bound (addressed by further explanation)

Reviewer FpME:
1. The contribution section is too long (addressed by shortening the section)
2. No guidance on choosing $n_1$ & missing sensitivity analysis of $n_1$ (addressed by adding experiments in Appendix A.3)
4. Missing reference (addressed by adding refs)
5. p>n (addressed by new experiments)
6. Independence in proof of Theorem 1 (addressed by further explanation that data are assumed to be fixed in this setup)
7. Missing Frobenius distance for unknown $d_0$ settings (addressed by explanation the reported distance is closely related to the Frobenius distance)

**Reviewer Scores:**

Reviewer 8gwY: 6 --> 7

Reviewer MBR7: 4 --> 4

Reviewer FpME: 6 --> 7

---

### Decision · Program_Chairs · 2026-01-26

Accept (Poster)